# Earthquake lubrication and healing explained by amorphous nanosilica

Christie D. Rowe [1], Kelsey Lamothe[1], Marieke Rempe[2,7], Mark Andrews[3], Thomas M. Mitchell[4], Giulio Di Toro [2,5], Joseph Clancy White[6] & Stefano Aretusini[5]

During earthquake propagation, geologic faults lose their strength, then strengthen as slip slows and stops. Many slip-weakening mechanisms are active in the upper-mid crust, but healing is not always well-explained. Here we show that the distinct structure and rate-dependent properties of amorphous nanopowder (not silica gel) formed by grinding of quartz can cause extreme strength loss at high slip rates. We propose a weakening and related strengthening mechanism that may act throughout the quartz-bearing continental crust. The action of two slip rate-dependent mechanisms offers a plausible explanation for the observed weakening: thermally-enhanced plasticity, and particulate flow aided by hydrodynamic lubrication. Rapid cooling of the particles causes rapid strengthening, and inter-particle bonds form at longer timescales. The timescales of these two processes correspond to the time-scales of post-seismic healing observed in earthquakes. In natural faults, this nanopowder crystallizes to quartz over 10s–100s years, leaving veins which may be indistinguishable from common quartz veins.

[1] Earth and Planetary Sciences, McGill University, Montréal, QC H3A 0E8, Canada. [2] Dipartimento di Geoscienze, Università degli Studi di Padova, Via Gradenigo 6, 35131 Padova, Italy. [3] Department of Chemistry, McGill University, 801 Sherbrooke St. W., Montréal, QC H3A 0B8, Canada. [4] Rock and Ice Physics and UCL Seismological Laboratory, Earth Sciences Department, University College London, Gower Street, London WC1E 6BT, UK. [5] Sezione di Tettonofisica e Sismologia, Istituto Nazionale di Geofisica e Vulcanologia, Roma, Italy. [6] Department of Earth Sciences, University of New Brunswick, Fredericton, NB E3B 5A3, Canada. [7] Present address: Institute for Geology, Mineralogy, and Geophysics, Ruhr-Universität Bochum, Universitätsstr. 150, 44780 Bochum, Germany. Correspondence and requests for materials should be addressed to C.D.R. (email: christie.rowe@mcgill.ca)

**S**lip weakening is fundamental to all earthquakes[1], but there is no known universal mechanism for controlling the loss of rock strength[2,3]. The lack of positive heat flow anomalies around major continental faults suggests that the shear resistance on faults may be extremely low during displacement, but paradoxically, stresses near the faults may be high[4]. The proposal of "silica gel" lubrication, thought to cause extreme loss of shear resistance in high velocity friction experiments on quartz-rich rock, offered a tantalizing explanation for dramatic coseismic weakening which might apply broadly within continental crust[5,6]. The weakening effect is apparently associated with the availability of water, and is potentially inhibited when experiments are performed under dry conditions, although the specific dependency may be complicated[7,8]. In experiments, the strength of sheared interfaces recovers over minutes to hours after shearing[5,6,9,10], but the causes and mechanisms remain enigmatic. We performed shearing experiments on quartz-rich rock, but did not find evidence for silica gel, where gel is a "jellylike substance formed by a colloidal solution in its solid phase"[11]. Here we show the composition and structure of the frictional wear material and propose an alternative explanation for the lubricating and healing behaviors.

## Results

**Friction experiments.** Novaculite (chert) cores were acquired from Dan's Whetstone Company, Percy, Arkansas, USA. The novaculite occurs in the whetstone quarry as cherty nodules within low-grade limestone. The novaculite consists of a porous aggregate of euhedral ~10 μm quartz grains with traces of calcite and oxide minerals, and abundant fluid inclusions along grain boundaries and healed fractures. The friction experiments were conducted at the Department of Geosciences of the Universit'a degli Studi di Padova, Padua, Italy, using the ROtary-Shear Apparatus ROSA. Two cylinders were sheared against each other at atmospheric humidity. After the experiment, the sample halves were recovered from the machine as one in order to preserve the wear material.

Shearing experiments were performed at equivalent velocities (as defined by [12]) of 100 μm/s, 1 mm/s, 1 cm/s, and 10 cm/s at normal stresses of 2.5 MPa for total displacements of 3 m and 30 m (Table 1). Weakening was observed in every experiment, and most experiments achieved a steady state friction value after > 0.3 m of slip. The friction coefficient evolved from ~0.7 gradually to a steady state value of ~0.45–0.6 in the 100 μm/s, 1 mm/s, and 1 cm/s experiments, and decreased dramatically to < 0.1 in the 10 cm/s experiments (Fig. 1a).

**Table 1 Summary of friction experiments and calculated temperature rise from FEM model**

| Sample run | Equivalent velocity (m/s) | Total slip (m) | Normal stress (MPa) | Bulk $\Delta T$ (°C) from FEM model | Bulk $\Delta T$ (°C) from half-space analytical solution | Peak $\Delta T_{flash}$ (°C) |
|---|---|---|---|---|---|---|
| 94 (failed) | 0.1 m/s in 10 s (velocity ramp) | 1 | 5 | – | – | – |
| 95* | 0.1 m/s in 2 s (velocity ramp) | 1 | 2.5 | – | – | – |
| 96 | 0.1 | 3 | 2.5 | 28.5 | 42.5 | 1006 |
| 97* | 0.1 | 3 | 2.5 | 23.6 | 45 | 1006 |
| 98* | 0.1 | 30 | 2.5 | – | – | 1006 |
| 99 | 0.1 | 3 | 2.5 | – | – | 1006 |
| 100 | 0.1 | 3 | 2.5 | – | – | 1006 |
| 101* | 0.1 | 3 | 2.5 | – | – | 1006 |
| 102* | 0.1 | 30 | 2.5 | – | – | 1006 |
| 110 (failed) | 0.001 | 3 | 2.5 | – | – | – |
| 111* | 0.01 | 3 | 2.5 | 38.0 | 60 | 318.2 |
| 112* | 0.0001 | 3 | 2.5 | 2.8 | 7 | 31.8 |
| 123* | 0.001 | 3 | 2.5 | 7.5 | 18 | 100.6 |

*denotes runs whose wear material was characterized in detail with one or more of TEM, SEM, FT-IR, and Raman Spectroscopy

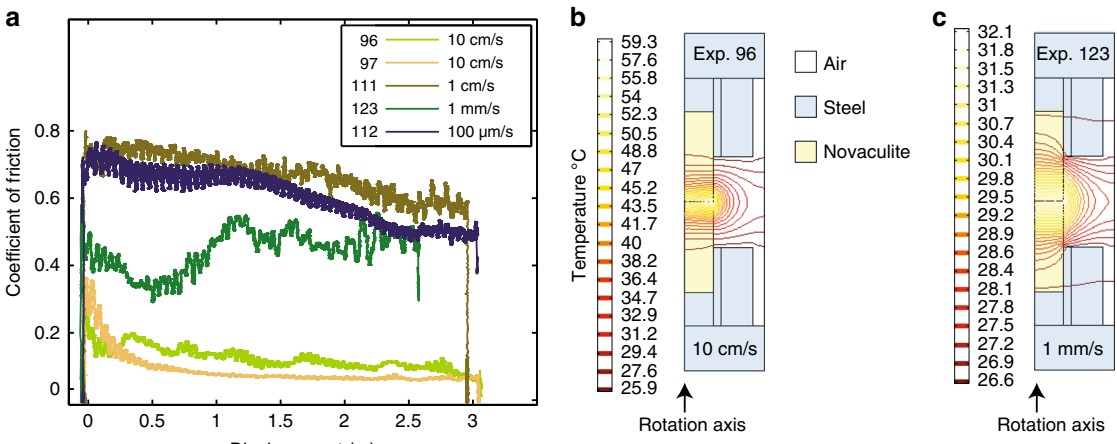

**Fig. 1** Results of friction and numerical experiments. **a** Coefficient of friction vs. displacement in novaculite friction experiments. **b** Temperature distribution from finite element model at the end of the experiments for fastest slip rate (10 cm/s, experiment 96) and **c** slower experiment (1 mm/s, experiment 123)

We estimated the bulk temperature rise of the slipping zone during the experiments using two methods: a finite element model with Comsol Multiphysics and a half-space analytical solution[13]. Fig. 1b, c show the geometry of the half-model, with novaculite in the center (density = 2750 kgm$^{-3}$, thermal diffusivity = 5 m$^2$s$^{-1}$, heat capacity = 0.75 × 10$^5$ J (kg°C)$^{-1}$); steel frame (density = 8000 kgm$^{-3}$, thermal diffusivity = 16 m$^2$s$^{-1}$, heat capacity = 0.5 × 10$^5$ J (kg°C)$^{-1}$), in contact with air (density = 0 kg/m$^3$, thermal diffusivity = 1 m$^2$s$^{-1}$, J (kg°C)$^{-1}$). The Comsol model incorporated time-varying friction measured during the experiment as an input. Both approaches show average slipping zone temperature rises of ≤ 60° in every experiment (Table 1). We also estimated the peak temperature at asperities during flash heating using the approach of Violay et al.[14] for a range of slip rates and asperity sizes. The results of temperature calculations are presented in Figs. 1, 2 and Table 1.

**Microstructure and composition of wear material.** The starting material (novaculite) is composed mostly of ~10–50 μm quartz with trace amounts of calcite and has rough-walled pores about 100 μm in diameter (Fig. 3a). In all experiments, wear material on the sheared interface formed a ~5–30 μm-thick layer of < 1 μm ellipsoidal particles (Fig. 3b, c, Fig. 4). High sphericity is characteristic of amorphous silica nanoparticles[15], so rolling is not required to explain their shape. The particles form clusters up to 100–1000 nm across (Fig. 3) which explode under the electron beam, suggesting the presence of volatiles (Fig. 3c; compare to[16]). The nanopowder layer is cut by discrete slip surfaces ~1 μm thick (Fig. 3b, white arrows). When the novaculite cores are opened to expose the sliding surface in plan view (Fig. 3c–f), the thin localized shears are revealed to be formed of discrete dense plates with smooth surfaces marked by slip-parallel striations, inter-layered within the nanopowder layer (Fig. 3e–f). We interpret these plates as the remnants of discrete slip surfaces. Fragments of plates are found in the inter-plate nanopowder (Fig. 3f), showing

that development, breaking and reforming of striated plates occurred during a single experiment. The sliding surface of the novaculite underneath the wear material displays a shiny polish (Fig. 3d, ss). This polish is due to smoothing of the novaculite surface by initial abrasion, as well as by smearing and adhesion of nanopowder across the novaculite to fill surface pores. We observe no evidence of viscous fluids on the slip surfaces, so no evidence supports the formation of gel or melt during the experiments (consistent with[9], c.f [3]).

We used transmission electron microscopy (TEM) to examine the crystallinity and nanostructure of the wear material (Fig. 4). The nanopowder is predominantly composed of 100–1000 nm clumps of rounded to ellipsoidal amorphous nanoparticles, ~10–100 nm in diameter, and chips of quartz (Fig. 4, similar to[9]). The relict quartz grains are larger (≤1 μm; Fig. 4a, b, white arrows), and have stepped, fractured edges showing damage caused by comminution and frictional wear. The powder layer contains angular fragments of more densely packed particles (Fig. 4c) which correspond to the shiny striated plates (Fig. 3e–f). At the sub-micron scale, it is clear that the sharp shiny plate is formed from gradational packing of the same amorphous nanoparticles, with increasing packing density toward the shiny slip surface (along white arrows in Fig. 4c). The fragments of broken slip surfaces are rotated relative to one another (as demonstrated by opposite directions of gradational packing in the two fragments of Fig. 4c), confirming that the slip surfaces formed and were broken up into fragments during sliding.

Raman spectroscopy was employed to characterize Si–O bonding in the nanopowder for comparison to crystalline and amorphous species of silica, and investigate the form of any water present. We used a 633 nm He–Ne excitation laser Raman microprobe spectrometer in the Materials Chemistry lab at McGill University. Spectra were collected from the powdered novaculite (starting material), the wear material, and a control nanosilica (a lab-made 2 μm nanosilica powder, Nyacol Nyasil 5; Fig. 5a). Shear-related structures were also interrogated without physical disturbance by using Confocal Raman microspectroscopy in situ (Fig. 6). Spectra were collected at evenly spaced points along transects from core to rim across the samples (Fig. 6).

The novaculite is an exact match for α-quartz[17]. The wear material spectra show a broad peak with weak structure in the region 100–600 cm$^{-1}$ punctuated by sharp peaks due to minor contributions from E (phonon) and A1 modes in α-quartz from the exposed patches of novaculite beneath the wear layer (Fig. 3c, ss; Fig. 5a; Fig. 6). Compared with typical signatures of amorphous silica, the classic modes associated with oligomeric ring structures are absent in our wear material (red bands in Fig. 5a), indicating that a variety of Si–O species are present with no dominant modes (see Supplementary Discussion for additional description). The strong polarization-dependence across all wavelengths (Fig. 5a) indicates that the wear material displays anisotropy, perhaps related to the shear-parallel smearing observed in the striated plates and on the slip surface (Fig. 3d–e). No silanol (Si–OH) or adsorbed water were observed by Raman (scans out to 4000 cm$^{-1}$, not shown in Fig. 5a; [18]). This spectrum is unlike previously published Raman spectra of glassy silica[15] or commercial nanosilica we analyzed (see Supplementary Discussion for details). In summary, the nanosilica wear material formed by shearing of quartz-rich rocks is amorphous but anisotropic, is anhydrous within the detection limit of Raman spectroscopy, and has a unique Raman fingerprint, distinguishable from amorphous silica from other sources.

The quantity of wear material on the slip surface varied with slip rate (Fig. 6), for experiments with the same amount of slip (Table 1). Some wear material was ejected and lost during the

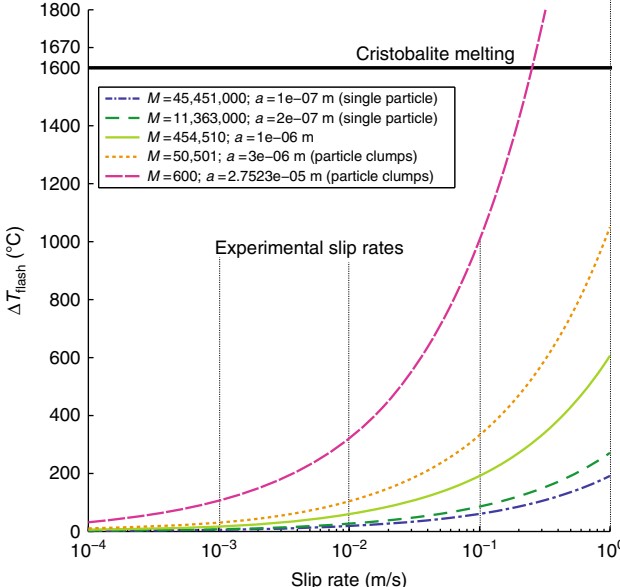

**Fig. 2** Flash heating temperature at asperities. Temperatures calculated for slip rates from 10$^{-4}$–10$^0$ m/s at normal stress = 2.5 MPa, showing the effect of asperity size ($a$) and number ($M$), between the size of single particles (10 nm) to the size of particle clumps (100s μm). The maximum flash heating temperature for any of our experiments is estimated to be ~1000 °C. Melting temperature of cristobalite is plotted for reference

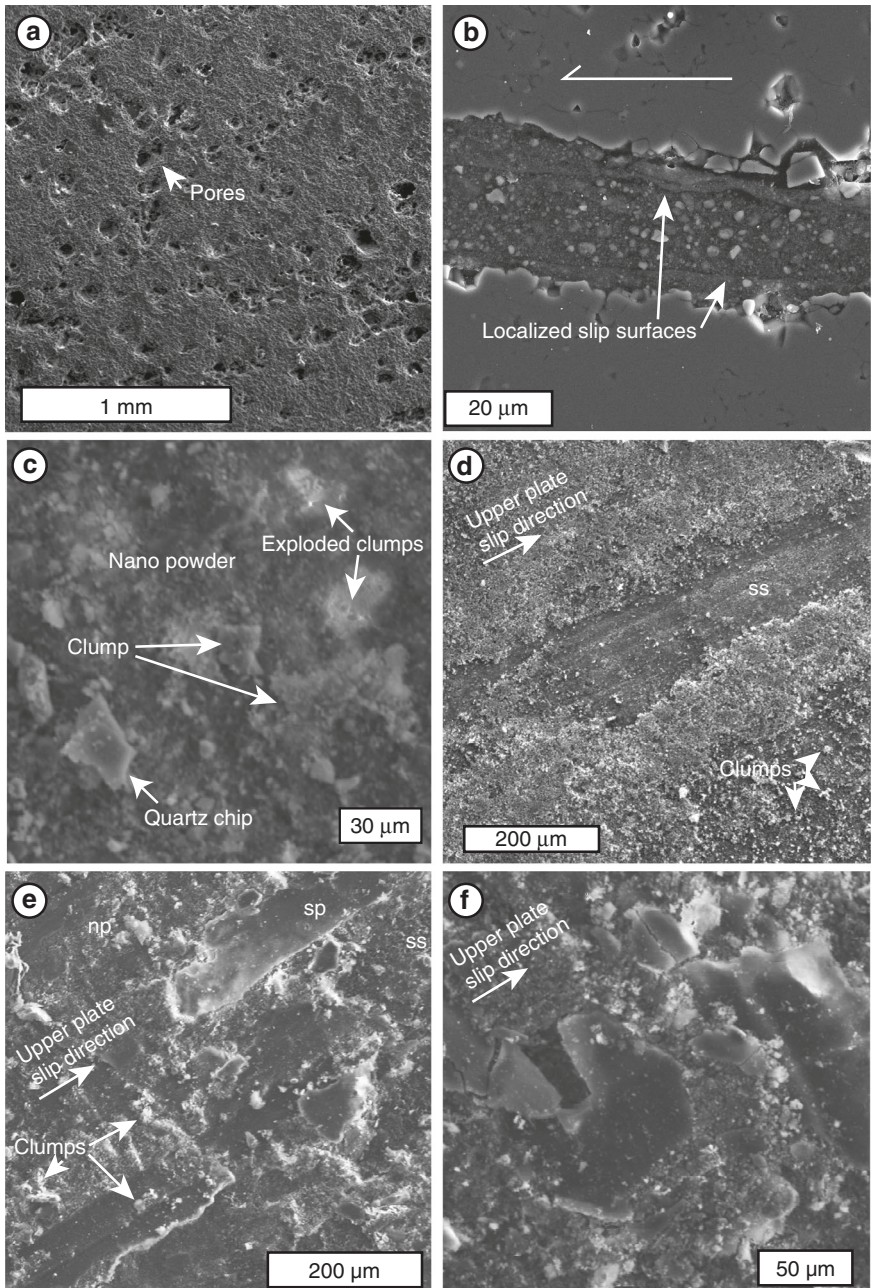

**Fig. 3** Secondary electron images of microstructures in novaculite and wear material, collected with JEOL 8900 microprobe at McGill University (beam current = 20 kV). Images **a** and **c–f** are of experiment 97. Image **b** is of experiment 101. **a** Surface of unsheared novaculate (starting material) with empty (100 μm) pores. **b** Cross-sectional view of wear material layer. White arrows indicate thin slip zones composed of compressed powder. **c** View normal to slip surface (ss) that was opened by separating novaculite cores. Clumps of ellipsoidal powder grains decorate surface and adhere to striated slip surface. Uneven distribution of wear material leaves exposed patches of slip surface with remaining 10 μm pores. **d** Nanopowder (np) compressed into striated plates (sp) parallel to slip direction. **e** Broken fragments of striated plates, interlayered with nanopowder, displaying development and cataclasis of multiple slip surfaces during single shear experiment. **f** Closeup on wear powder showing quartz chips, 10 μm clumps, and 1 μm particles. Bright clusters show debris from clumps which exploded during observation

experiments and during sample unloading, so our observation is only qualitative. However, comparing Experiments 111 (1 cm/s) and 112 (100 μm/s) shows that more of the surface is covered with wear material and the layer is thicker in the faster experiment (Fig. 6). On sample 111, which displayed extreme weakening (Fig. 1), in situ Raman transects across the sliding surface shows quartz peaks across the center ~30% of the diameter, while amorphous wear material covers the outer ~70%,

and the wear material is collected into dense smooth packs or plates which are concentrically smeared in the slip direction (Figs. 3, 6). On sample 112, which did not display substantial weakening (Fig. 1a), quartz peaks appear in the Raman transect in the center ~50% of the diameter, and in the outer rim of wear material, the powder is more evenly distributed. This observation of increased wear and amorphization at higher slip speeds is similar to previous experiments on clay-quartz mixtures[19].

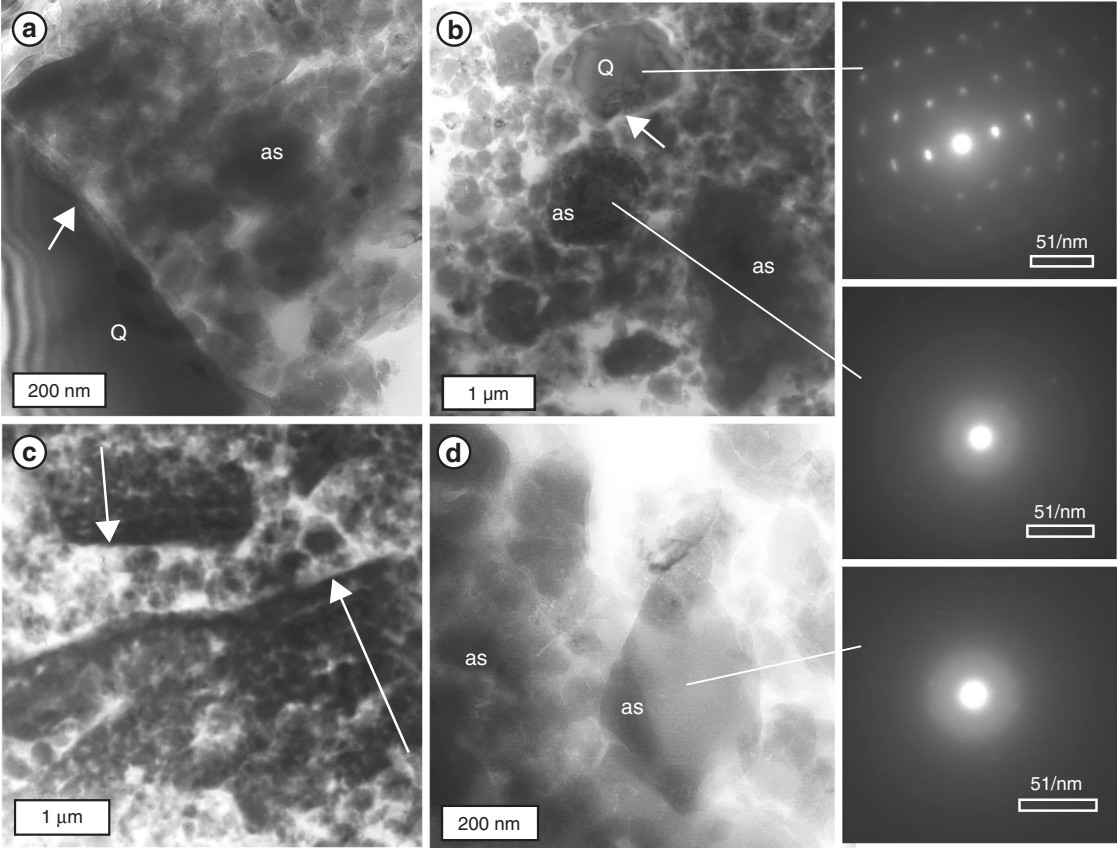

**Fig. 4** Transmission Electron Microscope images of wear material from experiment 101 (brightfield). **a** Quartz grain (Q) with rough fractured surface (white arrow). Clumps of amorphous silica particles (as) surround quartz grain. **b** Chips of crystalline quartz (top selected area electron diffraction pattern) and rounded clumps of amorphous particles (middle selected area electron diffraction pattern). At the scale of this image, nearly all amorphous silica grains are composites of smaller particles and fragments. **c** Broken fragments of striated plates composed of amorphous nanoparticles. Sharp slip surface composed of gradationally increasing packing density of particles toward slip surface (white arrows). Fragments are floating in loosely packed amorphous nanopowder. **d** Wear powder of rounded to elongate particles with grain size 10−100 nm that show no crystal structure (bottom selected area electron diffraction pattern)

In order to look for the presence of silanol (–Si–O–H), Fourier Transform Infrared (FT-IR) spectroscopy was collected using a PerkinElmer Spectrum TWO FT-IR instrument with a single-bounce diamond ATR crystal. The wear material displays FT-IR absorption band patterns distinct from both powdered novaculite (which matches $\alpha$-quartz[20]) and commercial amorphous nano-silica (Fig. 5b; Supplementary Note 1). The wear material spectra display features absent in the powdered novaculite, indicative of adsorbed $H_2O$ and silanol bending (Fig. 5b, Table 2), consistent with Hayashi et al.[9], and the absorbances for stressed Si–O–Si bonds are more pronounced than in the powdered novaculite. We observe water and silanol in the FT-IR spectra, but no evidence of silanol or hydroxyl was detected in the Raman spectra. As the detection limits of the FT-IR are ~10,000× lower than the Raman for hydroxyl and silanol, this result is consistent with a nanoparticle structure characterized by a thin hydrated silica layer rim surrounding an anhydrous amorphous silica core.

## Discussion

We have shown that the lubricating wear material produced in moderate to fast slip rate experiments on novaculite consists of spherical nanoparticles of amorphous silica with hydrated surfaces. Quartz can be amorphosed by high isotropic pressures (~15–25 GPa), comminution in a mortar and pestle at room conditions, or shock pressurization[21,22]. At similar normal load to our experiments (5 MPa), prepared quartz surfaces showed fractal distribution of contact size with an average contact stress on the order of 5 GPa[23], similar to the estimated contact stresses which generated amorphous wear material in previous experiments[10]. It follows that the smaller contacts would experience stresses above the isotropic amorphization stress (~15–25 GPa) even under static loading conditions. Experiments on nanocrystalline silicon showed that pressure induced amorphization (at 19 GPa) produced meta-stable supercooled glass which persisted after depressurization[24], potentially analogous to production of our amorphous particles. It is not clear whether comminution causes amorphization due to creating highly variable stress conditions resulting in locally reaching the isotropic amorphization stress, or whether an additional mechanical or thermal effect lowers the amorphization threshold[22,25]. In the case of comminution, high sphericity of wear particles appears to be a result of strain minimization in the material[25]. Amorphous silicate has been produced in other experiments at lower slip rates[26] and at lower slip rates with higher temperatures[27], without significant weakening, so it may be necessary for sufficient slip to form a continuous amorphous layer for weakening to occur[5].

Comparison of the wear material from experiments at different slip rates showed that the nanopowder was indistinguishable in

**a**

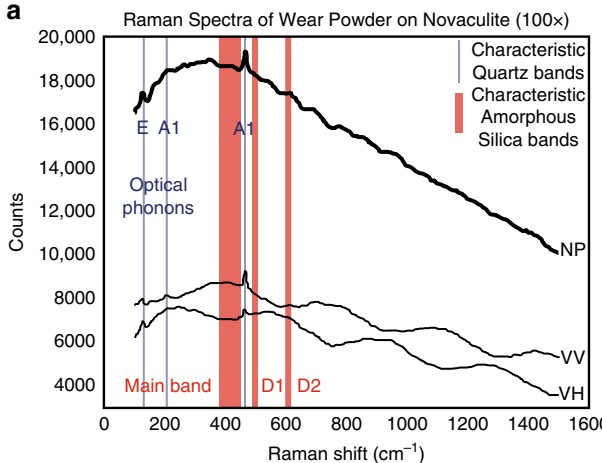

**b**

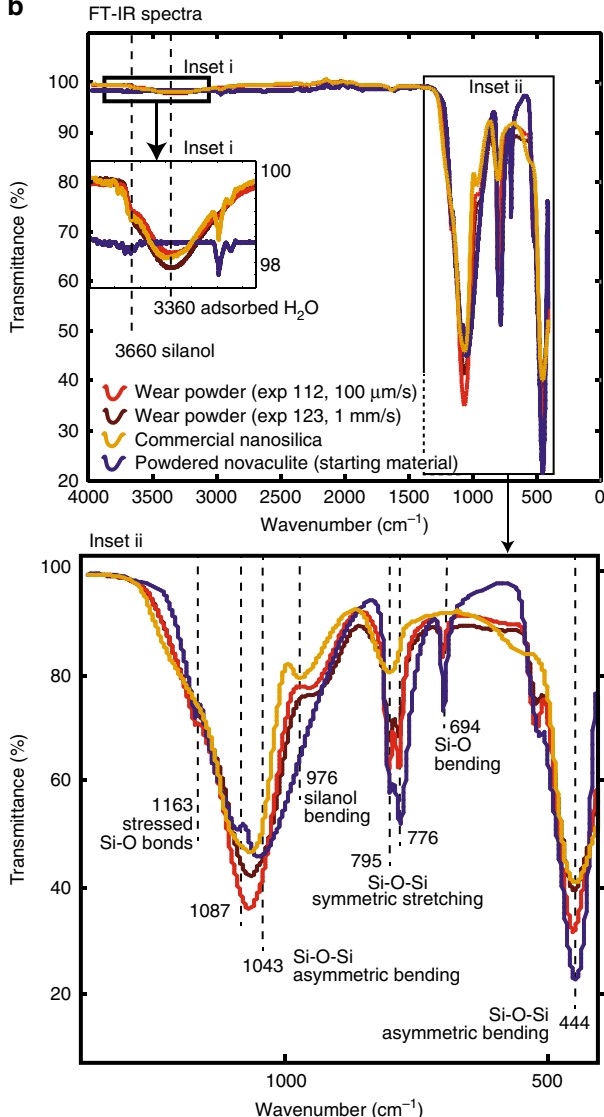

**Fig. 5** Raman and FT-IR spectra. **a** Raman spectra of the sliding surface showing non-polarized (NP) and decomposed polarized spectra (VH, VV; where first letter (V) refers to the vertical polarization direction of the laser, and second letter (H = horizontal, V = vertical) to the orientation of the analyzer). Red bars are normal vibration modes for commercial amorphous silica[31]. **b** FT-IR spectra of powdered novaculite (starting material), frictional wear powder from two shearing experiments and commercial nano silica. Small absorption peaks at 694 and 1163 cm$^{-1}$ indicate Si-O bond structures are present in the wear material, although not in long-range order (Inset **b**-ii)

slip rate (>1 cm/s) experiments (Fig. 1) must be explained by velocity-dependent properties, not just the presence of the wear material. The post-shearing strengthening[5,6,9] is dependent on the evolution of material properties during the static hold periods, potentially by time-dependent bonding[5,6]. We explain the rheo-logical observations with two deformation mechanisms that cause weakening and re-strengthening on two different time-scales: particulate flow assisted by hydrodynamic lubrication; and intraparticle plasticity (Fig. 7). The action of these two weakening mechanisms together is akin to superplasticity some-times described in crystalline materials which involves intra-particle deformation, as well as grain-boundary sliding between particles (c.f.[28]).

Although the amorphous wear material lacks long-range order, it is still dominated by silica tetrahedra, similar to crystalline species of silica. Si-O bonds which are dangling, stretched, bent (as shown in Fig. 5 and Table 2), or vacancies within and between silica tetrahedra all act as defects within the amorphous silica. Similar to crystalline materials, these defects are concentrated on particle surfaces[15]). Variations in the Si-O bond angle in silica glasses (including stress-induced varia-tions) cause changes in bulk moduli, solubility in acids, thermal expansion, viscosity, and rates of water diffusion in the glass[29]. Any defect which concentrates stress within the amorphous material plays a role during deformation which is analogous to (and can be modeled as) a dislocation in a crystalline structure, although these 'shear transformation zones' are not geometrically similar to dislocations[30]. The exceptional con-centrations of defects and vacancies in amorphous silica nanoparticles encourage hydration and plasticity[15,20,31], espe-cially at the small scales and high pressures as expected at asperity contacts (Fig. 7a;[32,33]). Yao et al.[33] have shown that heating, not just the presence of nanoparticles, is important for slip weakening. Hydration effects reduce the glass transition temperature and further promote plastic flow[34,35]. The exis-tence of the smeared nanoparticles forming the striated plates implies that the mild average frictional heating (Fig. 1b) experienced in the shearing experiments at 10 cm/s was suffi-cient to bring local patches above the glass transition tem-perature, allowing superplastic flow and the cohesion of wear powder to form the striated plates. The shear weakening effect would be reversed upon cooling, within seconds after the ces-sation of shearing.

Weakening in crystalline nanomaterials has been attributed to the activity of grain boundary sliding[36,37], but amorphous nanoparticles do not contain crystalline grain boundaries, and could even be work-hardening due to the rapid development of dislocation tangles[38]. Therefore, we invoke a superplastic flow mechanism in our amorphous nanopowders, but the details of the contributing mechanisms may be different than in crystalline wear powders. In amorphous silica, the mechanism of bulk flow is by silicon-oxygen bond switching and void migration, which can

grain size, structure, composition and water content. The lack of discernible difference between the wear material formed in dif-ferent experiments requires that the differences in shear resis-tance between the moderate slip rate (100 μm/s–1 cm/s) and fast

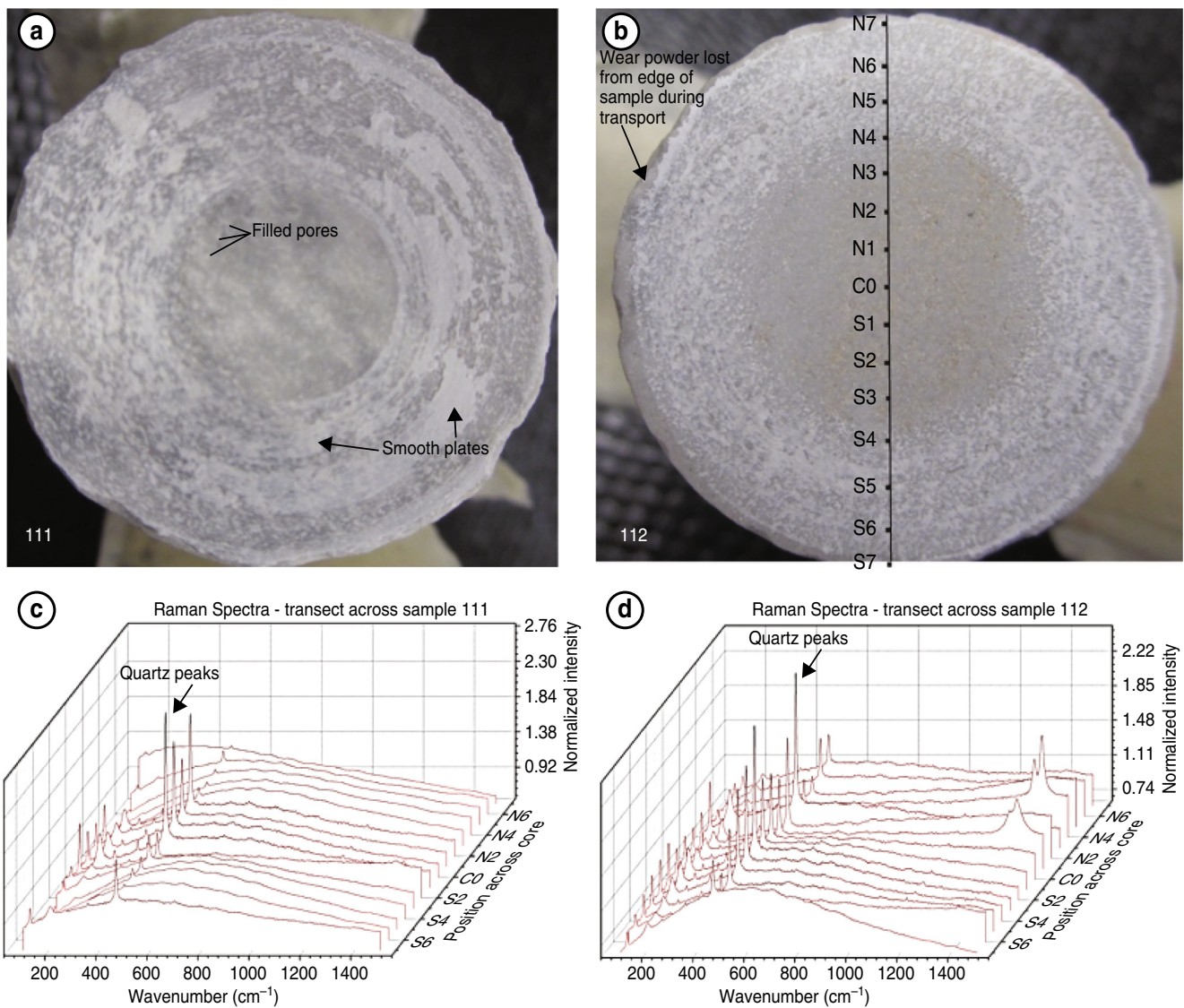

**Fig. 6** Photos of shear surfaces and corresponding Raman spectra transects. **a** Sample 111 (1 cm/s), and **b** Sample 112 (100 μm/s), showing discontinuous concentric rings of powder on sample surface. Raman spectra from transect scans are shown for each sample (**c** corresponds to sample 111, **d** corresponds to sample 112). The appearance and disappearance of the sharp quartz peaks along the transect can be correlated to the amount of visible wear material in the photos. See text for additional discussion

**Table 2 Peak positions in the FT-IR spectra and their corresponding vibrational modes**

| Vibrational mode | Position of peak (cm⁻¹) | | | | |
|---|---|---|---|---|---|
| | **Nanosilica** | **Commercial nanosilica (this study)** | **Wear material (this study)** | **Powdered novaculite (this study)** | **α-quartz** |
| Isolated (free) silanol[66] | 3747 | 3745 | | | |
| H-bonded silanol[66] | 3660 | 3650 | | | |
| Stretching of $H_2O$[66] | 3450 | 3348 | 3424 | | |
| Asymmetric stretching of Si-O-Si[20] | 1089 | 1066 | 1052 | 1087, 1043 | 1175, 1100 |
| Bending of Si-OH[67] | 972 | 1066 | | | |
| Symmetric stretching of Si-O-Si[20,68] | 812 | 798 | 797, 780 (double peak) | 795, 776 (double peak) | 797, 778 (double peak) |
| Symmetric bending of Si-O-Si[20] | | | 695 | 694 | 695 |
| Asymmetric bending of O-Si-O[20] | 474 | 539 (shoulder), 452 | 515 (shoulder), 447 | 510 (shoulder), 446 | 516 (shoulder), 470 |

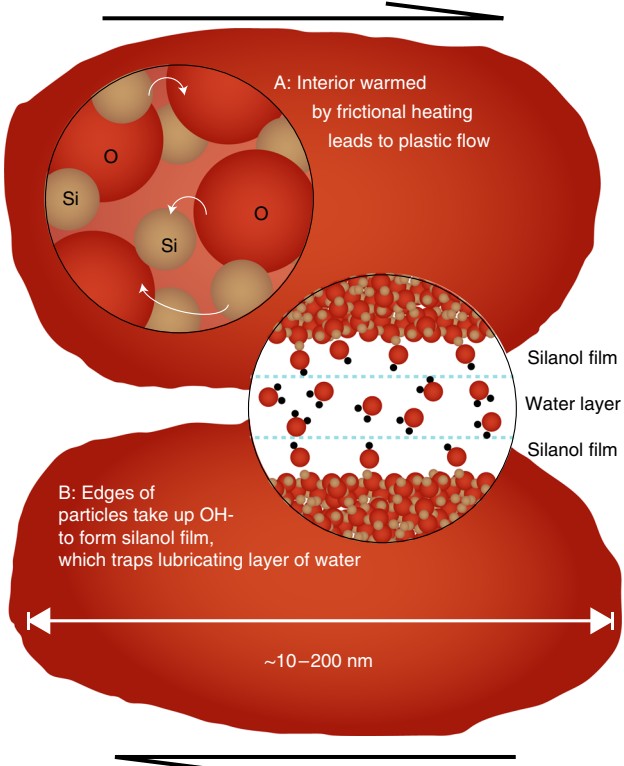

**Fig. 7** Schematic cartoon showing parallel operation of two slip weakening mechanisms. See text for discussion

change the number of atoms in oligomeric rings[39], and may explain the lack of any preferred coordination number in our amorphous silica wear material (Fig. 5a).

Simultaneous with intraparticle plastic deformation, hydrated particle surfaces cause a second mechanism of slip weakening (Fig. 7b;[40,41]). Hydration is particularly effective in stressed Si–O bonds on fresh surfaces[42], forming silanol films that trap water layers of unknown thickness between the particles[42,43], enabling hydrodynamic lubrication. Hydrated silica surfaces exhibit hydrophilic and hydrophobic regions, depending on silanol density[44], which create areas of interparticle repulsion and water adsorption. Hydrated surfaces on amorphous silica show a significant lubrication effect, which increases in magnitude with shearing velocity, particularly at high-stress and frictional contacts, where the increased pressure accentuates surface hydration[40,45].

Post-slip healing is caused by the reversal of the same mechanisms which cause the weakening. The hydrated surfaces are particularly favorable to inter-particle bonding when the relative motion ceases. In static contact with one another, the silanol groups rearrange to construct a silica tetrahedron (silox-ane), binding the particle surfaces together, and releasing a molecule of water[46]. The timescale of this bonding depends on contact aging, so the strength of the bond is time-dependent. This effect may explain instantaneous weakening upon re-shearing, as well as re-strengthening on ∼100 s timescales observed by Goldsby and Tullis[5]. We found evidence for this surface bonding in the agglomeration of particle clusters (Fig. 3c) that explode under the electron beam, showing that vaporization of the surface-bound water will disaggregate them. These are equivalent to Togo and Shimamoto's[47] finding of sintering between quartz gouge grains in high velocity experiments and the formation of shiny slip surfaces in experiments similar to ours, but they did not

test for the presence of amorphous silica which we observe forming these features.

Our experiments were modeled after those which first reported silica gel[5,6], but we have documented that the wear material produced in our experiments, which display the same weakening behavior, is not gel. Both the previous experiments[5,6] and our experiments produced nanopowders[48] and display similar weakening behaviors, but our experiments did not produce gel, perhaps due to different conditions. We therefore prefer the interpretation that the wear powder is likely responsible for the weakening behavior in both sets of experiments. Our results show that the behavior of hydrous amorphous silica nanoparticles at seismic slip rates can explain both the slip weakening and time-dependent recovery in laboratory experiments. The similarity of the wear material formed at all tested slip rates shows that the mere presence of amorphous silica nanoparticles is not enough to cause the observed drastic weakening. The whole range of tested velocities (100 μm/s–10 cm/s) are unique to earthquakes[49], but the lubrication only occurred at slip velocities >1 cm/s. Due to the strong velocity-weakening effect, the formation of this material likely promotes runaway slip. These fast slip rates are necessary to refresh interparticle contacts (preventing silica bonding) and to warm or pressurize[24] the interior of the amorphous particles above the glass transition. The wear material productivity was higher in the experiments which display this weakening, sug-gesting that a continuous layer of ≥10 μm thickness is required to activate silica weakening, c.f.[5,9,16]. Therefore, the formation of amorphous silica nanopowder might facilitate earthquake slip through mild weakening at small offsets or low seismic slip rates, and the extreme weakening (friction coefficient < 0.1) at peak slip rate could potentially result in complete stress drop on the effected portion of earthquake rupture surfaces. As real earth-quake faults are rough[50], the work done in frictional heating and amorphization is not uniform across natural slip surfaces. Compared to these experiments, a natural fault might require more or faster slip in order to build up a wear material layer of sufficient thickness to cause catastrophic weakening e.g.[51]. Or, a continuous layer may never develop, resulting in some residual strength on the fault, and preventing complete stress drop during earthquakes, e.g.,[52]. Although discrete sliding surfaces developed in patches during the experiments, they were broken and refre-shed during sliding, and no through-going principal slip surface or zone was found in our samples, including those that showed extreme weakening (Fig. 3b). One possible explanation is that the polished fragments lie at a slight angle to the slip plane, as they appear to be shingled with the forward edge (upper right in Fig. 3e and f) appearing to onlap the leeward edge of the neighboring fragment. It is unclear whether they formed in this orientation or were inclined due to drag of the opposite fault wall, but this pattern would prevent these slip surfaces from smoothly linking up to form a continuous shear surface across the whole sample.

Amorphous silica will crystallize to quartz within a few years at typical crustal temperatures[53], with crystallization rates enhanced by the presence of hydroxyl or other impurities and by differential stress[54]. Thin, translucent, nano-crystalline to micro-crystalline quartz layers have been reported on natural fault surfaces, con-taining patches of amorphous silica, and relict silica nanoparticles entombed in strain-free quartz crystals[55–57]. These continuous coatings of amorphous silica or cryptocrystalline quartz on nat-ural fault surfaces may constitute the rock record of silica lubri-cation in past earthquakes.

## Methods
**Friction experiments.** Experiments were performed using the ROtary-Shear Apparatus ROSA (built by MARUI & CO., LTD (model MIS-233-1-77) as designed

by T. Shimamoto). A detailed report of the design and capabilities of ROSA is provided by Rempe et al.[58]. The rotation of the axial column was controlled by the 11 kW servomotor; the normal load was applied (on the stationary column) via a pneumatic system. Mechanical data were collected at a rate of up to 1 kHz. For the experiments reported in this paper, cylindrical novaculite samples 25 mm in diameter and 3.5 cm in length were prepared and roughened with 150 grit SiC paper prior to the experiment to ensure equal roughness of the sliding surfaces.

**Flash heating temperature calculations**. We calculated the temperature increase by flash heating in water-poor asperities: with no free water in pores around the asperities to cool them down during the experiments (following[14], supplementary material).

In general, the temperature increase by frictional heating is described as:

$$\Delta T = \frac{\tau V \sqrt{t}}{\rho c_{\mathrm{p}} \sqrt{\alpha_{\mathrm{th}} \pi}} \quad (1)$$

Which at the asperity scale becomes:

$$\Delta T_{\mathrm{flash}} = \frac{\mu_{\mathrm{p}} P_{\mathrm{m}} V \sqrt{t_{\mathrm{c}}}}{\rho c_{\mathrm{p}} \sqrt{\alpha_{\mathrm{th}} \pi}} \quad (2)$$

The properties of the wear material are poorly constrained, so we chose values conservatively to place a maximum bound on the flash heating temperature at asperities. The temperature was calculated using the properties of water-poor type I silica glass (density $\rho$ (=2200 kgm$^{-3}$), heat capacity $c_{\mathrm{p}}$ (=1026 Jkg$^{-1}$K$^{-1}$)), and thermal diffusivity $\alpha_{\mathrm{th}}$ (= (6.07e−7 m$^2$s$^{-1}$)[59], $P_{\mathrm{m}}$ is indentation strength of quartz (=2.7e9 Pa,[60]) $\mu_{\mathrm{p}}$ as our peak friction coefficient (=0.7), $V$ is our slip rate ( = 0.0001–0.1 ms$^{-1}$), and $t_{\mathrm{c}}$ is contact time. Contact time is related to the radius of circular asperities $a$ and slip rate $V$ as:

$$t_{\mathrm{c}} = \frac{a}{V} \quad (3)$$

The radius of the asperities is related to the normal force $F$ ( = 1227 N) and $P_{\mathrm{m}}$.

$$a = \sqrt{\frac{F}{P_{\mathrm{m}} M}} \quad (4)$$

Figure 2 shows the temperature increase by flash heating versus slip rate in the experiments. We used the peak friction coefficient during the experiments ($\mu = 0.7$) to calculate the heating, so actual asperity temperatures were lower once weakening began.

**Post-experiment sample preparation and observation**. Following sliding experiments at Padova, samples were recovered intact from the shearing apparatus, secured and sealed with tape, and sent to McGill University. Some samples were cut longitudinally (along the core axis) in order to image the sliding surface and wear material in cross section (e.g., Fig. 3a, Fig. 4). Some samples were gently opened (top novaculate core removed) to expose the sliding surface (e.g., Fig. 3b–d).

The TEM foils were prepared by J.C. White at the University of New Brunswick. The first step is preparation of a thin section which contained the entire wear layer (cross section of the intact slip surface, e.g., Fig. 3a) using CrystalBond adhesive (manufactured by SPI Supplies) to adhere the sample to the glass slide. The thin section was polished and observations were made using standard petrographic microscope and scanning electron microscope (SEM). Then, 3 mm copper grids were glued to the polished surface of the thin section. The thin section was immersed in acetone to gently dissolve the CrystalBond, leaving a thin (30 μm) sheet of rock with copper foils on it. The copper foils help to keep the rock sheet intact especially through the slip surface area where it would be likely to break without reinforcement. Using tweezers, the thin sheet of rock is gently broken away from each copper grid, leaving only the pieces adhered to the grids. This mount is then thinned very slowly using an oblique beam at a low angle from an ion mill until the rock sheet is just perforated, and the margins of the hole are of the appropriate thickness for electron beam transparency. As the thin section was fully characterized and photographed prior to being broken up, we are able to determine exactly what is preserved and observed in the TEM foil and place that in context of the complete slip surface.

Backscattered electron images (Fig. 2, except for 2b) were collected by C. Rowe on the JEOL 8900 microprobe in the Earth and Planetary Sciences Department at McGill University. Figure 2b was collected by J.C. White using the JEOL 6400 Scanning Electron Microscope at the University of New Brunswick Microscopy and Microanalysis Facility. Transmission electron microscopy (TEM) images and diffraction patterns were collected by J.C. White using the JEOL 2011 Scanning Transmission Electron Microscope (STEM) at the Microscopy and Microanalysis Facility, University of New Brunswick. Adobe Photoshop was used to adjust

brightness and contrast of the backscattered electron and transmission electron images.

**Raman spectroscopy and polarizability experiments**. Raman spectra were collected by K. Lamothe using the laser Raman microprobe spectrometer with 633 nm He–Ne excitation laser in the Materials Chemistry lab at McGill University.

Raman spectroscopy uses a laser to interrogate a sample in order to measure the very small fraction of laser photos that are scattered inelastically, meaning the energy of the photos is shifted from interaction with the molecules in the sample. Measurement of the shift in wavelength of these photons, called the Raman shift, yields a spectrum that gives information about the vibrational modes of the molecules, which are specific to the chemical bonds and can therefore be used to gain insight into the molecular structure of the material. Raman bands are often assigned a Mulliken symbol to describe the symmetry of the modes that produced them[61]. The spectrum characteristic of quartz contains A1 and E bands, with A1 signifying the total symmetry and E signifying a doubly-degenerate, two-dimensional irreducible representation. A concise description of normal vibrational modes is given by Tuschel[62]. The bands associated with amorphous silica include D bands, a name reflective of disordered structure rather than a Muliken symbol which is associated with ring modes in silica[63,64].

Polarisibility experiments have shown that the wear material is highly polarization dependent (Fig. 5a). The R-band shows a strong polarization dependence for parallel ($\bar{X}(ZZ)X$; VV) versus the perpendicular ($\bar{X}(ZY)X$; VH) polarizations (Supplementary Fig. 1)[65]. The $Z$-axis refers to the axis along which the incident laser beam propagates to the sample and the scattered photons return (incident electric field). The $X$-axis refers to the 'horizontal' axis for the sample, generally parallel to the front of the stage on which the sample is mounted, and $Y$ is in the plane of the stage, perpendicular to the $X$-axis. Notation for polarizability experiments is given in the form of $a(bc)d$ where $a$ is the axis along which the incident laser propagates, $b$ is the axis of polarization of the incident laser by the polarizer, $c$ is the axis of polarization of the scattered photons by the analyzer, and $d$ is the axis of propagation of the scattered photons (Supplementary Fig. 1).

## Data availability
The Matlab codes used for temperature calculations including Fig. 1b are archived on GitHub at https://github.com/aretu/thermalmatlabcodes. All data is available from the authors on request.

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

## Acknowledgements

This work was supported by NSERC Discovery Grants to C.D.R. and J.C.W., the Wares Faculty Scholarship to C.D.R., and by the ERC CoG 614705 NOFEAR Grant to G.D.T. and M.R.

## Author contributions

C.D.R. conceived the study, performed microprobe and XRD analyses, and prepared the manuscript and figures. K.L. and M.A. performed Raman and FTIR experiments, interpreted the data, and contributed text and figures to the manuscript. M.R. and T.M. performed the shearing experiments with supervision and support from G.D.T., who contributed to interpretation of all results and performed temperature calculations. J.C.W. made SEM and TEM observations of the wear material and contributed figures. S.A. contributed three temperature models, supplementary figures, and text.

## Additional information

**Competing interests:** The authors declare no competing interests.

