## [Peer Review File · Nature Communications]

Reviewers' comments:

Reviewer #1 (Remarks to the Author):

Review of "Earthquake lubrication and healing explained by amorphous silica" by Rowe et al.

The paper reports on a set of moderate to high velocity friction experiments performed on quartz rich rocks (novaculite) over a range of slip velocities. Detailed microstructural observations using a wide range of methods (SEM, TEM, Raman, FTIR) clearly document the formation of amorphous silica particles. The particles have a unique fingerprint in the raman spectra which has not been observed in other more common quartz-rich materials. This alone is an important contribution because it allows experimentalist and field geologist to search for such fingerprints in fault rocks (provided they are fresh enough) and gives novel insight into the structure of wear products. The microstructural observations are further used to discuss the processes responsible for the observed mechanical behavior focusing on the post-slip healing part of the experiments which is rarely treated in detail. I enjoyed reading the paper and I recommend publication after some revisions as described below.

The authors infer that the amorphous silica particles have high concentrations of defects and vacancies which promote intraparticle plasticity – this might need some clarification as defects and vacancies are defined in crystalline materials, i.e. the absence of an atom in an otherwise periodic structure is a vacancy and a defect (dislocation I assume) is again an extra plane of atoms in a periodic structure. The amorphous particles have no such periodic structure though. What are the agents of deformation in the amorphous particles?

Further, the authors infer grain boundary sliding and superplastic flow as a deformation mechanism operative on the slip surfaces - both these terms are defined for crystalline materials subject to volume-conserving high temperature creep. I find the use of this terminology for deformation of porous(?) aggregates of amorphous particles slightly confusing. In my mind, a grain boundary is a region where two crystalline lattices make contact at a higher than some threshold angle (say 10°). What is a grain boundary in an amorphous structure? Maybe cataclastic flow of amorphous particles might be a better description of the deformation mechanism to avoid confusion?

As described on lines 44 – 51, it is very interesting to note that the thin principal slip surfaces are continuously being produced and destroyed – while the observation is clearly documented, the implications are that the principal slip surfaces accommodate displacement only intermittently. Do the slip surfaces strengthen after accommodating some slip which causes the deformation to localize elsewhere? Or, in other words, if the material on the principal slip surface is mechanically weak, why doesn't the deformation stay localized in that layer?

What is the role of temperature in the healing process (if any)? Heating during slip seems to be critical to reach low friction, does cooling account for the rapid re-strengthening immediately after the displacement is stopped?

As the authors note in the introduction, the experiments appear to be very similar in terms of slip velocity, material used and mechanical data to the experiments performed by Di Toro et al. (2004) who concluded that the observed mechanical behavior is caused by silica gel, however no silica gel was observed in the experiments reported here – what is the key experimental difference that leads to the formation (or absence) of silica gel?

Line 56-57: "associated" is used twice in the sentence.

Reference:

Di Toro, G., Goldsby, D., Tullis, T. Friction falls towards zero in quartz rock as slip velocity approaches seismic rates (2004) *Nature*, 427-6973, pp 436-439

Reviewer #2 (Remarks to the Author):

This paper describes the results of friction experiments on bare rock surfaces and they found dramatic frictional weakening at sub-seismic slip rates (~ 10 cm/s). The mechanical data is sound and convincing. The key observation is that they found widespread amorphous nanosilicate in their post-experimental samples. To confirm this, this manuscript contains a workman-like characterization of the Raman spectra and FT-IR analyses of their post-deformational samples. I have not done these analyses myself and tried to learn from the literature during the reviewing time (mostly from the reference list). Their analyses and the results shown in the paper do make sense to me. In other words, all the data are excellent, new, and well-conceived. Probably, further check from a specialist is needed.

However, I find that the discussion contains a lot of speculations.

1) Grain boundary sliding (plasticity) with hydrodynamic lubrication was inferred to be dominant weakening mechanism. However, without quantitative analysis, it is difficult to convince the readers that that plasticity of amorphous nanosilicate can explain the weakening. The authors have referred to the work by Yao et al. [2016]. In their more recent paper in *GRL*, they examined the possibility that if superplasticity can explain the weakening observed in nano-power of MgO. Although diagnostic polygonal texture with triple junctions has been observed in their samples, the calculation showed that the calculated flow stress is many orders of magnitude too high to explain the observed friction. In contrast, similar calculation has been done by De Paola et al. [2015], showing that plasticity can explain the weakening observed. Bear in mind that in either Yao et al. [2016], De Paola et al. [2015] or Green et al. [30]'s experiments, the samples were subjected to higher temperatures. If amorphous nanosilica can cause weakening under relatively low temperatures, more evidence, at least quantitative analysis, is needed.

2) Without some critical experiments, hydrodynamic lubrication is even more speculative. The evidence provided by the FT-IR analyses is quite convincing for the presence of water layer, but it is difficult to draw the conclusion that it plays a crucial role in the lubrication. A simple experiment can be done under dry conditions (e.g. under N₂ environment).

3) In the present paper, the authors argued that "the faster slip rates (> 1 cm/s) are necessary to ... warm the interior of the amorphous particles above the glass transition". How about the slip surface temperatures at different slip rates? If the temperatures is quite different, then it involves a sort of "thermal weakening". If not, the weakening should occur at slip rate lower than 1 cm/s (related to Figure 6).

4) The effect of fault healing is not straightforward in the present paper, at least from the mechanical data. The microstructure does show some grain sintering (inter-particle bonds), but similar structure/effect has been emphasized many times in the literature [e.g. Togo and Shimamoto, 2012].

In terms of novelty, the observation of amorphous nanosilicate on bare rock surfaces and the related interpretations are new, although as pinpointed by the authors (in lines 127-128), "small amounts of amorphous silicate have been produced in other experiments at lower slip rates and higher temperatures, without weakening" [27, 28]. It is important to note that there are also some previous frictional experiments producing amorphous silica [Hayashi and Tsuyumi, 2010; Nakamura et al., 2012], both published in *GRL* (though it is difficult to judge just from the microstructure if there are the same or not).

Finally, the paper is of great interests to both experimental and geophysical communities. The observation of the presence of amorphous nano-silica and dynamic weakening are both convincing, but they are not necessarily related. It would be a potentially important contribution if the authors can somehow show that flow stress of amorphous nano-silica is weaker or can be more easily activated by mild temperature rise than crystallized nano-silica. It would give more impact if the paper can compare silica gel and amorphous nanosilicate.

Specific comments

Figure 1: why negative displacement at the very early state?

Line 107: If plasticity is related to the local stress at contact asperities, then plasticity is expected to be insensitive to the normal stress applied. Do you have experiments under variable normal stress?

Line 144: If amorphous nano-silica needs to be warmed in order to become weaker, experiments at lower slip rates and higher normal stress can also produce weakening. Is this the case?

Figure 6: I very much like this diagram. Do you mean “slightly interior warmed”? You really need to measure or calculate the temperature on the slip surfaces.

- 1) De Paola, N., R. E. Holdsworth, C. Viti, C. Collettini, and R. Bullock (2015), Can grain size sensitive flow lubricate faults during the initial stages of earthquake propagation?, *Earth Planet. Sci. Lett.*, 431, 48–58.
- 2) Yao, L., S. Ma, A. R. Niemeijer, T. Shimamoto, and J. D. Platt (2016), Is frictional heating needed to cause dramatic weakening of nanoparticle gouge during seismic slip? Insights from friction experiments with variable thermal evolutions, *Geophys. Res. Lett.*, 43, 6852–6860.
- 3) Nakamura, Y., J. Muto, H. Nagahama, I. Shimizu, T. Miura, and I. Arakawa (2012), Amorphization of quartz by friction: Implication to silica-gel lubrication of fault surfaces, *Geophys. Res. Lett.*, 39, L21303, doi:10.1029/2012GL053228.
- 4) Hayashi, N., and A. Tsutsumi (2010), Deformation textures and mechanical behavior of a hydrated amorphous silica formed along an experimentally produced fault in chert, *Geophys. Res. Lett.*, 37, L12305, doi:10.1029/2010GL042943.
- 5) Togo, T. and T. Shimamoto (2012), Energy partition for grain crushing in quartz gouge during subseismic to seismic fault motion: An experimental study, *Journal of Structural Geology*. doi:10.1016/j.jsg.2011.12.014.

Reviewer #3 (Remarks to the Author):

General Comments:

This is an interesting paper and should be published, but I believe it would be greatly improved by some revision. I have a considerable number of general and specific comments and hopefully considering them will improve the paper. Many of my comments are perhaps a bit self-serving and are intended to give more “credit” to relevant prior work (which I have been involved in over the years!) and to point out some interesting and in some cases puzzling contrasts between these interesting new results and prior ones.

These experiments themselves are very similar to ones done previously and referred to in reference numbers (4; 5), as well as described in my review paper that I refer to below as (#). However, the authors observe no silica gel in the samples from their experiments, which is in contrast to inferences as well as observations we have made on our experimental samples. Since the experiments were very similar and were done on essentially the same novaculite rock, this difference in whether or not silica gel was involved is puzzling.

Several possibilities exist, including that

- 1) gel was present in their experiments, but it was hard to see and they missed it
- 2) there is a difference in the water content of the samples, perhaps due to a difference in humidity of the experiments, and if these experiments were drier, gel might not have been produced
- 3) something about the combination of slip speed and normal stress made a difference

4) the lack of gel in their experiments could result from their weak samples being hotter than the previous experiments

The authors should consider the possibility of confronting this difference between their and previous results head on and try to explain it. As the paper now stands, the contrast is present, but it is not really addressed. This is not very satisfying for the literature as a whole. An explanation for the difference in observed gel is not obvious.

Brief thoughts on 1-4 above are:

1) This possibility is addressed in discussion of lines 50-51 below. Summarizing, it is really hard to see the gel flow features and high magnification images of exactly the right spots are needed to see them. Which experimental sample(s) has been examined for the presence of gel is not given in the discussion of Figure 2.

2) Perhaps the water content was somehow different, although both were done under atmospheric humidity, and in our gel-evident experiments extreme dryness was required to make weakening vanish. Their FTIR measurements show that at least some water is present in these experiments too. So it's hard to see how their samples could be so dry as to eliminate gel.

3) We see similar weakening at conditions with slip velocities of 3.2 mm/s and normal stresses from 28 to 112 MPa, to 0.1 m/s and normal stresses of 5 MPa, with slip displacements from ranging from 0.7 m to 63 m. So although our normal stresses are never lower than 5 MPa and those in this work are all 2.5 MPa, the experiments otherwise span the same conditions. We have not examined samples from all of our experiments that show weakening with SEM imaging like that shown in (#). However, the mechanical behavior of all the experiments are similar, so we infer that silica gel is responsible for all our observed weakening.

4) The temperatures presumably do not get higher in the experiments of this work than in those of (4; 5), so it is hard to see how this could be the explanation, although temperatures are not given for this present work. Higher temperature could dry the samples out enough to eliminate gel.

Specific comments by line number:

Sentence in middle of abstract: "Two slip rate-dependent mechanisms cause the weakening: thermally-enhanced plasticity, and hydrodynamic lubrication." After reading the paper I do not believe that they have proven this statement with regard to either mechanism. They suggest them as hypotheses to explain their data, but they by no means prove that these explanations are correct. The sentence should be changed to something like: "Two slip rate-dependent mechanisms are plausible explanations for the observed weakening: thermally-enhanced plasticity, and hydrodynamic lubrication." Note however that my comments referring to line 118 suggest that hydrodynamic lubrication is not a term that should not be used for what they envision.

Line 2: insert a new reference number (I'll call it #) after the "2" in "(2)", so "(2; #)". That, somewhat difficult to find but very relevant, reference is:

Tullis, T. E. (2015), Mechanisms for friction of rock at earthquake slip rates, Chapter 5, in *Treatise on Geophysics*, v.4, Earthquake Seismology, edited by H. Kanamori, pp. 131-152, Elsevier Ltd., Oxford.

It should be cited in general somewhere for high speed weakening mechanisms – this seems like a good place. I've sent a digital reprint to the first author, who can distribute it to the others.

Line 10: might change the end of the sentence " . . . for silica gel" to " . . . for silica gel, even though the conditions are similar to those for experiments in which silica gel is observed (4; 5; #, Fig. 11)." Here # is intended to be that citation number for the new reference above. Fig. 11 of that reference shows a SEM image of silica gel from the experiments of (4; 5).

Line 11: insert "to silica gel" between "explanation" and "for"

Line 25: Add to the end of the sentence that now ends in "other" so it ends "other at atmospheric humidity." No mention is made of the humidity, but I assume it to be uncontrolled in room air. We have found that the humidity makes a huge difference in both the weakening and the gel formation and so the humidity should be stated. If its dry enough, the weakening goes away, but that has to be extremely dry. So far this is only published in this AGU abstract:

Titone, B., K. Sayre, G. Di Toro, D. L. Goldsby, and T. E. Tullis (2001), The role of water in the extraordinary frictional weakening of quartz rocks during rapid sustained slip, *Eos Trans. AGU*, 82, T31B-0841.

Line 27: "Equivalent velocities" is referred to here. This needs to be defined or explained. The slip velocity goes from 0 in the center of the solid cylindrical samples to a maximum at the outer diameter. To what radius do the quoted velocities refer?

Line 37: the clusters referred to here are mentioned enough times that it would be helpful if they were pointed out in Figure 2 as well as in Figure 3, since Figure 2 is referred to here. At this point, a reader is not sure if you are referring to the fairly large plates visible in Figure 2 B&C and/or to smaller pieces visible there and in Figure 2D.

Figure 2C: Magnification is not adequate to show the 10 um pores referred to in the caption. Point to them with an arrow if they are important and/or increase the magnification. What is the point of referring to them?

Line 50-51: It is not clear to me that the lack of observation of "viscous fluids" on the slip surfaces means that they were not there. Note that the SEM observations of what were formerly viscous fluids in the experiments described in (4; 5: and #) were made at a much higher magnification than those illustrated here in Figure 2 (see #, Fig 11) and the gel was not easy to distinguish. The layer of silica gel in those experiments was only ~3 microns thick and when looking perpendicular to the smooth slip surfaces no viscous flow type features were visible. It was only at the margins of pre-existing pits in those smooth surfaces where the gel had been partially smeared into the pits that the flow features could be seen. It seems quite possible that unrecognized gel exists in the experiments being reported on here. Whether the weakening reported on here could be due to the existence of gel, simply to the presence of the clumps of amorphous nano-particles, or perhaps some combination, seems to be presently unclear.

Line 52-53: These observations of the wear material are good and useful, but note that it is very difficult in TEM to image the actual smooth slip surface shown in Figure 2B. We tried to do this in our 1990 TEM study of sheared gouges (27), but found that the thinning process of making the TEM foil removed the material right at the slip surface. Modern FIB techniques might get around this problem, but the discussion in this paper does not indicate that they tried to or were able to look directly at the material exactly adjacent to the main slip surface in TEM. The observations of apparently relic slip surfaces in Figure 3 C are quite interesting, but these surfaces may differ from ones that had very large amounts of slip on them.

Line 73 (and on line 135): I understand the idea of oligomeric complexes but why are they referred to as rings? I realize that space is short, but a tiny bit of background here would help. What makes them have that shape or what makes us think they do or should? Actually, though I still don't know why rings occur, the first paragraph of the Supplementary Materials discuss this a bit. I suggest just adding a ref in this part of the main text to the Supplementary Materials and then in that location there would be room to perhaps explain why rings occur. Clearly this question is tangential to this paper, but it would be helpful to get a bit more on this into the Earth Science literature!

Line 79: Is the wavenumber range of expected Si-OH or adsorbed water within the range shown in Figure 4A? It would help to give the waver number range expected, and either point out that it is, or is not, covered in Figure 4A.

Figure 4A. The title of this on the figure uses the word “Chert.” Change this to “Novaculite” to be consistent with rest of paper.

Figure 5: The writing on the axes is too small to read, even when blown up. I suggest trying labeling every other major tic mark and doubling the font size. This is a problem on all three axes. It is not clear to me that any of the parts of this Figure really add that much to the paper – consider omitting. The only point seems to be that the amount of different materials varies on the surface, which Figure 2B already shows, albeit at a finer scale. Nothing is made of the difference in the amount of material on the surface or the difference in the spectra between the 0.01 and the 0.0001 m/s sliding rate samples illustrated. If there are some points to be made, then retain the figures and say more about them.

Lines 88-94: If anything can be said about the amount of water or OH in the sample from the spectra, then something should be said, or if it is not possible, then say that. This is an important issue since whether this material should be considered a gel or what the viscosity of the gel might be at different strain rates presumably depends on water content, even though we probably don't know much about that. So what quantitatively can be said from the combination of the IR and the Raman data would be useful to say.

Lines 90-101: Would be helpful here to refer to references 4 and 5 and their description of the thixotropic behavior of their gel, which is exactly what you observe with your material. The role played by the competition between time dependent creation of bonds and the strain dependent disruption of them seems the same in either case.

I initially thought that perhaps the situation is not the same here, since in that earlier work weakening was observed at lower velocities (4) where the temperature was inferred to not get higher than 140 deg C. Here the weakening is only seen at slip velocities of 0.1 m/s where the frictional heating can raise the temperature higher. But actually it seems likely that the temperature is not higher in these experiments than in the experiments of (5) where the velocities are the same, the slip distances are similar, and the normal stress is actually twice as large as in the current work. As mentioned above in the discussion of point 4) at the end of my general discussion, it is unlikely that the temperature was higher in these experiments than in all of those earlier ones that show weakening (4; 5; #). On line 111 “mild frictional heating” is mentioned. In fact, it would be very helpful to give some estimate of the temperature likely to have been attained, based on some measurements or calculations. All that is said (line 112) is that it inferred to be above the glass transition temperature. Giving values of the estimated temperature as well as whatever is known about the glass transition temperature will make it possible for the reader (and reviewer!) to evaluate how likely this inferred crossing of the glass transition temperature is.

Line 118: The term “hydrodynamic lubrication” is used here as well as on lines 104 and in the abstract. I don't think this term is appropriately used here and I recommend removing mention of it in all three of these places. In engineering practice, in a 2001 paper by Brodsky and Kanamori, and in the review paper (ref #), this term is used to mean flow of bulk fluid that builds up normal-stress-supporting pressures by flowing in appropriately shaped confined spaces between sliding solids. Here you use it to mean shearing of mono- to multiple-layers of water. I strongly discourage the use of this term for what is envisioned here. I fear it will lead to confusion in the literature since it then would be used with multiple meanings.

In fact it seems to me that what you describe is not really a second mechanism, but rather an essential component of the superplastic flow that you describe in the previous paragraph. For superplastic flow to occur one needs not only the particles themselves to be able to deform so as to accommodate the changes in shape needed as they change neighbors, they also have to be able to undergo sliding on the boundaries of the particles. This would be enhanced by the presence of thin water films on the surfaces of the particles as you illustrate in Figure 6. Thus I suggest that you do some rewriting of these paragraphs to integrate the two thoughts into two components of one weakening mechanism. This is a standard way of breaking down the processes in structural

superplasticity, namely it occurs by grain-boundary sliding (one component) with either dislocation or diffusion accommodation (the other component).

If this doesn't quite capture your thoughts about how the two components contribute to the deformation, then discuss that explicitly. For example, perhaps describing it as plastically accommodated grain-boundary sliding puts too much emphasis on the grain boundary sliding component, then contrast it with that idea and suggest that it is better described as plastically deforming particles that are also able to easily slide past one another.

Line 127: In ref 27 it was not small amounts (i.e. was 40-50% - Table 1 for quartzite in ref 27). Perhaps just remove "Small amounts of" from the start of the sentence and change "have" to "has".

Line 128: Change to "rates and lower temperatures (27) and lower rates and higher temperatures (28)." I suggest this change since ref 27 is at room T, which the sentence implies not to be the case.

Line 128-129: I suggest that after the reference to (28), replace the rest of the sentence with "Weakening seems to require both slip to occur at intermediate to high velocity and slip to be sufficient to form a continuous amorphous layer (4)." The latter was true in the experiments of (27; 4; 5; this work) but without elevated slip rate there is no weakening. Elevated slip rate can be as slow as 3.2 mm/s in the work reported in (4). Also note that in ref (4) at the end of paragraph [17] we suggested that sufficient slip is needed to make a smooth surface of amorphous material and gel, hence my insertion of ref [4] as the end of this sentence.

Line 136-137: Not clear that rolling matters – if it does, why isn't the material weak at low slip velocities? I suggest omitting this sentence?

Comments on Supplement:

First two lines: R band is mentioned – what does that mean? This is referred to again near the end of this paragraph. Giving wave numbers for R band would help. It is not discussed in Figure 4A, caption, or text.

Also the discussion on lines 2&3 with the notation in parentheses isn't clear to most readers, although I suppose to a Raman expert it would be. Adding an explanation of the nomenclature would help.

FTIR section, second paragraph, first sentence: Complementary to what? The previous paragraph was already talking about IR results so exactly what this sentence is trying to say is unclear. Rewrite.

Last paragraph of this section: Both "694" and "695" are used to refer to the same peak. "695" was used in first paragraph. Decide which to use and use only one.

Reviewer #1 (Remarks to the Author):

Review of “Earthquake lubrication and healing explained by amorphous silica” by Rowe et al.

The paper reports on a set of moderate to high velocity friction experiments performed on quartz rich rocks (novaculite) over a range of slip velocities. Detailed microstructural observations using a wide range of methods (SEM, TEM, Raman, FTIR) clearly document the formation of amorphous silica particles. The particles have a unique fingerprint in the raman spectra which has not been observed in other more common quartz-rich materials. This alone is an important contribution because it allows experimentalist and field geologist to search for such fingerprints in fault rocks (provided they are fresh enough) and gives novel insight into the structure of wear products. The microstructural observations are further used to discuss the processes responsible for the observed mechanical behavior focusing on the post-slip healing part of the experiments which is rarely treated in detail. I enjoyed reading the paper and I recommend publication after some revisions as described below.

The authors infer that the amorphous silica particles have high concentrations of defects and vacancies which promote intraparticle plasticity – this might need some clarification as defects and vacancies are defined in crystalline materials, i.e. the absence of an atom in an otherwise periodic structure is a vacancy and a defect (dislocation I assume) is again an extra plane of atoms in a periodic structure. The amorphous particles have no such periodic structure though. What are the agents of deformation in the amorphous particles?

Amorphous materials lack long-range order, but they are not without molecular structure and these molecules (e.g. silica tetrahedral, bonded to one another in various configurations) can have vacancies and defects similar to those in the same sites within crystalline materials. The resistance to defect migration is lower than in a fully crystalline material, but there are still bonds that must break and re-form when the defects migrate. So although the constitutive relationships are not well defined, the qualitative behavior of solid-state deformation processes is often approached using crystal plastic deformation as an end-member analog model. A paragraph explaining this process and the analogy to deformation of crystalline materials has been added to the discussion (lines 151-159).

Further, the authors infer grain boundary sliding and superplastic flow as a deformation mechanism operative on the slip surfaces - both these terms are defined for crystalline materials subject to volume-conserving high temperature creep. I find the use of this terminology for deformation of porous(?) aggregates of amorphous particles slightly confusing. In my mind, a grain boundary is a region where two crystalline lattices make contact at a higher than some threshold angle (say 10°). What is a grain boundary in an amorphous structure? Maybe cataclastic flow of amorphous particles might be a better

description of the deformation mechanism to avoid confusion?

We have changed 'grain boundary sliding' to 'particulate flow' to avoid confusion. We prefer 'particulate flow' over 'cataclastic flow' because cataclastic flow typically involves grain breakage, and we don't observe any evidence of fracture.

As described on lines 44 – 51, it is very interesting to note that the thin principal slip surfaces are continuously being produced and destroyed – while the observation is clearly documented, the implications are that the principal slip surfaces accommodate displacement only intermittently. Do the slip surfaces strengthen after accommodating some slip which causes the deformation to localize elsewhere? Or, in other words, if the material on the principal slip surface is mechanically weak, why doesn't the deformation stay localized in that layer?

This is a fascinating question that almost certainly warrants further experiments to explore. Our observation is that the individual plates of polished material are discontinuous with ragged edges on the fragments, so we infer that they have broken. It is possible that these plates initiate in small patches within a slipping granular layer but if they are not coplanar with one another, they cannot link up without requiring stepping across deforming grains. They were probably never part of a single slip surface spanning the entire sample. Such roughness would not be sustainable when shearing in a rotary geometry so parts of the slip surface would be destroyed by the linkage. A brief discussion on this point has been added to the Discussion section. (Lines 218-226).

What is the role of temperature in the healing process (if any)? Heating during slip seems to be critical to reach low friction, does cooling account for the rapid re-strengthening immediately after the displacement is stopped?

We have added two different forms of temperature calculations to investigate the bulk temperature change in the sample. Both approaches indicate peak bulk temperatures increased only a few 10s of degrees during the experiments. Without a constitutive law available for internal deformation of the amorphous particles, it is not possible to relate this quantitatively to weakening, although the influence of any warming at all is likely to weaken the material. As this temperature anomaly is on average very small ($\leq 60^\circ$ for all experiments), we expect that the sample would return to room temperature within minutes to hours. This is consistent with the timescale of observed re-strengthening. This is also consistent with the timescale of interparticle bonding, so we can't be sure which is more important.

As the authors note in the introduction, the experiments appear to be very similar in terms of slip velocity, material used and mechanical data to the experiments performed by Di Toro et al. (2004) who concluded that the observed mechanical behavior is caused by silica gel, however no silica gel was observed in the experiments reported here – what is the key experimental difference that leads to the formation (or absence) of silica gel?

See response to similar question from Reviewer 3 below.

Line 56-57: "associated" is used twice in the sentence.

One of the 'associated's has been replaced with 'caused by'

Reference:

Di Toro, G., Goldsby, D., Tullis, T. Friction falls towards zero in quartz rock as slip velocity approaches seismic rates (2004) Nature, 427-6973, pp 436-439

Reviewer #2 (Remarks to the Author):

This paper describes the results of friction experiments on bare rock surfaces and they found dramatic frictional weakening at sub-seismic slip rates (~ 10 cm/s). The mechanical data is sound and convincing. The key observation is that they found widespread amorphous nanosilicate in their post-experimental samples. To confirm this, this manuscript contains a workman-like characterization of the Raman spectra and FT-IR analyses of their post-deformational samples. I have not done these analyses myself and tried to learn from the literature during the reviewing time (mostly from the reference list). Their analyses and the results shown in the paper do make sense to me. In other words, all the data are excellent, new, and well-conceived. Probably, further check from a specialist is needed.

However, I find that the discussion contains a lot of speculations.

1) Grain boundary sliding (plasticity) with hydrodynamic lubrication was inferred to be dominant weakening mechanism. However, without quantitative analysis, it is difficult to convince the readers that that plasticity of amorphous nanosilicate can explain the weakening. The authors have referred to the work by Yao et al. [2016]. In their more recent paper in GRL, they examined the possibility that if superplasticity can explain the weakening observed in nano-power of MgO. Although diagnostic polygonal texture with triple junctions has been observed in their samples, the calculation showed that the calculated flow stress is many orders of magnitude too high to explain the observed friction. In contrast, similar calculation has been done by De Paola et al. [2015], showing that plasticity can explain the weakening observed. Bear in mind that in either Yao et al. [2016], De Paola et al. [2015] or Green et al. [30]'s experiments, the samples were subjected to higher temperatures. If amorphous nanosilica can cause weakening under relatively low temperatures, more evidence, at least quantitative analysis, is needed.

In the Yao et al. and De Paola et al. studies, the deforming materials were crystals conforming to power law flow laws, so the temperature dependence of the rheology was understood and the strength could be calculated. In our case, we know the nanosilica is amorphous (has no long range order) but we have very little additional information about its structure and rheology. So, it is not possible to quantitatively predict the strength of the wear material. The friction data from the experiments demonstrate that the wear material is significantly weaker than the starting material, and increasingly weak at increasing slip rates. Our explanation is compatible with all previously published experimental observations and is well-founded in the materials science literature on the processes of amorphization and the behavior of amorphous materials and nanoparticles, including chemical and surface effects. Lines 169-176 now compare the crystalline materials of previous workers with the amorphous nanopowders we have produced.

2) Without some critical experiments, hydrodynamic lubrication is even more speculative. The evidence provided by the FT-IR analyses is quite convincing for the presence of

water layer, but it is difficult to draw the conclusion that it plays a crucial role in the lubrication. A simple experiment can be done under dry conditions (e.g. under N₂ environment).

Numerous studies in the materials science literature have established the ubiquitous presence and lubricating effect of a hydrated surface layer on silica glass (Donose et al., 2005; Opitz et al., 2003). Shearing experiments under dry conditions were performed by Tullis and Goldsby but the details have never been reported in the peer-reviewed literature. The results are summarized in a series of USGS Grant Reports. It appears that initial dry experiments did not show the precipitous weakening, suggesting that water is necessary for the mechanism to occur. However, subsequent dry experiments apparently did show the weakening effect for reasons not yet understood. As the original data are not publicly available, we have added a general statement of this result to the manuscript but are not able to explore the issues in detail (see Tullis and Goldsby 2002; 2003; 2004...Final Grant Reports, USGS)

3) In the present paper, the authors argued that “the faster slip rates (> 1 cm/s) are necessary to ... warm the interior of the amorphous particles above the glass transition”. How about the slip surface temperatures at different slip rates? If the temperatures is quite different, then it involves a sort of “thermal weakening”. If not, the weakening should occur at slip rate lower than 1 cm/s (related to Figure 6).

We now include thermal modeling which shows that the average temperature on the fault surface reaches no more than $\sim 60^\circ\text{C}$ in any experiment. Thus, higher temperatures can only be achieved at asperities due to slip surface heterogeneities (as is true in every experiment), and there is no modeling approach that allows estimation of the magnitude of peak temperatures at these locations as it is impossible to know the size, number, and spatial distribution of the dynamically evolving pattern of real contact area. We know that plastic deformation of particles occurred because it produced the fused shiny plates within the granular layer. However, a second weakening mechanism acts simultaneously (inter-particle sliding, lubricated by hydrated surfaces). So the bulk weakening effect must be attributable to both mechanisms and we cannot distinguish the relative importance of either one at any given slip rate.

4) The effect of fault healing is not straightforward in the present paper, at least from the mechanical data. The microstructure does show some grain sintering (inter-particle bonds), but similar structure/effect has been emphasized many times in the literature [e.g. Togo and Shimamoto, 2012].

We have added the thermal modeling and modified the discussion to more clearly describe the effects that we can constrain. We have added a citation to Togo and Shimamoto 2012 to note the strong similarity of their experimental observations to ours.

In terms of novelty, the observation of amorphous nanosilicate on bare rock surfaces and the related interpretations are new, although as pinpointed by the authors (in lines 127-128), “small amounts of amorphous silicate have been produced in other experiments at lower slip rates and higher temperatures, without weakening” [27, 28]. It is important to note that there are also some previous frictional experiments producing amorphous silica [Hayashi and Tsuyumi, 2010; Nakamura et al., 2012], both published in GRL (though it is difficult to judge just from the microstructure if there are the same or not).

These papers are cited extensively throughout the manuscript where direct comparisons are possible.

Finally, the paper is of great interests to both experimental and geophysical communities. The observation of the presence of amorphous nano-silica and dynamic weakening are both convincing, but they are not necessarily related. It would be a potentially important contribution if the authors can somehow show that flow stress of amorphous nano-silica is weaker or can be more easily activated by mild temperature rise than crystallized nano-silica. It would give more impact if the paper can compare silica gel and amorphous nanosilicate.

Silica gels take myriad forms, both in nature and in man-made materials, so a comparison with some ideal of silica gel would be wildly speculative. Likewise, amorphous silica can have a wide range of properties depending on chemistry, impurities, pH, structure, and deformational history so further constraints would require direct deformation experiments on wear material and artificial nano-materials, similar to Nakamura et al., (2012).

Specific comments

Figure 1: why negative displacement at the very early state?

The displacement transducer measurement was not precisely zeroed before the start of the experiment, but it has a shift of -0.015 that we subtracted from the displacement data

Line 107: If plasticity is related to the local stress at contact asperities, then plasticity is expected to be insensitive to the normal stress applied. Do you have experiments under variable normal stress?

We did not perform variable normal stress experiments.

Line 144: If amorphous nano-silica needs to be warmed in order to become weaker, experiments at lower slip rates and higher normal stress can also produce weakening. Is this the case?

Our experiments were all performed at the same normal stress.

Figure 6: I very much like this diagram. Do you mean “slightly interior warmed”? You really need to measure or calculate the temperature on the slip surfaces.

Thermal model has been added to the paper.

- 1) De Paola, N., R. E. Holdsworth, C. Viti, C. Collettini, and R. Bullock (2015), Can grain size sensitive flow lubricate faults during the initial stages of earthquake propagation?, *Earth Planet. Sci. Lett.*, 431, 48–58.
- 2) Yao, L., S. Ma, A. R. Niemeijer, T. Shimamoto, and J. D. Platt (2016), Is frictional heating needed to cause dramatic weakening of nanoparticle gouge during seismic slip? Insights from friction experiments with variable thermal evolutions, *Geophys. Res. Lett.*, 43, 6852–6860.
- 3) Nakamura, Y., J. Muto, H. Nagahama, I. Shimizu, T. Miura, and I. Arakawa (2012), Amorphization of quartz by friction: Implication to silica-gel lubrication of fault surfaces, *Geophys. Res. Lett.*, 39, L21303, doi:10.1029/2012GL053228.

- 4) Hayashi, N., and A. Tsutsumi (2010), Deformation textures and mechanical behavior of a hydrated amorphous silica formed along an experimentally produced fault in chert, *Geophys. Res. Lett.*, 37, L12305, doi:10.1029/2010GL042943.
- 5) Togo, T. and T. Shimamoto (2012), Energy partition for grain crushing in quartz gouge during subseismic to seismic fault motion: An experimental study, *Journal of Structural Geology*. doi:10.1016/j.jsg.2011.12.014.

Reviewer #3 (Remarks to the Author):

General Comments:

This is an interesting paper and should be published, but I believe it would be greatly improved by some revision. I have a considerable number of general and specific comments and hopefully considering them will improve the paper. Many of my comments are perhaps a bit self-serving and are intended to give more “credit” to relevant prior work (which I have been involved in over the years!) and to point out some interesting and in some cases puzzling contrasts between these interesting new results and prior ones.

These experiments themselves are very similar to ones done previously and referred to in reference numbers (4; 5), as well as described in my review paper that I refer to below as (#). However, the authors observe no silica gel in the samples from their experiments, which is in contrast to inferences as well as observations we have made on our experimental samples. Since the experiments were very similar and were done on essentially the same novaculite rock, this difference in whether or not silica gel was involved is puzzling.

Several possibilities exist, including that

- 1) gel was present in their experiments, but it was hard to see and they missed it
- 2) there is a difference in the water content of the samples, perhaps due to a difference in humidity of the experiments, and if these experiments were drier, gel might not have been produced
- 3) something about the combination of slip speed and normal stress made a difference
- 4) the lack of gel in their experiments could result from their weak samples being hotter than the previous experiments

The authors should consider the possibility of confronting this difference between their and previous results head on and try to explain it. As the paper now stands, the contrast is present, but it is not really addressed. This is not very satisfying for the literature as a whole. An explanation for the difference in observed gel is not obvious.

We wish to highlight that our study was motivated by the seminal work performed in Prof. Tullis' lab. Our manuscript, based on new microstructural and microanalytical investigations and techniques, that in some cases were not available 15 years ago, offers an alternate interpretation to explain silica weakening which seeks to replace the interpretation of Tullis and coauthors (Goldsby & Tullis 2002, Di Toro et al., 2004, Tullis 2015). In these seminal studies, the authors inferred the existence of silica gel and its critical role in dynamic fault weakening based on the interpretation of the following mechanical data:

1. experimental faults in quartz-rich rock were extremely weak when sheared for slips of several cm to meters from subseismic (Goldsby and Tullis, 2002) to seismic (Di Toro et al., 2004) slip rates;

2. the slip-weakening behavior was observed to environment-dependent. According to a series of slide-hold-slide experiments conducted in summer 2001 and presented in an abstract of the American Geophysical Union Fall Meeting in 2001 (Titone, B., K. Sayre, G. Di Toro, D. L. Goldsby, and T. E. Tullis (2001), *The role of water in the extraordinary frictional weakening of quartz rocks during rapid sustained slip*, Eos Trans. AGU, 82, T31B-0841.), the friction coefficient decreased under room humidity conditions and did not decrease under “dry” conditions (the experiment was conducted under an atmosphere of 100% N₂ on oven-dried novaculite). The “dry” N₂ atmosphere should impede the formation of silica gels, which requires the presence of H₂O. However, these interesting experiments were also extremely challenging to perform and, to date and as far as we know (one of the authors of our manuscript, Giulio Di Toro, co-authored these preliminary studies), these results have not been reproduced so far. Given the well-known outstanding reputation of the experimental data produced at Brown University by Prof. Tullis group, the lack of a second series of experiments that reproduces the first series may explain why these experimental data have not been published yet.

In any case, silica gel is not the only possible macroscopically thixotropic material composed of silica and water. In these previous studies published in peer-reviewed journals (Goldsby and Tullis, 2002; Di Toro et al., 2004) the authors did not report microstructural or microanalytical evidence in support of the presence of silica gels in the slipping zones. In fact, the ground-breaking contribution of these papers was not really the particular type of weakening mechanism, but perhaps that cohesive rocks such as novaculite, which according to the previous works have a friction coefficient of ca. 0.7 (the so called “Byerlee law”: Byerlee, Pure and Applied Geophysics, 1978) can have a much lower friction coefficient when sheared for slip distances and slip rates typical of moderate to large in magnitude earthquakes. These findings were confirmed by many studies in the years and opened a new field of research in earthquake physics and could explain several seismological observations.

As Tullis (2015) states, “*Extraordinary weakening that could not be explained by any other known mechanism was observed in a variety of experiments at relatively rapid slip rates and earthquake-like sliding displacements, eventually leading to the postulation of this mechanism.*” The existence of silica gel as a lubricating fault agent was inferred based on the data available, and this proposal was very reasonable based on that experimental evidence. Our paper provides an alternative idea, which is consistent with the experimental evidence (dynamic fault weakening) and is supported by microstructural and microanalytical studies (FTIR, TEM, micro-Raman, etc.). Based on the evidence reported here, our position is that there was no gel produced in our experiments and, perhaps, was not any gel in the previously published experiments either.

Tullis and Goldsby (Final Technical Report, USGS, 2006) also note briefly that the presence of comminuted wear material about 60 μm-thick may be necessary for the weakening mechanism to occur. None of the earlier peer-reviewed publications or reports had noted the existence of this wear layer and the thickness of the ‘gel layer’ was not reported. Is it possible that Tullis and Goldsby produced the same amorphous nano-silica we observe but instead focused on, and attributed the weakening effect to the ~3-4 μm-thick shiny layer on the surface of the novaculite core instead of the wear material?

Perhaps this layer is equivalent to the thin amorphous layers we observed adhered to the core surface (c.f. our Figures 2D and 5 which show wear material smeared to form a shiny film on the novaculite).

We have strengthened the language in the abstract and discussion (Lines 145-147; 180-182; 196-197) to place the interpretation in sharper contrast to the previous publications.

Reference: Tullis, T. and Goldsby, D. (2006) Laboratory experiments on rock friction focused on understanding earthquake mechanics, Final Technical Report for Grant Number 05-HQGR0087, US Geological Survey, 11 p.

Brief thoughts on 1-4 above are:

1) This possibility is addressed in discussion of lines 50-51 below. Summarizing, it is really hard to see the gel flow features and high magnification images of exactly the right spots are needed to see them. Which experimental sample(s) has been examined for the presence of gel is not given in the discussion of Figure 2.

Figure 2 shows images of experiments 97 and 101. Specific identifications have been added to the figure caption.

2) Perhaps the water content was somehow different, although both were done under atmospheric humidity, and in our gel-evident experiments extreme dryness was required to make weakening vanish. Their FTIR measurements show that at least some water is present in these experiments too. So it's hard to see how their samples could be so dry as to eliminate gel.

It is reasonable to assume that there was no substantive difference in humidity between our experiments and the Goldsby and Tullis (2002) and Di Toro et al. (2004) experiments. The difference is in the kind of microstructural and microanalytical observations we made to characterize the wear material. To date no microstructural or microanalytical characterization data has been published in peer-reviewed journals by Tullis and collaborators, so we cannot make any specific comparisons with their wear material. The single photo (which appears in Niemeijer et al. 2012, and same photo in Tullis, 2015) shows some sintered material. The latter is labeled 'unslid novaculite' in Tullis 2015, but does not resemble the novaculite (shown in our Fig 2A) but does resemble our layers of particulate material: our Figure 2. Based on our study of wear material from multiple experiments we are able to identify what is characteristic of the wear material. No comparable observations are available on record from Tullis and collaborators so it is not possible to make a direct comparison at this time.

3) We see similar weakening at conditions with slip velocities of 3.2 mm/s and normal stresses from 28 to 112 MPa, to 0.1 m/s and normal stresses of 5 MPa, with slip displacements from ranging from 0.7 m to 63 m. So although our normal stresses are never lower than 5 MPa and those in this work are all 2.5 MPa, the experiments otherwise span the same conditions. We have not examined samples from all of our experiments that show weakening with SEM imaging like that shown in (#). However, the mechanical behavior of all the experiments are similar, so we infer that silica gel is responsible for all our observed weakening.

As discussed in our responses to previous two comments, our experiments showed the same mechanical behaviour as Prof. Tullis' experiments. We wish to stress that, motivated by the previous studies conducted by Prof. Tullis' group, we wished to perform dedicated microstructural and micro-analytical investigations (that in some cases were not available 15 years ago) to identify the presence of silica gels in the experimental slipping zones. We found other types of compounds made by and water, but not silica gels.

4) The temperatures presumably do not get higher in the experiments of this work than in those of (4; 5), so it is hard to see how this could be the explanation, although temperatures are not given for this present work. Higher temperature could dry the samples out enough to eliminate gel.

This is a very important point. Two forms of thermal modeling have been performed, see new Fig. 1B. Our peak temperatures in the high velocity experiments did not exceed 60° on average, similar to those of the Brown University group.

Specific comments by line number:

Sentence in middle of abstract: "Two slip rate-dependent mechanisms cause the weakening: thermally-enhanced plasticity, and hydrodynamic lubrication." After reading the paper I do not believe that they have proven this statement with regard to either mechanism. They suggest them as hypotheses to explain their data, but they by no means prove that these explanations are correct. The sentence should be changed to something like: "Two slip rate-dependent mechanisms are plausible explanations for the observed weakening: thermally-enhanced plasticity, and hydrodynamic lubrication." Note however that my comments referring to line 118 suggest that hydrodynamic lubrication is not a term that should not be used for what they envision.

Modified as suggested

Line 2: insert a new reference number (I'll call it #) after the "2" in "(2)", so "(2; #)". That, somewhat difficult to find but very relevant, reference is:

Tullis, T. E. (2015), Mechanisms for friction of rock at earthquake slip rates, Chapter 5, in Treatise on Geophysics, v.4, Earthquake Seismology, edited by H. Kanamori, pp. 131-152, Elsevier Ltd., Oxford.

It should be cited in general somewhere for high speed weakening mechanisms – this seems like a good place. I've sent a digital reprint to the first author, who can distribute it to the others.

Reference was added

Line 10: might change the end of the sentence "... for silica gel" to "... for silica gel, even though the conditions are similar to those for experiments in which silica gel is observed (4; 5; #, Fig. 11)." Here # is intended to be that citation number for the new reference above. Fig. 11 of that reference shows a SEM image of silica gel from the experiments of (4; 5).

The image was produced in one of the experiments performed in Prof. Tullis lab and we now cite the paper suggested by Prof. Tullis in his review. However, since there are, as far as we know, no dedicated microanalysis published in the literature that show that the structure reported in the figure is made of silica gel, we wrote "interpreted as silica gel" (refs. 5 and 6).

Line 11: insert "to silica gel" between "explanation" and "for"

Line 25: Add to the end of the sentence that now ends in "other" so it ends "other at atmospheric humidity." No mention is made of the humidity, but I assume it to be uncontrolled in room air. We have found that the humidity makes a huge difference in both the weakening and the gel formation and so the humidity should be stated.

Modified as suggested.

If its dry enough, the weakening goes away, but that has to be extremely dry. So far this is only published in this AGU abstract:

Titone, B., K. Sayre, G. Di Toro, D. L. Goldsby, and T. E. Tullis (2001), The role of water in the extraordinary frictional weakening of quartz rocks during rapid sustained slip, *Eos Trans. AGU*, 82, T31B-0841.

We are familiar with the Titone et al abstract, but the data, as mentioned above, have never appeared in peer-reviewed form. In addition, a Tullis and Goldsby (2003) grant progress report stated that Titone's experiments, due to the technical challenges, could not be replicated after many attempts (attempting much drier conditions than the initial experiment). Given the difficulties in replication, we cannot repeat the Titone et al. interpretations with certainty. We have added a statement to this effect in the introduction. Clearly there is a need to replicate these experiments using dedicated assemblies (e.g. a vacuum chamber). However, as stated before, maybe silica gel is not the only possible macroscopically thixotropic material composed of silica and water. So, these studies should be supported by the state-of-the-art microstructural and microanalytical investigations.

Citation: Tullis and Goldsby (2003) Laboratory experiments on rock friction focused on understanding earthquake mechanics. Final Technical Report for Grant Number 02-HQGR0070, US Geological Survey.

Line 27: "Equivalent velocities" is referred to here. This needs to be defined or explained. The slip velocity goes from 0 in the center of the solid cylindrical samples to a maximum at the outer diameter. To what radius do the quoted velocities refer?

We use Hirose and Shimamoto (2005)'s definition of equivalent velocity as the velocity that corresponds to the total frictional work done on the sample (integrating across the radius). The equivalent velocity corresponds to the slip rate at about 2/3 of the sample radius. A citation has been added.

Line 37: the clusters referred to here are mentioned enough times that it would be helpful if they were pointed out in Figure 2 as well as in Figure 3, since Figure 2 is referred to here. At this point, a reader is not sure if you are referring to the fairly large plates visible in Figure 2 B&C and/or to smaller pieces visible there and in Figure 2D.

Modified as suggested.

Figure 2C: Magnification is not adequate to show the 10 um pores referred to in the caption. Point to them with an arrow if they are important and/or increase the magnification. What is the point of referring to them?

A new image has been added to Figure 2. We consider it important to compare the original surface of the novaculite to the post-shearing surface in order to ascertain which features of the post-shearing surface are uniquely related to shearing. The pores on the original surface (Fig. 2A) contribute some roughness to the sliding surface and their absence on the post-sliding surface (Fig. 2D-F) demonstrate smearing/coating of the surface by wear material.

Line 50-51: It is not clear to me that the lack of observation of “viscous fluids” on the slip surfaces means that they were not there. Note that the SEM observations of what were formerly viscous fluids in the experiments described in (4; 5: and #) were made at a much higher magnification than those illustrated here in Figure 2 (see #, Fig 11) and the gel was not easy to distinguish. The layer of silica gel in those experiments was only ~3 microns thick and when looking perpendicular to the smooth slip surfaces no viscous flow type features were visible. It was only at the margins of pre-existing pits in those smooth surfaces where the gel had been partially smeared into the pits that the flow features could be seen. It seems quite possible that unrecognized gel exists in the experiments being reported on here. Whether the weakening reported on here could be due to the existence of gel, simply to the presence of the clumps of amorphous nano-particles, or perhaps some combination, seems to be presently unclear.

The exact image presented by Tullis in his 2015 paper is, to our knowledge, the only image of the Brown lab’s silica gel ever published. The initial publications reporting the existence of silica gel clearly stated that the presence of gel was *inferred* from the hold-strengthening behavior. No microanalytical characterization of the wear material was reported at that time, and we believe that it is not only the shape of the microstructure that matters, but also its chemistry and physical state (glass, gel, etc.)

The microstructural and microanalytical observations presented in our manuscript were made at a variety of scales from cm to nm, using FE-SEM, FTIR, micro-Raman, microprobe and TEM. We observed the entire slip face of several samples. We present images at a wide range of magnifications looking for the microstructures described by Prof. Tullis and co-authors. We spent tens of hours at the FE-SEM and TEM and we did not find any example of “fluid-like” structures similar to those reported by Prof. Tullis and co-authors. We thoroughly investigated by the slip surface face-on and in cross-sections of several samples. We are confident we have observed the representative materials and structure of the slip surface and wear material. The wear layer is dominated by nanopowder. In many 10s of hours of observation by three of the authors, we observed no ‘gooey’ textures, stretched material in bridges, or smooth rounded pores that characterize the image published by Tullis’ group. The description given by Prof. Tullis in this review comment is the most detailed we have found and suggests that the ‘gel-like’ textures are quite rare on the sample surface. According to Tullis and Goldsby (2006), particulate wear material may have been more abundant but was not studied.

The reviewer suggests that the 'gel' textures are present in our samples but we missed them. We feel that we did due diligence with careful microstructural and microanalytical investigations (tens of hours of FE-SEM and TEM observations and tens of hours of FTIR and Raman spectroscopy studies) as recognized by the two other reviewers. Our conclusion is that the so-called "gel-textures" were not present in the slipping zones produced in our experiments. We can only speculate that in the 2002 experiments, the flow structures found were due to solidification of friction melts produced at the asperity scale (which may be unlikely given the low average temperatures achieved on the sliding surfaces), or were smears of amorphous material. In fact the gel "flow" structures were found in slow slip experiments performed at large normal stress (Goldsby and Tullis 2002) may be similar to the amorphous flowing material described by Pec et al. (2012, EPSL) produced at large confining pressures in granitoid rocks.

Here we wish to focus on the microstructures found in our experiments as we know exactly where they are located in experimental slip zones and after thorough study, we can rule out any features similar to Tullis and co-author's flow feature. Moreover, as stated above, we wish to highlight that we did not find silica gel in any of the many analyzed samples from our experimental slip zone.

Line 52-53: These observations of the wear material are good and useful, but note that it is very difficult in TEM to image the actual smooth slip surface shown in Figure 2B. We tried to do this in our 1990 TEM study of sheared gouges (27), but found that the thinning process of making the TEM foil removed the material right at the slip surface. Modern FIB techniques might get around this problem, but the discussion in this paper does not indicate that they tried to or were able to look directly at the material exactly adjacent to the main slip surface in TEM. The observations of apparently relic slip surfaces in Figure 3 C are quite interesting, but these surfaces may differ from ones that had very large amounts of slip on them.

The TEM foils were prepared from thin sections made from intact (both rock cores and wear material kept together) experimental samples. They were prepared in cross-sectional view from previously mapped and documented thin sections (e.g. the section shown in Figure 2B) so the boundaries of the slip zone and the location of discrete localized surfaces is unambiguous. One of us (JCW) has specialized in detailed TEM studies of similarly delicate natural and experimental fault materials over several decades. We have added a section to the supplementary material to explain our process so that future workers may also have more success. Attempts to prepare TEM foils using SEM-FIB resulted in destruction of the samples. The nanostructure of the wear material is sufficiently delicate that we conclude that the Ga beam of the FIB is too energetic to prepare a TEM foil without destroying the essential structures we wished to observe. JCW's method of preparation with a lower-energy ion beam is much slower and more gentle and resulted in excellent preservation of micro- to nano-structure (Fig. 3).

Line 73 (and on line 135): I understand the idea of oligomeric complexes but why are they referred to as rings? I realize that space is short, but a tiny bit of background here would help. What makes them have that shape or what makes us think they do or should? Actually, though I still don't know why rings occur, the first paragraph of the Supplementary Materials discuss this a bit. I suggest just adding a ref in this part of the main text to the Supplementary Materials and then in that location there would be room to

perhaps explain why rings occur. Clearly this question is tangential to this paper, but it would be helpful to get a bit more on this into the Earth Science literature!

A review of silica oligomerization and ring structures is outside the scope of this paper, but the extensive literature we have cited establishes that the oligomeric rings are characteristic for amorphous silica in various forms. Because of space limitations, we have added a suggested reference to the relevant section in the supplementary material for further reading.

Line 79: Is the wavenumber range of expected Si-OH or adsorbed water within the range shown in Figure 4A? It would help to give the wavenumber range expected, and either point out that it is, or is not, covered in Figure 4A.

The characteristic water bands are at 2800 and 4000 cm^{-1} , and scans out to those wavenumbers revealed no H_2O . Raman spectroscopy is relatively insensitive to the presence of water, so we performed the FT-IR spectroscopy (Figure 4B).

Figure 4A. The title of this on the figure uses the word “Chert.” Change this to “Novaculite” to be consistent with rest of paper.

Modified as suggested.

Figure 5: The writing on the axes is too small to read, even when blown up. I suggest trying labeling every other major tick mark and doubling the font size. This is a problem on all three axes. It is not clear to me that any of the parts of this Figure really add that much to the paper – consider omitting. The only point seems to be that the amount of different materials varies on the surface, which Figure 2B already shows, albeit at a finer scale. Nothing is made of the difference in the amount of material on the surface or the difference in the spectra between the 0.01 and the 0.0001 m/s sliding rate samples illustrated. If there are some points to be made, then retain the figures and say more about them.

We have increased the font size on the axes on Figure 5 (now Figure 4). We added some more description to the text to emphasize the important observations displayed in this figure (lines 83-97). The differences in absorbance may be attributable to path length through the wear material where there are thickness variations.

Lines 88-94: If anything can be said about the amount of water or OH in the sample from the spectra, then something should be said, or if it is not possible, then say that. This is an important issue since whether this material should be considered a gel or what the viscosity of the gel might be at different strain rates presumably depends on water content, even though we probably don't know much about that. So what quantitatively can be said from the combination of the IR and the Raman data would be useful to say.

As we reported in the submitted manuscript, the amount of water in the wear material is insufficient to be detected by Raman spectroscopy. Further quantification is not possible due to uncertainties in path length due to thickness and packing variations.

Lines 90-101: Would be helpful here to refer to references 4 and 5 and their description of the thixotropic behavior of their gel, which is exactly what you observe with your material.

The role played by the competition between time dependent creation of bonds and the strain dependent disruption of them seems the same in either case.

Modified as suggested

I initially thought that perhaps the situation is not the same here, since in that earlier work weakening was observed at lower velocities (4) where the temperature was inferred to not get higher than 140 deg C. Here the weakening is only seen at slip velocities of 0.1 m/s where the frictional heating can raise the temperature higher. But actually it seems likely that the temperature is not higher in these experiments than in the experiments of (5) where the velocities are the same, the slip distances are similar, and the normal stress is actually twice as large as in the current work. As mentioned above in the discussion of point 4) at the end of my general discussion, it is unlikely that the temperature was higher in these experiments than in all of those earlier ones that show weakening (4; 5; #). On line 111 “mild frictional heating” is mentioned. In fact, it would be very helpful to give some estimate of the temperature likely to have been attained, based on some measurements or calculations. All that is said (line 112) is that it inferred to be above the glass transition temperature. Giving values of the estimated temperature as well as whatever is known about the glass transition temperature will make it possible for the reader (and reviewer!) to evaluate how likely this inferred crossing of the glass transition temperature is.

Thermal models have been added to the paper (Fig 1), but this only represents the average temperature rise on the surface, while temperatures and stresses are likely to be much higher than average at some asperity contacts. This concept is further explored in the discussion (Lines 159-168):

“The exceptional concentrations of defects and vacancies in amorphous silica nanoparticles encourage hydration and plasticity (14; 16; 20), especially at the small scales and high pressures similar to that expected at asperity contacts (Fig 6A; 32; 33). Yao et al.(34) have shown that heating, not just the presence of nanoparticles, is important for slip weakening. Hydration effects reduce the glass transition temperature and further promote plastic flow (35; 36). The existence of the smeared nanoparticles forming the striated plates implies that the mild average frictional heating (Fig. 1B) experienced in the shearing experiments at 10 cm/s was sufficient to bring local patches above the glass transition temperature, allowing superplastic flow and the cohesion of wear powder into the striated plates. The shear weakening effect would be reversed upon cooling, within seconds after the cessation of shearing.”

Line 118: The term “hydrodynamic lubrication” is used here as well as on lines 104 and in the abstract. I don’t think this term is appropriately used here and I recommend removing mention of it in all three of these places. In engineering practice, in a 2001 paper by Brodsky and Kanamori, and in the review paper (ref #), this term is used to mean flow of bulk fluid that builds up normal-stress-supporting pressures by flowing in appropriately shaped confined spaces between sliding solids. Here you use it to mean shearing of mono- to multiple-layers of water. I strongly discourage the use of this term for what is envisioned here. I fear it will lead to confusion in the literature since it then would be used with multiple meanings.

With respect, we disagree with the reviewer. We have demonstrated the existence of a fluid layer on the outside of the wear particles. Hydrodynamic lubrication is defined as the condition when a continuous layer of fluid separates the solid materials during shear so they are not in frictional contact, and the shear resistance is then controlled by the properties of the fluid layer. The definition offered by the reviewer is more similar to the definition of *elastohydrodynamic* lubrication (which was the subject of the Brodsky and Kanamori paper), which is the special case of significant pressurization of the fluid during hydrodynamic lubrication. We do not invoke elastohydrodynamic lubrication because we have no information about the fluid pressure and assume that the fluid films coating the wear particles were unconfined. Therefore, hydrodynamic lubrication accurately describes our view of the conditions for granular flow in the wear material. It is important to keep this mechanism separate from intragranular plasticity for the healing step as well.

In fact it seems to me that what you describe is not really a second mechanism, but rather an essential component of the superplastic flow that you describe in the previous paragraph. For superplastic flow to occur one needs not only the particles themselves to be able to deform so as to accommodate the changes in shape needed as they change neighbors, they also have to be able to undergo sliding on the boundaries of the particles. This would be enhanced by the presence of thin water films on the surfaces of the particles as you illustrate in Figure 6. Thus I suggest that you do some rewriting of these paragraphs to integrate the two thoughts into two components of one weakening mechanism. This is a standard way of breaking down the processes in structural superplasticity, namely it occurs by grain-boundary sliding (one component) with either dislocation or diffusion accommodation (the other component).

It is correct to say that as originally defined, superplastic flow requires two contributing mechanisms. In this case (perhaps in all cases) the two mechanisms may follow different weakening paths as conditions evolve during slip. So it is still important to discuss them separately, even if the behavior of the whole can be described by a single term. This is particularly relevant when comparing weakening mechanisms in amorphous nanoparticles to similar processes in crystalline nanoparticles as has recently been described for calcite (e.g. De Paola et al., 2015) where rheological laws are available to describe one of the component processes.

If this doesn't quite capture your thoughts about how the two components contribute to the deformation, then discuss that explicitly. For example, perhaps describing it as plastically accommodated grain-boundary sliding puts too much emphasis on the grain boundary sliding component, then contrast it with that idea and suggest that it is better described as plastically deforming particles that are also able to easily slide past one another.

In this paper we seek to explain not only the weakening phase of the deformation, but also the re-strengthening after cessation of shear. These two phenomena work in concert during weakening but the recovery of each mechanism would progress on different timescales. Therefore we retain the original structure of our discussion differentiating the two mechanisms, and have added a sentence to the discussion to clarify the comparison to the superplasticity as the reviewer suggests.

Line 127: In ref 27 it was not small amounts (i.e. was 40-50% - Table 1 for quartzite in ref 27). Perhaps just remove "Small amounts of" from the start of the sentence and change "have" to "has".

Modified as suggested.

Line 128: Change to “rates and lower temperatures (27) and lower rates and higher temperatures (28).” I suggest this change since ref 27 is at room T, which the sentence implies not to be the case.

Modified as suggested.

Line 128-129: I suggest that after the reference to (28), replace the rest of the sentence with “Weakening seems to require both slip to occur at intermediate to high velocity and slip to be sufficient to form a continuous amorphous layer (4).” The later was true in the experiments of (27; 4; 5; this work) but without elevated slip rate there is no weakening. Elevated slip rate can be as slow as 3.2 mm/s in the work reported in (4). Also note that in ref (4) at the end of paragraph [17] we suggested that sufficient slip is needed to make a smooth surface of amorphous material and gel, hence my insertion of ref [4] as the end of this sentence.

We have added the attribution to Goldsby and Tullis for suggesting that continuity of the wear material layer is required. However, the quantity/thickness of wear material has not been reported for any of the previously published experiments so we do not have access to data to support the complete statement.

Line 136-137: Not clear that rolling matters – if it does, why isn’t the material weak at low slip velocities? I suggest omitting this sentence?

Modified as suggested

Comments on Supplement:

First two lines: R band is mentioned – what does that mean? This is referred to again near the end of this paragraph. Giving wave numbers for R band would help. It is not discussed in Figure 4A, caption, or text.

Also the discussion on lines 2&3 with the notation in parentheses isn’t clear to most readers, although I suppose to a Raman expert it would be. Adding an explanation of the nomenclature would help.

Additional background information and definitions have been added to the supplement.

FTIR section, second paragraph, first sentence: Complementary to what? The previous paragraph was already talking about IR results so exactly what this sentence is trying to say is unclear. Rewrite.

We have re-written. Apologies for the earlier errors.

Last paragraph of this section: Both “694” and “695” are used to refer to the same peak. “695” was used in first paragraph. Decide which to use and use only one.

Modified as suggested.

We thank the Reviewers and Editor for very helpful comments, and especially thank the Editor for their patience.

Reviewers' comments:

Reviewer #1 (Remarks to the Author):

2nd review of Rowe et al. "Earthquake lubrication and healing explained by amorphous silica"

The main concern raised by the second review of prof. Terry Tullis is the fact that no "silica gel" (H₂O rich silica exhibiting fluid-like microstructure) was observed in the experiments reported by Rowe and co-authors and the fact that Rowe et al. suggest in the current manuscript that no silica gel was present in the earlier experiments of the Brown group. The second review provides ample details and pictures of the material observed in previous experiments conducted at Brown. I am convinced that the material shown in the reviewer attachment is indeed some kind of amorphous silica gel as proposed in earlier papers, and that this material is distinct from the material observed by Rowe and co-authors. Unfortunately, these images were never published in peer-reviewed journals so the authors of the current manuscript lacked this important information when writing their paper and replying to the review. This episode highlights the problem with very short papers where not enough space for full data presentation is available in my point of view. The two possible explanations offered by Terry Tullis, i.e. 1) either large difference in SEM magnification used to document the amorphous material in the reviewer attachment; or 2) one order of magnitude lower slip distances in the experiments of Rowe et al. compared to previous experiments are both credible. Rowe and co-authors disclose in the rebuttal that many hours were spent at the SEM looking for silica-gel like microstructures but none were found. I assume that the published pictures are only a small subset of all images produced and analyzed and therefore I tend to favor the latter explanation that not enough slip was induced in the samples to form silica gel – this might be the easiest explanation for the discrepancy. Previous experiments on granite gouges sheared to large strains under low stresses (Yund et al. 1990) as well as sheared to modest strains under high stresses (Pec et al. 2012, 2016) indicate that the total amount of work (i.e. force x displacement) is what ultimately matters for the production of very fine-grained, even amorphous materials in my opinion.

The revised manuscript of Rowe et al. is of high quality, the findings are well documented, timely and very significant so I suggest publication with minor revisions and integration of the comments from the second review of Terry Tullis. The concerns from previous reviews were well addressed in the rebuttal and updated manuscript. I think that the issue of silica gel presence & absence will be best solved if the authors acknowledge its existence in previous experiments and discuss the differences in the experimental set-up and total slip experienced by the samples as a possible solution to the mystery. One further and poorly-traceable difference in the experiments could be due to different room humidity conditions in Providence and Padova given that the gel is clearly hydrated and doesn't form (not reproducibly apparently...) in very dry rocks. Future experiments should aim at controlling or at least monitoring humidity during the experiments. More work will be certainly needed to fully resolve this discrepancy and I hope that both groups will continue working on this fascinating subject.

Minor point:

Line 130 (...≈15-25 MPa... should be GPa)

References:

- Yund, R. A., Blanpied, M. L., Tullis, T. E., & Weeks, J. D. (1990). Amorphous material in high strain experimental fault gouges. *J. Geophys. Res.*, 95, 15589–15602
- Pec, M., Stünitz, H., Heilbronner, R., Drury, M., & de Capitani, C. (2012). Origin of pseudotachylites in slow creep experiments. *Earth and Planetary Science Letters*, 355–356(0), 299–310. <https://doi.org/10.1016/j.epsl.2012.09.004>

Pec, M., Stünitz, H., Heilbronner, R., & Drury, M. (2016). Semi-brittle flow of granitoid fault rocks in experiments. *Journal of Geophysical Research: Solid Earth*, 121(3), 1677–1705.
<https://doi.org/10.1002/2015JB012513>

Reviewer #2 (Remarks to the Author):

The 2nd round review:

The authors have put substantial efforts into the revision and have provided detailed responses to my and other reviewers' comments. In some sense, the paper was renovated; now it reads much better, and the logic makes more sense to me. I appreciate the authors' attitude on re-visiting the previous work published on high-impact journals and providing new interpretations based on detailed observations using advanced techniques. The paper is of great interests to experimental and field-geology communities. This work would be suitable for publication in *Nature Communications*.

I have one major concern on the proposed weakening mechanism(s):

First, for any granular (particulate) material, all possible weakening mechanisms play their roles at two scales: intergranular interactions and intragranular plasticity. Therefore the proposed weakening mechanisms (particulate flow and intraparticle plasticity) cannot be wrong, although I do not think it is appropriate to refer to Ashby and Verrall (1973) to illustrate the combination of these two mechanisms (the added sentence 148-150). In the theory proposed by Ashby and Verrall, grain boundary sliding (gbs) is mostly an artificial effect (deformation is accommodated by grain boundary diffusion).

Second, the revised manuscript added the temperature calculations (Figure 1). I agree that "hydration effects reduce the glass transition temperature and further promote plastic flow". However, by looking at the manuscript (e.g. Figure 6) and the response letter, the authors tend to believe that rapid weakening is caused by frictional heat (temperature rise) at local asperities, which one would easily think to be "flash heating". Here my point is that the temperature calculations do not straightforwardly support the interpretation. To go beyond this point, my suggestion is to investigate the flash weakening temperatures of the amorphous nanopowder, although I understand the calculation as such would be suffer from uncertainties.

Third, I found the statement of thermally-activated mechanism is confusing (e.g. abstract). Yes, intraparticle plasticity is a thermal activated mechanism. Is not particulate flow (intergranular flow)?

Forth, the authors argued that "intraparticle plasticity" is attributed to the activity of grain boundary sliding, although the involved processes might be different from that for crystalline nanoparticles (Lines 169-176). Is "grain boundary sliding" not a specific type of particulate flow? To my understanding, particulate flow (intergranular flow) involves a series of processes including grain boundary sliding, grain rolling, grain wearing and so on.

My second concern is about the abstract:

The abstract reads appealing; but after reading through the manuscript for several times, I found the application to fault healing is not thoroughly discussed. It is not clear how these two weakening mechanisms correspond to different time-scales of fault healing either.

Typo in the abstract: "10s-100s years"

Reviewer #3 (Remarks to the Author):

Review of Revised Rowe et al Paper by Terry Tullis

Note this review is not nearly as succinct or as organized as I would have liked. Some of the points are made in multiple places as they arose in different ways. If I spent a lot more time on it I could consolidate the similar comments so there would be less reference to saying something similar elsewhere. However, I've already spent too much time on it and the editors understandably are wanting my response!

This review contains several components:

- 1) Comments prompted by the reviews of the other two reviewers and the authors' rebuttals
- 2) Responses to the authors' rebuttals of my original review
- 3) Images from a PDF file that was created from a PPT file that is pertinent to both item 1 above and 2 below
- 4) Specific review of the revised manuscript, both general comments and specific line-by-line ones.

1) Comments prompted by the reviews of the other two reviewers

Reviewer 1

Comment and reply concerning lines 44-51. It is interesting that our experiments and those described in this paper differ in that our samples produced a continuous shiny slip surface separating the two sample halves, as the image in Figure 1 below shows. I assume the difference to be the total amount of slip. Note that the slip for the Figure 1 sample was 62 meters, whereas for most of theirs was 3 meters and the largest two, that they do not image in the paper, was 30 m.

Reviewer 2

I agree with the general comment that the "discussion contains a lot of speculation." In general, in my opinion, the proposed processes for weakening are presented in a much more definite way than is appropriate for what is known to be true. I discuss this in more detail in many places below. Relative to thermally induced weakening (item 1) contributing to what they now call 'particulate flow,' I think it is pretty speculative to say that the modest temperature rises that they have now calculated are sufficient to cause a transition across the glass transition. What is required for that to occur in this amorphous material is presumably not known, and it is certainly possible, However, it remains speculative and the statements made are much more definitive than is warranted. Regarding "hydrodynamic lubrication," (item 2), I discuss this in more detail below, In contrast to this reviewer, it does seem plausible to me that some H₂O molecules along particle boundaries could well be responsible for a reduced resistance to interparticle sliding. It remains speculative, but seems plausible to me. My biggest objection to this component of their explanation for the weakness of their samples is the use of the term 'hydrodynamic lubrication' to describe it. More on that below.

In the rebuttal to item 2, the authors say that "Subsequent dry experiments did show the weakening effect of reasons not yet understood." We think they were not

dry enough, as is noted briefly on Figure 9 below. I discuss this more fully in a paragraph where I comment on their rebuttal of my original comments (search for the word 'dry' – the paragraph contains it numerous times.

It is interesting that the authors response to Reviewer 2's last general comment is that "silica gels take myriad forms" even though the reviewer had not used the term 'gel.' In our earlier work we did not try to go into any specific description of exactly what we meant by silica gel for exactly the reason stated by the authors. Our intent was that it meant non-crystalline silica, with some unknown quantity of water included somehow within it, that was able to flow at high shear rates and be brittle at lower rates, namely that it was thixotropic. Our qualitative explanation for this was that its alternate behaviors involved a competition between a strain-dependent tendency to depolymerize bonds and weaken and a time-dependent tendency for bonds to strengthen and polymerization to occur. We referred to publications that showed silica gels to be thixotropic. So we tried to hit a happy medium between qualitative explanations that made sense without trying to go into unwarranted speculations about the details of the processes occurring during deformation in the weak state. As far as we were concerned, the flow could occur on a molecular scale and involve relative motion all the way down to the level of silica tetrahedra, or it could involve motion on a larger scale between globules or particles of silica gel that often exist. The observed flow features in the highest magnification images in Figures 2-5 (i.e. Figure 5 suggest that the scale of the flowing units may be pretty small.

The response to Reviewer 2's specific comment that refers to line 144 suggests that either lower slip rates or high normal stress could also produce weakening. The author's rebuttal is only that they did not perform experiments at other normal stresses. But they did perform them at lower slip rates. The result in terms of weakening was that some was observed at 0.1, 0.01, and 0.001 times lower than the their "standard" rate of 0.1 m/s. However it is interesting that the weakening does not become progressively lower as the velocity decreases – the strength at 1 mm/s is an outlier, being lower than at both higher and lower speeds, rather than being intermediate. For unknown reasons friction data is notorious for not being as reproducible as one would expect, so I don't find this particularly problematical, but it does weaken the story a bit. The bottom line on this is that there is not significant weakening at lower velocities (10 cm/s and 10 microns/s) but of course how much one would expect isn't clear! So while I agree with the expectations of the second reviewer concerning the expected effect of velocity on the weakening if it is thermally induced, I think the results are ambiguous.

2) Responses to the authors' rebuttals of my original review

At the end of one paragraph in their rebuttal the authors say:

"Based on the evidence reported here, our position is that there was no gel produced in our experiments and, perhaps, was not any gel in the previously

published experiments either.” This is at the heart of my objection to their position and hopefully the attached images and my suggestions as to why they could have missed the evidence for flow, will change their minds.

Clearly the difference is in the explanation for the weakening we both observe. At the moment I cannot conclusively explain why Rowe et al. have not seen the textural features that we did, although I offer some likely suggestions later in this review. Note how large our displacements were (10-20 m) to get the sufficiently smooth surface for the gel on it to act as a lubricating layer. This is in contrast to the much more rapid weakening with slip shown by Rowe et al. in Figure 1 of their ms.

Unfortunately we did not publish SEM images or the chemical analysis we made of the smooth sliding surface at the time we submitted our published papers. I think we did not have had those results by then, I’m not sure, but we got them later and they confirmed our original interpretation of silica gel. Publishing just the images and the chemical analysis for water seemed too little to make a publication, so we never submitted it anywhere. In this case it clearly is a problem. I did include one SEM image in my review paper (15) that the authors note in their response, but notably in this discussion they only refer to what we label “unslid novaculite in pit” and make no comment on what we considered to be most significant features in the image [which is Figure 2 below and also Figure 12 in ref (15)] to the clearly marked flow features (“flow features; silica gel”). I may or may not have chosen the best single image, to include in (15) but as can be seen from the attached, there are more of them in a similar vein.

The images below not only show more instances of the flow features in higher magnification SEM near the edges of the pits than the SEM images of Figure 2 of the submitted ms, they also include some images related to other items in the same long section of their rebuttal. One slide shows the determination of the H content in that thin surface layer done by FRES. It shows that the water content of the surface layer has a water content of around 2%. This technique has the advantage that it can analyze a surface layer, does not need such thick sections as does IR, and is more sensitive than Raman.

So I still believe we had weakening by silica gel. Note from one slide that it doesn’t occur if there is less than 50% silica in a sample, something that presumably would be found by the authors if they studied other rocks.

I also included the results the authors refer to in their rebuttal that show how our undergrads managed to turn the weakening on and off by alternately wetting and drying the samples. As the authors noted in their rebuttal, this is only in an abstract and the figure is not in the literature. I do not see how our students could have done anything wrong to produce these results! It is notoriously hard to remove trace amounts of water from samples and when we tried to replicate the right amount of drying we couldn’t – I think that is because we didn’t hit a magic window of the right amount of heating and simultaneous dry atmosphere when we did nominally the

same thing they did with a heat gun. We either didn't get it hot enough and it didn't get dry enough, or it got too hot and the epoxy holding the samples to the steel grips debonded. We lost patience with trying, but it would be good to go back and try again to do!

As you can see, I still believe we had flowing silica gel in the surface layer of our samples and that it is responsible for our observed high-displacement weakening. Hopefully these images will convince the authors that we had such flowing material in our samples, even though Figure 12 of (15) [Figure 2 below] seems not to have done so! They say at the end of their long response to my comment 3 “We found other types of compounds made by and water, but not silica gels.” Elsewhere in this review (detailed comments relative to Lines 51-52, 58-62, and 198-199) I give possible explanations as to why they might have missed such flow features in their careful imaging. In some sense this discussion may be simply semantic, since we both believe we have an abundance of amorphous material with a non-negligible water content. One of the main reasons we still prefer the term silica gel, is that it is descriptive and the word gel implies a sort of continuous flowing plasticity (with the shifting parts moving relative to one another at some scale smaller than the observations). In fact it is not clear to me that if they were to do TEM analysis of something that we would call flowing silica gel that it would differ too much from that they image Figure 3. Still, I suspect our material would have less fragmented- and particulate-appearing material, perhaps because it has undergone much higher strain. It would be interesting to do TEM on our gel.

With reference to the question of whether they had flow-like features in their samples and their arguments that they did not, they give a description of their efforts to find it in their long response to my comments on lines 50-51. All I can say is that in spite of all of this, it is obvious that we have such features in our samples and they apparently do not. In the locations in my current review referred to above I offer some possible reasons, somewhat more detailed in this version of my review, as to why they may not have seen these features. My explanations may not be correct, but there is some explanation and is not that we didn't see them, as the images below make clear! They speculate that the flow features could be due to the solidification of friction melts produced at asperity contacts, although they admit that the low average fault-surface temperatures make this unlikely. We have extensively studied our samples that show weakening due to another distinct weakening mechanism, namely flash melting or flash weakening at asperity contacts, at even higher slip rates and have never seen any such flow features in them (the total displacements are much smaller). Such localized melting does not create features like we images below. They suggest that they may be smears of amorphous material, and that may be closer to the truth, but clearly they are more than small smears – a mass of flowing material is produced in our experiments.

At one other place in their rebuttal they say “The single photo (which appears in Niemeijer et al. 2012, and same photo in Tullis, 2015) shows some sintered material. The latter is labeled ‘unslid novaculite’ in Tullis 2015, but does not

resemble the novaculite (shown in our Fig 2A).” Note however that their Figure 2 and our figure 2 below [and Figure 12 of (15)] are at totally different scales – the magnification on their image in their Figure 2A is much too low to see the details seen in the pit in our Figure – theirs has a scale bar of 1 mm and ours of 15 microns. They go on to say that the material that we label ‘unslid novaculite’ “does resemble our layers of particulate material: our Figure 2.” It is certainly possible that what we label ‘unslid novaculite in pit’ is actually some gouge material that has undergone some strain and underlies the smooth 3-5 micron-thick layer of silica gel rather than being totally undeformed virgin novaculite. Unfortunately it is not possible to determine a strain profile across the entire boundary separating the two novaculite blocks. Nevertheless, in all cases when we have been able to study a strain profile across a boundary separated by synthetic gouge containing marker materials (e.g. *Scruggs and Tullis, 1998*) what is always found is that nearly all the slip occurs on the shiny localized slip surfaces within the gouge and not within the surrounding gouge. The strain is all concentrated in the weaker material and based on everything we can determine in our experiments, that occurs by shearing within and slip upon the shiny silica gel layer imaged in Figures 1-5 below.

Scruggs, V. J., and Tullis, T.E., Correlation between velocity dependence of friction and strain localization in large displacement experiments on feldspar, muscovite and biotite gouge, *Tectonophysics*, 295, 15-40, 1998.

In response to my comments on the use of the term “hydrodynamic lubrication” referring to line 118 in their original paper, they argue that the use of the term is totally appropriate for what they intend to mean. I still strongly disagree. Hydrodynamic lubrication in the engineering literature is restricted to the situation where a converging space allows build up of pressure. In the case of elasto-hydrodynamic lubrication that converging space may develop due to elastic deformation, but normal hydrodynamic lubrication requires the same converging geometry and bulk hydrodynamic flow of fluid in the space, not simple shear of a thin liquid layer, especially one that may be only a few molecules thick. In my more detailed comments on lines 178-190 of the revised below I go into this in more detail. I did a number of web searches on hydrodynamic lubrication and invariably, as I said in my initial review, they included features that are not present in their proposed usage. I give a representative description based on looking at a range of web sites. Every one I looked at contained the same general details of what the term means. This term should not be used in this paper.

3) Relevant Images

Figure 1.

Figure 2.

Figure 3.

Figure 4.

Figure 5.

FRES - Forward Recoil Elastic Spectrometry for Hydrogen Detection

Figure 6.

μ_{ss} decreases with SiO_2 content above 50 wt. %

*Roig Silva
et al., 2004*

Figure 7.

Controlled Humidity Test in 1-atm Apparatus

Figure 8.

Controlled Humidity Tests

Water is needed for the weakening (we have so far been unable to reproduce these results, but we still believe them - it is very hard to get sample dry enough)

Titone et al. (2001)

Figure 9.

4) Specific review of the revised manuscript

Overall comments:

1. Can interparticle flow of amorphous material really cause the amount of weakening they observe? The other reviews question this as do I. As they note, we do have evidence of normal frictional strength in our Yund et al. paper (28) and the sample contained 40 to 50 percent of amorphous material near the localized slip surfaces. This is a large enough amount of such material that if it were weak, the sample would be. The authors infer that the modest temperature rise associated with the higher rates of sliding is enough to weaken this material and for it to cross above the glass transition. Both aspects of this are questionable, given that the effect of temperature on weakening and on the glass transition in such hydrous amorphous material is not independently known. Their explanation perhaps is not implausible, but it is presented as though it is known to be the case. I think it is much to speculative to say that the < 60 deg C elevations in T in their experiments are enough to change the mechanical properties of amorphous silica and to cross above the glass transition for this material. Again, it is possible, but very speculative, and should be presented at the most as a possible explanation that requires further investigation.
2. I still don't like the term hydrodynamic lubrication, after again having looked up its definition in the engineering literature. This mechanism refers to sliding that involves riding on a high pressure film of water that is pressurized by the dynamics of the shearing situation, and includes viscous effects of finite masses of water. This is not at all what is envisioned in Figure 6, which is a molecular level mechanism. It would totally confuse the literature if this comes to be termed by the geological community as hydrodynamic lubrication.
3. Their samples may or may not contain silica gel, but we certainly did as the attached images of Figure 2-5 show. Refer back to the middle paragraph concerning Reviewer 2's comments for what we mean by 'silica gel.' The flow features that we subsequently found in the attached images supported our inferences that the samples contain a flowing gel material. As discussed above, just because they didn't have it doesn't mean that we didn't. And it still seems possible that they might have had it on their really shiny localized surfaces within the gouge. We had to slide pretty far to get it smeared over our entire surface (Figures 1 and above).
4. Although the authors note the obvious comparison between their mechanical results and our earlier ones in references (5, 6) it is notable that the weakening they observe occurs with much less slip than does ours. Namely our experiments take about a meter of slip to weaken to their steady-state value whereas theirs seem to take only about 200 mm of slip. In some of our experiments (Figure 9 above) the weakening takes much more slip, up to 10

- or 20 m. It is not clear why ours show so much variability or whether the difference in this regard between our results and theirs is meaningful.
5. One solution would be to publish this paper as a set of interesting observations and remove all the interpretation part so that does not appear in the literature as an apparently proven additional weakening mechanism without being really understood. Note that I in no way dispute that they have amorphous silica in their experiments. We first showed this in reference 28 in 1990. Importantly, however, it was not associated with weakening even though up to 40-50% of the material was amorphous silica. In fact, however, the slip velocities were so low that the temperature cannot have risen more than a few degrees C at the most, so it is at least possible that the modest temperature rises from their experiments could affect the amorphous silica properties as they claim.

Details:

Line 11 Insert ref 5 and 6 along with 9 and 10, since in both papers we also show strengthening.

Line 11-12 They didn't find evidence for silica gel. Here at least they don't say we might be wrong about having it, although they do in their rebuttal!

Line 34-35 "friction decreased gradually" – not quite true – if they mean with slip then that is not the case for the 1 mm/s data, and if they mean gradually with slip velocity, then again it is not true for the 1 mm/s data. Might be better to remove the word "gradually" and add "although the 1 mm/s data don't fit the systematic pattern." This could be done by moving the phrase on line 36 to the beginning of the sentence and then adding this caveat just before the end of the reordered sentence. Honesty is always the best policy – it is better for the authors to point it out than to have it "discovered" by someone who looks at the figure carefully!

Line 38 This reference to a pseudo-analytical solution is confusing. I assume they simply use the truly analytical solution of Carslaw and Jaeger which is exact for a half space, a situation that they of course do not have, but it should be locally OK even in their radial symmetry small diameter sample. I suggest replacing "pseudo-analytical" with "half-space analytical"

Line 42-43 In looking at the temperatures in Figure 1B I believe there may be an error in labeling. The T's are described as being "Temperature Increase" but I think they must be "Temperature" with the boundary condition at the ends being perhaps 25 deg. Seem very unlikely to me that the temperature rise near the steel can be 25 deg. I also find their contour line designations poorly chosen. The yellow lines are hard to distinguish from each other, as are the red ones. Since this is all in the region of high temperature gradients, it makes it hard to know how hot it got as a function of distance from the

sliding surface, something that is important for their thermal interpretation for the cause of weakening.

Also the caption to Figure 1B should say that the results are from the FEM modeling (i.e., not the C&J solution). And perhaps replace in that caption the word “anomaly” with “distribution”

Another thing that needs to be specified is whether they took the time-varying experimentally-observed value of the friction coefficient as the input for the FEM model. I assume this to have been the case, since friction varies a lot in experiment 96. They should also say what they did in the case of the C&J analytical solution, since that assumes a time-invariant heat source. Maybe they somehow tried to account for that, but it's not simple to do so they need to say more if they somehow tried to do that.

Perhaps an even more interesting question is how the temperatures varied as a function especially of time but also position, say along a line perpendicular to the sliding surface taken at the radius of the effective velocity, at about 2/3 of the sample radius. They could show a series of profiles taken at different slip displacements for experiment 96. The point here is that since the input heat source (from the decaying friction) drops rapidly in the first 0.3 m of slip, it seems possible that the temperatures were actually interestingly higher during the first 0.3 m of slip than they are at the end of the experiment, which lasts 30 seconds, and is the time for which Figure 1B shows the final FEM results. This should be an easy output to get from the FEM calculations.

Line 51-52 These localized shear surfaces are pretty interesting, but are very poorly imaged in Figure 2B. It would certainly be nice to get a better image of them. I note that the face-on images of them from taking sample 97 apart has relatively low magnification SEM images of these surfaces. In the image files I have attached to this review, note that the flow features in what we believe to be silica gel are seen in much higher magnification SEM images. Furthermore we don't see them on the smooth shiny surface of the sample (that we attribute to being a smeared-out layer of silica gel), but only in the edges of the original pits (like those they show in their Figure 1A) where the gel has been dragged over the edge of the plateau and into the pits. The amount of slip that is required to create this thin layer of continuous (except for the pits) silica gel is tens of meters. Only two of their experiments (98 and 102 slid that far – Table 1). Although they studied the wear materials from those samples in detail (* in Table 1) they show no images of those samples in Figure 2. Were those samples taken apart for SEM imaging as was 97 or were they only studied in cross section as was experiment 101? They may have no silica gel in their samples, but it still seems that they could have! Amorphous silica in the presence of water becomes hydrated (it is used as a desiccant) and becomes a silica gel – its properties depend on how much water is in it.

Lines 58-62 This interpretation of their localized slip surfaces is the same as our interpretation of our entire smoothed surface. As described above, and shown in the attached images, one of which is included in reference 3, the gel flow features cannot be seen on the smooth surface itself, but only near the edges of the pits where it get smeared partially into a hole. Because their shiny localized slip surfaces are more ephemeral and within the gouge layer rather than covering the entire sample surface, they don't have the opportunity to see the flow features that we observe at high magnification in SEM.

Caption to figure 4 – Reference is made to HH, HV, VH, and VV in the caption, but in the figure HH and HV are not shown, but only the latter two. Presumably HH and HV should be removed from the caption. And to make a more clear connection to the supplement, I assume that these are the items within the parenthesis in the notation in the supplement, such as Z(HV)Z and Z(VV)Z. Some rewriting would make this more clear.

Lines 75-97 There is a rather long discussion of Raman spectroscopy results, but the bottom line of that seems to not end up contributing a lot to the paper, even though they did a bunch of work on it. I wonder if putting nearly all of this material in the Supplement and including only a summary like that in lines 95-97 in the body of the paper, but include lines 98-110. It's fine to leave it all in the paper, but it does make it longer and distracts perhaps a bit from the overall paper's impact. I found the anisotropy the most interesting part of this. If it is possible to say more about the nature of the preferred bond orientations from this and thus the nature of the strain in the amorphous silica, or something else besides the fact of anisotropy and the obvious "perhaps related to the shear-parallel smearing observed in the striated plates and on the slip surface" (lines 91-92), it would be interesting.

Line 111-121 They should say something about what the FT-IR spectra say about the quantity of water present, not just that "adsorbed water" was there. Often this is how FT-IR spectra are used – for quantitative analysis, not just existence of species. Given the debate about the role of water as combined with amorphous silica (i.e. is it "silica gel" or not) knowing how much water is there is very important. Presumably the amorphous nanosilica has a known quantity of water in it, so some quantitative comparisons could be made. I don't know enough about FT-IR spectroscopy to know what wavelength range is typically used to infer water content or whether the appropriate ones are shown in Figure 4B. Something should be said about this issue. I note in their response to my earlier comment on lines 88-94, that they say they are unable to determine the quantity of water in the sample from the FT-IR measurements, due to "uncertainties in path length due to thickness and packing variations." This is totally reasonable response to my question – some explicit statement to this effect should be made in the paper to quell the sort of question that I was asking in this paragraph (I'd forgotten their response to my earlier comment).

Lines 113-115 This sentence says the wear material differs from the commercial amorphous nanosilica. These differences are not described in the text, but only in the Supplement. That location should be referred, and if they feel some part of that is particularly interesting it should be specifically mentioned in the text here.

Supplement final paragraph – the peak at ~694 is twice referred to as being at ~695 – change that to 694 for clarity as is labeled in Figure 4B.

Lines 119-121. This sentence may indeed be true, but the combination of FT-IR and Raman spectra by no means proves this postulated configuration to be the case. As far as I can see it only shows that the water content is below the Raman detection limits, not where it resides. It is important to state the Raman detection limits.

Table 2. It is not clear what the * for some table entries refer to.

Line 123-125 The “hydrated surfaces” part of this is important in their argument, but all that I see that addresses this is the label of “3360 adsorbed H₂O” within the inset of the top half of Figure 4B and perhaps the “3660 silanol” in the same inset and the “976 silanol bending” of the larger lower inset in Fig. 4B. Typically silanol groups (Si-O-H) are found on the surfaces of quartz, e.g. However, silanol can exist in the interior of the gel gobules, not just on the surface and the FT-IR seem not to be distinguishable. I’m no expert, but a quick web search turned up this reference, for example, that seems to me to call into question the uniqueness of your interpretation of your FR-IR spectra. This needs more discussion.

Davydov, V.Y., Kiselev, A.V., and Zhuravl, L.T., 1964, Study of the Surface and Bulk Hydroxyl Groups of Silica by Infra-red Spectra and D₂O-exchange, Transactions of the Faraday Society, 60, 2254-2264.

Line 124-138 In the case of study 28, as far as we were able to ascertain, a layer of quartz gouge as close to the localized slip surface as we could get was comprised of up to 40-50% amorphous silica. While this is clearly not a continuous layer of 100% amorphous material, it should be significantly weaker than pure comminuted crystalline quartz if amorphous silica is weaker at room temperature and low slip rates. However, it was not. This contributed to our conclusion that the thixotropic properties of silica gel, not the mere presence of amorphous silica, contributes to the high speed weakening.

Lines 139-146 I totally agree with this conclusion, The explanation for it is what is in question.

Lines 146-148 Here is where we disagree. Based on our experiments, we believe that the presence of water within the amorphous silica created a silica gel that

has flowed at high strain rates as a thixotropic material. So we both call upon water to contribute to the material being weak, but we believe the material has undergone more continuous flow when it was deforming rapidly, where as they call upon particulate flow. In some sense this may be a matter of scale, since the flow of a silica gel must involve the relative motion of units within the gel. We are agnostic as to the nature of these units since we have no data that bears on it. They could be as small as Si-O₄ tetrahedra or as large as hydrated SiO₂ globules that would be small enough, perhaps 50 nm (?), that they would be indistinguishable in our SEM images in which the surfaces of the flowing gel look perfectly smooth.

Lines 163-168 The qualitative role of water and temperature alluded to here is undoubtedly correct. What is unclear is whether the unknown water contents and modes temperature rises involved in these experiments are sufficient to cause crossings of the glass transition (their references (35, 36) refer to organic materials so offer no quantitative help here.

Lines 178-190 I strenuously object to the use of the term hydrodynamic lubrication here and everywhere else in the paper. They are using a well-established term in the lexicon of machinery to refer to something totally different. If this term becomes accepted with this new meaning in the geologic literature we will have done a great disservice to clear communication. The accepted meaning of hydrodynamic lubrication applies to the situation where: 1) A lubricant layer, which must be a viscous fluid, separates the surfaces; 2) The fluid undergoes hydrodynamic flow in the space between the moving surfaces; and 3) the surfaces between which the fluid films move must be convergent. It is this convergence, combined with the hydrodynamic flow of the viscous fluid film, that builds up the hydrodynamic pressure within the fluid that holds the surfaces apart. In my previous review I objected to their use of this term, but muddied the waters by bringing up elastohydrodynamic lubrication which is also irrelevant here, but is not the term they are using. I note that reviewer number 2 also objected to the authors' use of the term hydrodynamic lubrication. That objection was more based on a question as to whether the water inferred to be at the particle boundaries acts as a lubricant. I share that concern somewhat, although they cite many references supporting the idea and it is probably correct. However, I still strongly object to using the term because it has another accepted meaning. There is not a thick enough layer of water there to cause the kind of hydrodynamic flow that this terminology implies, it is not thick enough to act as a viscous fluid, and the converging geometry required does not clearly exist. Some other terminology should be used.

Figure 4. Reviewer 2 likes this. It indeed makes clear what they have in mind as a model, and therefore has some value. What seems most speculative to me is what is described in the "A" part of the drawing. It is not clear to me that there is enough warming to lead to plastic flow, since the temperature rises are not large, but it is certainly possible. I also think it is likely that water in the bulk of the

particles is likely to contribute to their ability to flow, i.e. it is silica gel. I have proof only of the bulk flow of the material from our SEM photos, not of the details of why it happens. However, it is clearly thixotropic since the textures show that it flows at high rates and is weak whereas it fractures and is strong when the rapid slip is over.

Lines 198-199 I do not believe that they have shown this. They have no evidence that silica gel is present, but they have none that it is not (this is of course hard to do). Our evidence that it is present, is primarily the flow features that clearly are seen in our samples as is shown in some of the figures that I have attached to this review (only one of which is published), and that the surface layer of our samples has approximately 2 percent water based on FRES analysis for hydrogen in this thin layer. Although they have done extensive, careful, and interesting imaging of their samples, as I discuss earlier, due to the low slip amounts of the samples they used to look at the sliding surfaces (at least as shown in the provided figures), the relatively low magnification of their submitted SEM images compared to ours, and the lack of favorable geometry to see flow features at the edge of their slip surface, they could well have missed the small but significant flow features contained in our sample that show flow occurred in our surface layer.

Line 199-201 I more or less agree with this sentence, but would be in perfect agreement if it says "material" rather than "particles." Again the difference may be more in terminology than in substance. We chose to call this "silica gel" because it clearly flowed, contained amorphous silica and water, and behave thixotropically. This last property, thixotropy, we attribute to a competition between time-dependent healing and strain dependent disruption of bonds in the material. We chose not to discuss the possible temperature dependence of this process, because we didn't think that the shear-heating temperature increase was enough to have a significant effect, but no one seems to have quantitative knowledge concerning that. Clearly it is above absolute zero, so time-dependent healing should occur.

Line 207-208 Maybe "above the glass transition," but as this T is not known, this remains speculative. If the behavior being weak can be taken as a proof of this, then I guess it may be true, but I'd rather see an independent determination that such a known temperature was crossed.

Lines 227-233 I think we all agree that "silica lubrication" may be important in earthquakes, although using this term is a bit vague. I still like "silica gel" better because it is more specific, but it may or may not apply to their results. "Silica lubrication" is a bit problematical as a term because people will immediately say, "how can quartz cause lubrication," since silica means SiO₂. Also, one has to be careful not to end up getting confused with lubrication via the use of silicone oil.

The suggestion that gel may not have been necessary for Tullis et al experiments was raised in the rebuttal to Reviewer 3 (Tullis)'s previous review (NOT in our manuscript, in any draft). This was incorrectly described by R1 in his recommendation, which was supported by Editor Plail, that we correct this statement in the manuscript. Thus there is no statement to remove from the manuscript, but to address the spirit of this comment we have included the statement in the discussion (Lines 205-209):

“Both the previous experiments (Goldsby and Tullis, 2002; Di Toro et al., 2004) and our experiments produced nanopowders (Tullis and Goldsby, 2006) and display similar
Page 3 of 25

weakening behaviors, but our experiments did not produce gel, perhaps due to different conditions. We therefore prefer the interpretation that the wear powder is likely responsible for the weakening behavior in both sets of experiments. “

I hope this will satisfy R3 and the editors that we do not deny the production of silica gel in the Tullis group experiments, but we believe that a comparison to our results show that gel is not required to produce the weakening effects. Further investigations are obviously necessary to understand the relationship between the wear powder and the gel and determine which experimental conditions might have caused the difference in outcome of the two sets of experiments. This is further addressed in our replies to R3 comments below.

Reviewer #1 (Remarks to the Author):

2nd review of Rowe et al. “Earthquake lubrication and healing explained by amorphous silica”

The main concern raised by the second review of prof. Terry Tullis is the fact that no “silica gel” (H₂O rich silica exhibiting fluid-like microstructure) was observed in the experiments reported by Rowe and co-authors and the fact that Rowe et al. suggest in the current manuscript that no silica gel was present in the earlier experiments of the Brown group. The second review provides ample details and pictures of the material observed in previous experiments conducted at Brown. I am convinced that the material shown in the reviewer attachment is indeed some kind of amorphous silica gel as proposed in earlier papers, and that this material is distinct from the material observed by Rowe and co-authors. Unfortunately, these images were never published in peer-reviewed journals so the authors of the current manuscript lacked this important information when writing their paper and replying to the review. This episode highlights the problem with very short papers where not enough space for full data presentation is available in my point of view. The two possible explanations offered by Terry Tullis, i.e. 1) either large difference in SEM magnification used to document the amorphous material in the reviewer attachment; or 2) one order of magnitude lower slip distances in the experiments of Rowe et al. compared to previous experiments are both credible. Rowe and co-authors disclose in the rebuttal that many hours were spent at the SEM looking for silica-gel like microstructures but none were found. I assume that the published pictures are only a small subset of all images produced and analyzed and therefore I tend to favor the latter explanation that not enough slip was induced in the samples to form silica gel – this might be the easiest explanation for the discrepancy. Previous experiments on granite gouges sheared to large strains under low stresses (Yund et al. 1990) as well as sheared to modest strains under high stresses (Pec et al. 2012, 2016) indicate that the total amount of work (i.e. force x displacement) is what ultimately matters for the production of very fine-grained, even amorphous materials in my opinion.

The revised manuscript of Rowe et al. is of high quality, the findings are well documented, timely and very significant so I suggest publication with minor revisions and integration of the comments from the second review of Terry Tullis. The concerns from previous reviews were well addressed in the rebuttal and updated manuscript. I think that the issue of silica gel presence & absence will be best solved if the authors acknowledge its existence in previous experiments and discuss the differences in the experimental set-up and total slip experienced by the samples as a possible solution to the mystery. One further and poorly-

traceable difference in the experiments could be due to different room humidity conditions in Providence and Padova given that the gel is clearly hydrated and doesn't form (not reproducibly apparently...) in very dry rocks. Future experiments should aim at controlling or at least monitoring humidity during the experiments. More work will be certainly needed to fully resolve this discrepancy and I hope that both groups will continue working on this fascinating subject.

Addressed in reply to editor's comments above.

Minor point:

Line 130 (...≈15-25 MPa... should be GPa)

Corrected, thank you.

References:

Yund, R. A., Blanpied, M. L., Tullis, T. E., & Weeks, J. D. (1990). Amorphous material in high strain experimental fault gouges. *J. Geophys. Res*, 95, 15589–15602

Pec, M., Stünitz, H., Heilbronner, R., Drury, M., & de Capitani, C. (2012). Origin of pseudotachylites in slow creep experiments. *Earth and Planetary Science Letters*, 355–356(0), 299–310. <https://doi.org/10.1016/j.epsl.2012.09.004>

Pec, M., Stünitz, H., Heilbronner, R., & Drury, M. (2016). Semi-brittle flow of granitoid fault rocks in experiments. *Journal of Geophysical Research: Solid Earth*, 121(3), 1677–1705. <https://doi.org/10.1002/2015JB012513>

Reviewer #2 (Remarks to the Author):

The 2nd round review:

The authors have put substantial efforts into the revision and have provided detailed responses to my and other reviewers' comments. In some sense, the paper was renovated; now it reads much better, and the logic makes more sense to me. I appreciate the authors' attitude on re-visiting the previous work published on high-impact journals and providing new interpretations based on detailed observations using advanced techniques. The paper is of great interests to experimental and field-geology communities. This work would be suitable for publication in *Nature Communications*.

I have one major concern on the proposed weakening mechanism(s):

First, for any granular (particulate) material, all possible weakening mechanisms play their roles at two scales: intergranular interactions and intragranular plasticity. Therefore the proposed weakening mechanisms (particulate flow and intraparticle plasticity) cannot be wrong, although I do not think it is appropriate to refer to Ashby and Verrall (1973) to illustrate the combination of these two mechanisms (the added sentence 148-150). In the theory proposed by Ashby and Verrall, grain boundary sliding (gbs) is mostly an artificial effect (deformation is accommodated by grain boundary diffusion).

In citing Ashby and Verrall, we used the construction (c.f.) which means “compare to”. We did not mean to suggest an equivalency with their mechanism, we are using this reference to illustrate the contrast between the deformation of amorphous materials vs. past work on

crystalline materials. We have been unable to find any relevant literature on the equivalent mechanisms accommodating grain boundary sliding in amorphous nanopowders.

Second, the revised manuscript added the temperature calculations (Figure 1). I agree that “hydration effects reduce the glass transition temperature and further promote plastic flow”. However, by looking at the manuscript (e.g. Figure 6) and the response letter, the authors tend to believe that rapid weakening is caused by frictional heat (temperature rise) at local asperities, which one would easily think to be “flash heating”. Here my point is that the temperature calculations do not straightforwardly support the interpretation. To go beyond this point, my suggestion is to investigate the flash weakening temperatures of the amorphous nanopowder, although I understand the calculation as such would be suffer from uncertainties.

We have calculated an upper estimate of the peak temperature during asperity flash heating based on the peak friction measured during our experiments and indenter strength of quartz. Even in the fastest slip rate experiments the upper bound on asperity temperature rise is $\sim 1000^{\circ}\text{C}$, significantly below the temperature required to melt quartz. We have no lower bound on temperature as the true properties of the wear material are unknown. The results of these calculations have been added to the manuscript (Table 1) and the details are now in the supplementary material.

Third, I found the statement of thermally-activated mechanism is confusing (e.g. abstract). Yes, intraparticle plasticity is a thermal activated mechanism. Is not particulate flow (intergranular flow)?

Particulate flow (or particle switching) does not specifically require thermal activation, although some relevant mechanisms which facilitate particulate flow might be more efficient at higher temperatures. The intraparticle plasticity is clearly thermally-driven. Therefore we have not expanded on potential second or third-order effects of temperature on particulate flow.

Forth, the authors argued that “intraparticle plasticity” is attributed to the activity of grain boundary sliding, although the involved processes might be different from that for crystalline nanoparticles (Lines 169-176). Is “grain boundary sliding” not a specific type of particulate flow? To my understanding, particulate flow (intergranular flow) involves a series of processes including grain boundary sliding, grain rolling, grain wearing and so on.

The reviewer misquotes our manuscript, in fact we wrote that ...”weakening in crystalline nanoparticles has been attributed to the activity of grain boundary sliding...”. To avoid future misunderstanding, we have changed “crystalline nanoparticles” to “crystalline nanomaterials”.

My second concern is about the abstract:

The abstract reads appealing; but after reading through the manuscript for several times, I found the application to fault healing is not thoroughly discussed. It is not clear how these two weakening mechanisms correspond to different time-scales of fault healing either. Typo in the abstract: “10s-100s years”

The application to fault healing is discussed in two areas of the discussion (short timescales: siloxane bonding, 100s timescale, lines 192-204; and long timescales: crystallization of quartz, lines 238-244. The timescales of the two healing mechanisms correspond to laboratory healing and natural fault healing timescales. "10s-100s years" is not a typo; it refers to the natural fault case of quartz crystallization.

Review of Revised Rowe et al Paper by Terry Tullis

Note this review is not nearly as succinct or as organized as I would have liked. Some of the points are made in multiple places as they arose in different ways. If I spent a lot more time on it I could consolidate the similar comments so there would be less reference to saying something similar elsewhere. However, I've already spent too much time on it and the editors understandably are wanting my response!

This review contains several components:

- . 1) Comments prompted by the reviews of the other two reviewers and the authors' rebuttals
- . 2) Responses to the authors' rebuttals of my original review
- . 3) Images from a PDF file that was created from a PPT file that is pertinent to both item 1 above and 2 below
- . 4) Specific review of the revised manuscript, both general comments and specific line-by-line ones.

1) Comments prompted by the reviews of the other two reviewers Reviewer

1 Comment and reply concerning lines 44-51. It is interesting that our experiments and those described in this paper differ in that our samples produced a continuous shiny slip surface separating the two sample halves, as the image in Figure 1 below shows. I assume the difference to be the total amount of slip. Note that the slip for the Figure 1 sample was 62 meters, whereas for most of theirs was 3 meters and the largest two, that they do not image in the paper, was 30 m.

As our work is focused on the weakening mechanism, only the short displacement experiments, which showed substantial weakening, were illustrated in the paper. Any effects or materials that formed only in the longer slip experiments are not required to explain the weakening effects.

Reviewer 2

I agree with the general comment that the "discussion contains a lot of speculation." In general, in my opinion, the proposed processes for weakening are presented in a much more definite way than is appropriate for what is known to be true. I discuss this in more detail in many places below. Relative to thermally induced weakening (item 1) contributing to what they now call 'particulate flow,' I think it is pretty speculative to say that the modest temperature rises that they have now calculated are sufficient to cause a transition across

the glass transition. What is required for that to occur in this amorphous material is presumably not known, and it is certainly possible, However, it remains speculative and the statements made are much more definitive than is warranted. Regarding “hydrodynamic lubrication,” (item 2), I discuss this in more detail below, In contrast to this reviewer, it does seem plausible to me that some H₂O molecules along particle boundaries could well be responsible for a reduced resistance to interparticle sliding. It remains speculative, but seem plausible to me. My biggest objection to this component of their explanation for the weakness of their samples is the use of the term ‘hydrodynamic lubrication’ to describe it. More on that below.

See comments below.

In the rebuttal to item 2, the authors say that “Subsequent dry experiments did show the weakening effect of reasons not yet understood.” We think they were not dry enough, as is noted briefly on Figure 9 below. I discuss this more fully in a paragraph where I comment on their rebuttal of my original comments (search for the word ‘dry’ – the paragraph contains it numerous times).

It is interesting that the authors response to Reviewer 2’s last general comment is that “silica gels take myriad forms” even though the reviewer had not used the term ‘gel.’ In our earlier work we did not try to go into any specific description of exactly what we meant by silica gel for exactly the reason stated by the authors. Our intent was that it meant non-crystalline silica, with some unknown quantity of water included somehow within it, that was able to flow at high shear rates and be brittle at lower rates, namely that it was thixotropic. Our qualitative explanation for this was that its alternate behaviors involved a competition between a strain-dependent tendency to depolymerize bonds and weaken and a time-dependent tendency for bonds to strengthen and polymerization to occur. We referred to publications that showed silica gels to be thixotropic. So we tried to hit a happy medium between qualitative explanations that made sense without trying to go into unwarranted speculations about the details of the processes occurring during deformation in the weak state. As far as we were concerned, the flow could occur on a molecular scale and involve relative motion all the way down to the level of silica tetrahedra, or it could involve motion on a larger scale between globules or particles of silica gel that often exist. The observed flow features in the highest magnification images in Figures 2-5 (i.e. Figure 5 suggest that the scale of the flowing units may be pretty small).

We initiated this work in an effort to add detail to the original (speculative) identification of gel, because this nontechnical definition and lack of descriptive data prevented comparisons to natural faults. The activity of a proposed deformation mechanism produced in the laboratory cannot be verified in nature without criteria for identification. The Tullis group did not specify any criteria for the identification of gel beyond the thixotropic behavior. ‘Gel’ has a scientific definition, which is more specific than the general description offered here, and this distinction was not made in the Tullis group articles. Not all thixotropic materials are gels. We have now included the definition of gel in the manuscript (Lines 12-13).

The response to Reviewer 2’s specific comment that refers to line 144 suggests that either lower slip rates or high normal stress could also produce weakening. The author’s rebuttal is only that they did not perform experiments at other normal tresses. But they did perform them at lower slip rates. The result in terms of weakening was that some was observed at 0.1, 0.01, and 0.001 times lower than the their “standard” rate of 0.1 m/s. However it is

interesting that the weakening does not become progressively lower as the velocity decreases – the strength at 1 mm/s is an outlier, being lower than at both higher and lower speeds, rather than being intermediate. For unknown reasons friction data is notorious for not being as reproducible as one would expect, so I don't find this particularly problematical, but it does weaken the story a bit. The bottom line on this is that there is not significant weakening at lower velocities (10 cm/s and 10 microns/s) but of course how much one would expect isn't clear! So while I agree with the expectations of the second reviewer concerning the expected effect of velocity on the weakening if it is thermally induced, I think the results are ambiguous.

Our data (Fig 1A) show that at the slower velocities tested, the coefficient of friction weakened from ~0.75 to ~0.55. This is well within the range of uncertainty and a significant result.

2) Responses to the authors' rebuttals of my original review

At the end of one paragraph in their rebuttal the authors say:

“Based on the evidence reported here, our position is that there was no gel produced in our experiments and, perhaps, was not any gel in the previously published experiments either.” This is at the heart of my objection to their position and hopefully the attached images and my suggestions as to why they could have missed the evidence for flow, will change their minds.

Clearly the difference is in the explanation for the weakening we both observe. At the moment I cannot conclusively explain why Rowe et al. have not seen the textural features that we did, although I offer some likely suggestions later in this review. Note how large our displacements were (10-20 m) to get the sufficiently smooth surface for the gel on it to act as a lubricating layer. This is in contrast to the much more rapid weakening with slip shown by Rowe et al. in Figure 1 of their ms.

Unfortunately we did not publish SEM images or the chemical analysis we made of the smooth sliding surface at the time we submitted our published papers. I think we did not have had those results by then, I'm not sure, but we got them later and they confirmed our original interpretation of silica gel.

The chemical analyses showing the presence of water bonded to the silica are equally compatible with the Tullis interpretation of silica gel and the Rowe interpretation of hydrated surfaces on amorphous silica particles. Our FTIR and Raman experiments essentially cover the same ground. The gel hypothesis makes no quantitative predictions about water content. Therefore these data don't clearly distinguish between the two hypotheses.

Publishing just the images and the chemical analysis for water seemed too little to make a publication, so we never submitted it anywhere. In this case it clearly is a problem. I did include one SEM image in my review paper (15) that the authors note in their response, but notably in this discussion they only refer to what we label “unslid novaculite in pit” and make no comment on what we considered to be most significant features in the image [which is Figure 2 below and also Figure 12 in ref (15)] to the clearly marked flow features

("flow features; silica gel"). I may or may not have chosen the best single image, to include in (15) but as can be seen from the attached, there are more of them in a similar vein.

The number of images collected is not at issue, we feel the important thing is to determine what micro/nano structures are broadly *characteristic* of the sliding surface and therefore likely to control the macroscopic behavior of the entire sliding surface. That is why we studied the entire surface of the slid core in the SEM and photographed continuous transects across the entire diameter (not presented in the manuscript as they are mostly the same and pretty boring, but these images can be produced if required). We can categorically say that the features labeled "unslid novaculite in pit" by Tullis et al. does not resemble our unslid novaculite (Figure X below and manuscript Fig 2A) but it does resemble the appearance of layers of wear powder produced in our experiments (Fig Y of this letter). At these magnifications we would have clearly identified the flowy structures observed by Tullis et al. if they were present in our sample. Tullis et al. have never reported, in publications nor in this lengthy exchange of review and reply, whether the 'flowy' images are characteristic of the entire sample surface, nor have explained why they removed the particulate wear material before beginning characterization. The particulate wear material was reported in the USGS grant reports but not reported in the peer-reviewed papers. In our view this is an important omission.

Figure X: Unslid novaculite reference sample at higher magnification than in Figure 2A of the manuscript.

Figure Y: Wear powder particles and clumps (~1-30 microns)

Figure Z: Shiny sliding surface overlying wear powder, x430

The images below not only show more instances of the flow features in higher magnification SEM near the edges of the pits than the SEM images of Figure 2 of the

submitted ms, they also include some images related to other items in the same long section of their rebuttal. One slide shows the determination of the H content in that thin surface layer done by FRES. It shows that the water content of the surface layer has a water content of around 2%. This technique has the advantage that it can analyze a surface layer, does not need such thick sections as does IR, and is more sensitive than Raman.

Our Raman and FTIR analyses were conducted in situ and on wear powder independent of the sliding surface. Again, it appears Tullis et al. cleaned the powder from the surface before doing the FRES analyses (although this is not explicitly stated in R3's comments)? Thus, it is difficult to make direct comparisons. Regardless, both groups identified water bonded to silica on the slid surfaces so again, this identification does not differentiate between our interpretations.

So I still believe we had weakening by silica gel. Note from one slide that it doesn't occur if there is less than 50% silica in a sample, something that presumably would be found by the authors if they studied other rocks.

We show that this wear material can form from quartz. Quartz is a dominant crustal mineral, and fault zones are preferentially mineralized with quartz veins in many rock types. We do not extend any interpretations to other rocks.

I also included the results the authors refer to in their rebuttal that show how our undergrads managed to turn the weakening on and off by alternately wetting and drying the samples. As the authors noted in their rebuttal, this is only in an abstract and the figure is not in the literature. I do not see how our students could have done anything wrong to produce these results! It is notoriously hard to remove trace amounts of water from samples and when we tried to replicate the right amount of drying we couldn't – I think that is because we didn't hit a magic window of the right amount of heating and simultaneous dry atmosphere when we did nominally the same thing they did with a heat gun. We either didn't get it hot enough and it didn't get dry enough, or it got too hot and the epoxy holding the samples to the steel grips debonded. We lost patience with trying, but it would be good to go back and try again to do!

Our community (the earth sciences) does not have as strict standards for reproducibility as some other fields, and we recognize that these experiments are very difficult to complete successfully so that replication is rare. However, repeated failure to replicate, especially when that failure is as of yet inexplicable, relegates the initial result to anecdotal. It may very well be right, but is not reasonable to expect other scientists to treat result as if it has been proven. The burden of proof is on the claimant.

As you can see, I still believe we had flowing silica gel in the surface layer of our samples and that it is responsible for our observed high-displacement weakening. Hopefully these images will convince the authors that we had such flowing material in our samples, even though Figure 12 of (15) [Figure 2 below] seems not to have done so! They say at the end of their long response to my comment 3 "We found other types of compounds made by and water, but not silica gels." Elsewhere in this review (detailed comments relative to Lines 51-52, 58-62, and 198-199) I give possible explanations as to why they might have missed such flow features in their careful imaging. In some sense this discussion may be simply semantic, since we both believe we have an abundance of amorphous material with a non-negligible water content. One of the main reasons we still prefer the term silica

gel, is that it is descriptive and the word gel implies a sort of continuous flowing plasticity (with the shifting parts moving relative to one another at some scale smaller than the observations). In fact it is not clear to me that if they were to do TEM analysis of something that we would call flowing silica gel that it would differ too much from that they image Figure 3. Still, I suspect our material would have less fragmented- and particulate-appearing material, perhaps because it has undergone much higher strain. It would be interesting to do TEM on our gel.

“Gel” has a specific definition. (From Encyclopedia Britannica: *Gels are colloids (aggregates of fine particles, ... dispersed in a continuous medium) in which the liquid medium has become viscous enough to behave more or less as a solid.*) Prof. Tullis’ definition seemingly describes continuous deformation but does not relate to a particular structure of matter.

Gels have solid particles or molecules (colloids) suspended in liquid. Gels are distinguished from sols due to the either chemical or physical interaction between the particles forming a structure which lends the gel its macroscopic properties including a non-zero yield strength, and in the case of some (but not all) gels, thixotropy.

We do not use the term ‘gel’ for our wear material because we don’t see any evidence of an interconnected liquid in which the particles dispersed (and no evidence of syneresis which would reveal the disappearance or evaporation of such a liquid). If the Tullis et al. experiments produced gel, what composed the dispersed particles and the liquid medium that held them? The ‘flowy textures’ appear to be homogeneous hydrous silica (probably amorphous although no evidence has been given to support this). Given the longer displacement of the Tullis et al. experiments, perhaps the flowy material is composed of melted or smeared amorphous silica particles? Or perhaps given the additional frictional work in the experiments, a gel was eventually formed with nanoparticles as the colloids? Even with the additional observations provided by Prof. Tullis in his review there is not enough evidence available to classify the flowy material or link it to the definition of ‘gel’.

With reference to the question of whether they had flow-like features in their samples and their arguments that they did not, they give a description of their efforts to find it in their long response to my comments on lines 50-51. All I can say is that in spite of all of this, it is obvious that we have such features in our samples and they apparently do not. In the locations in my current review referred to above I offer some possible reasons, somewhat more detailed in this version of my review, as to why they may not have seen these features. My explanations may not be correct, but there is some explanation and is not that we didn’t see them, as the images below make clear!

Tullis and collaborators have clearly demonstrated the flowy features, on their experimental samples, but these observations alone are sufficient to identify the material as silica gel, nor to tie it exclusively to the frictional weakening.

They speculate that the flow features could be due to the solidification of friction melts produced at asperity contacts, although they admit that the low average fault-surface temperatures make this unlikely. We have extensively studied our samples that show weakening due to another distinct weakening mechanism, namely flash melting or flash weakening at asperity contacts, at even higher slip rates and have never seen any such flow features in them (the total displacements are much smaller). Such localized melting does not create features like we images below. The suggest that they may be smears of

amorphous material, and that may be closer to the truth, but clearly they are more than small smears – a mass of flowing material is produced in our experiments.

Even with the newly provided documentation, Tullis and collaborators have not stated whether flowy material covers the entire sample surfaces or how its presence or abundance differed between experiments where weakening was observed or where it was not observed.

At one other place in their rebuttal they say “The single photo (which appears in Niemeijer et al. 2012, and same photo in Tullis, 2015) shows some sintered material. The latter is labeled ‘unslid novaculite’ in Tullis 2015, but does not resemble the novaculite (shown in our Fig 2A).” Note however that their Figure 2 and our figure 2 below [and Figure 12 of (15)] are at totally different scales – the magnification on their image in their Figure 2A is much too low to see the details seen in the pit in our Figure – theirs has a scale bar of 1 mm and ours of 15 microns.

In our manuscript, we presented images of relevant features at the scale appropriate to show the essential characteristics of the features. Above, we present an image of the novaculite at much higher magnification (Figure X) which is unfortunately not of very high quality but it is sufficient to demonstrate that the primary crystalline structure with pores is essentially the same at low magnifications and high magnifications, and does not have powder fragments visible on the surface or in pores as in the Tullis example labeled ‘unslid novaculite’.

They go on to say that the material that we label ‘unslid novaculite’ “does resemble our layers of particulate material: our Figure 2.” It is certainly possible that what we label ‘unslid novaculite in pit’ is actually some gouge material that has undergone some strain and underlies the smooth 3-5 micron-thick layer of silica gel rather than being totally undeformed virgin novaculite.

We concur.

Unfortunately it is not possible to determine a strain profile across the entire boundary separating the two novaculite blocks. Nevertheless, in all cases when we have been able to study a strain profile across a boundary separated by synthetic gouge containing marker materials (e.g. *Scruggs and Tullis, 1998*) what is always found is that nearly all the slip occurs on the shiny localized slip surfaces within the gouge and not within the surrounding gouge. The strain is all concentrated in the weaker material and based on everything we can determine in our experiments, that occurs by shearing within and slip upon the shiny silica gel layer imaged in Figures 1-5 below.

R3 implies localization by comparison to other experiments. We directly show evidence of localization in cross-section of these experiments (Fig 2B of the manuscript) and this localization is essential to our detailed description and explanation of the formation of shiny plates (Figs 2E and 2F).

Scruggs, V. J., and Tullis, T.E., Correlation between velocity dependence of friction and strain localization in large displacement experiments on feldspar, muscovite and biotite gouge, *Tectonophysics*, 295, 15-40, 1998.

In response to my comments on the use of the term “hydrodynamic lubrication” referring

to line 118 in their original paper, they argue that the use of the term is totally appropriate for what they intend to mean. I still strongly disagree. Hydrodynamic lubrication in the engineering literature is restricted to the situation where a converging space allows build up of pressure. In the case of elasto-hydrodynamic lubrication that converging space may develop due to elastic deformation, but normal hydrodynamic lubrication requires the same converging geometry and bulk hydrodynamic flow of fluid in the space, not simple shear of a thin liquid layer, especially one that may be only a few molecules thick. In my more detailed comments on lines 178-190 of the revised below I go into this in more detail. I did a number of web searches on hydrodynamic lubrication and invariably, as I said in my initial review, they included features that are not present in their proposed usage. I give a representative description based on looking at a range of web sites. Every one I looked at contained the same general details of what the term means. This term should not be used in this paper.

We address this issue below where R3 presents his detailed comments.

4) Specific review of the revised manuscript

Overall comments:

1. Can interparticle flow of amorphous material really cause the amount of weakening they observe? The other reviews question this as do I. As they note, we do have evidence of normal frictional strength in our Yund et al. paper (28) and the sample contained 40 to 50 percent of amorphous material near the localized slip surfaces. This is a large enough amount of such material that if it were weak, the sample would be. The authors infer that the modest temperature rise associated with the higher rates of sliding is enough to weaken this material and for it to cross above the glass transition. Both aspects of this are questionable, given that the effect of temperature on weakening and on the glass transition in such hydrous amorphous material is not independently known. Their explanation perhaps is not implausible, but it is presented as though it is known to be the case. I think it is much to speculative to say that the < 60 deg C elevations in T in their experiments are enough to change the mechanical properties of amorphous silica and to cross above the glass transition for this material. Again, it is possible, but very speculative, and should be presented at the most as a possible explanation that requires further investigation.

The Yund et al. experiments were not performed at fast velocity. It is not clear whether the amorphisation process is similar under those conditions to the faster experiments of Goldsby and Tullis or Di Toro et al. Nothing is known about the glass transition of the amorphous hydrous silica, so any discussion of conditions of glass transition is necessarily speculative. However, given our observation of sintering and smearing of particles, it is clear that coherent, distributed deformation affected the particles nearest the slipping surface. Whether the work applied to the amorphous hydrous silica was thermal or mechanical, we are confident that the material has temporarily crossed the glass transition. We propose temperature variability on the slip surface (which is well known phenomena in all frictional sliding conditions) as a potential explanation. We also added a new reference to the manuscript (Deb et al. 2001) which explains how high pressures at asperity contacts could also drive the material across the glass transition.

2. I still don't like the term hydrodynamic lubrication, after again having looked up its definition in the engineering literature. This mechanism refers to sliding that involves riding on a high pressure film of water that is pressurized by the dynamics of the shearing situation, and includes viscous effects of finite masses of water. This is not at all what is envisioned in Figure 6, which is a molecular level mechanism. It would totally confuse the literature if this comes to be termed by the geological community as hydrodynamic lubrication.

Figure 6 shows a layer of water trapped in between silica-bonded OH groups (silanol). Macroscopic descriptions may not address the role of charge and chemistry in trapping water layers, but the water layers between any set of particles are certainly affected by these phenomena. Prof Tullis has not mentioned any specific references that apparently conflict with our understanding of 'hydrodynamic lubrication' so we are unable to follow his argument further.

3. Their samples may or may not contain silica gel, but we certainly did as the attached images of Figure 2-5 show. Refer back to the middle paragraph concerning Reviewer 2's comments for what we mean by 'silica gel.' The flow features that we subsequently found in the attached images supported our inferences that the samples contain a flowing gel material. As discussed above, just because they didn't have it doesn't mean that we didn't. And it still seems possible that they might have had it on their really shiny localized surfaces within the gouge. We had to slide pretty far to get it smeared over our entire surface (Figures 1 and above).

Shiny surfaces (mirror faults) are any surface that is smooth on the scale of visible light reflection (amplitudes of roughness < about 100 nm over lateral scales of about 550 nm, Simon-Tov et al., 2013, *Geology* v. 41, 703-706). Fault mirrors are produced in a variety of lithologies and by a variety of mechanisms. The shiny surface could represent many other processes other than gel formation.

4. Although the authors note the obvious comparison between their mechanical results and our earlier ones in references (5, 6) it is notable that the weakening they observe occurs with much less slip than does ours. Namely our experiments take about a meter of slip to weaken to their steady-state value whereas theirs seem to take only about 200 mm of slip. In some of our experiments (Figure 9 above) the weakening takes much more slip, up to 10 or 20 m. It is not clear why ours show so much variability or whether the difference in this regard between our results and theirs is meaningful. 5. One solution would be to publish this paper as a set of interesting observations and remove all the interpretation part so that does not appear in the literature as an apparently proven additional weakening mechanism without being really understood. Note that I in no way dispute that they have amorphous silica in their experiments. We first showed this in reference 28 in 1990. Importantly, however, it was not associated with weakening even though up to 40-50% of the material was amorphous silica. In fact, however, the slip velocities were so low that the temperature cannot have risen more than a few degrees C at the most, so it is at least possible that the modest temperature rises from their experiments could affect the amorphous silica properties as they claim.

The disagreement with previous interpretations is based directly on detailed

characterization of the wear material. We stand by our interpretation and will not drop it from the manuscript. The conditions of experiments in the 1990 paper were low strain, low strain rate. Amorphous material is not all the same! It must be understood and characterized separately for different circumstances.

Details:

Line 11 Insert ref 5 and 6 along with 9 and 10, since in both papers we also show strengthening.

Modified as suggested

Line 11-12 They didn't find evidence for silica gel. Here at least they don't say we might be wrong about having it, although they do in their rebuttal!

We stand by our comments in the rebuttal that the observations presented by Tullis and collaborators do not uniquely identify the presence of silica gel, there are still alternative hypotheses which explain Tullis and collaborators' observations equally well, and also explain our observations.

Line 34-35 "friction decreased gradually" – not quite true – if they mean with slip then that is not the case for the 1 mm/s data, and if they mean gradually with slip velocity, then again it is not true for the 1 mm/s data. Might be better to remove the word "gradually" and add "although the 1 mm/s data don't fit the systematic pattern." This could be done by moving the phrase on line 36 to the beginning of the sentence and then adding this caveat just before the end of the reordered sentence. Honesty is always the best policy – it is better for the authors to point it out than to have it "discovered" by someone who looks at the figure carefully!

We have changed "decreased from" to "evolved from" to include the variability in experiment 123.

Line 38 This reference to a pseudo-analytical solution is confusing. I assume they simply use the truly analytical solution of Carslaw and Jaeger which is exact for a half space, a situation that they of course do not have, but it should be locally OK even in their radial symmetry small diameter sample. I suggest replacing "pseudo-analytical" with "half-space analytical"

Modified as suggested

Line 42-43 In looking at the temperatures in Figure 1B I believe there may be an error in labeling. The T's are described as being "Temperature Increase" but I think they must be "Temperature" with the boundary condition at the ends being perhaps 25 deg.

Corrected, thank you.

Seem very unlikely to me that the temperature rise near the steel can be 25 deg. I also find their contour line designations poorly chosen. The yellow lines are hard to distinguish from each other, as are the red ones. Since this is all in the region of high temperature gradients, it makes it hard to know how hot it got as a function of distance from the sliding surface, something that is important for their thermal interpretation for the cause of

weakening.

The key take-home point from this figure is that the peak temperatures are too low to melt quartz (~1700°C).

Also the caption to Figure 1B should say that the results are from the FEM modeling (i.e., not the C&J solution). And perhaps replace in that caption the word “anomaly” with “distribution”

Modified as suggested

Another thing that needs to be specified is whether they took the time- varying experimentally-observed value of the friction coefficient as the input for the FEM model. I assume this to have been the case, since friction varies a lot in experiment 96. They should also say what they did in the case of the C&J analytical solution, since that assumes a time-invariant heat source. Maybe they somehow tried to account for that, but it's not simple to do so they need to say more if they somehow tried to do that.

The FEM model inputs include shear stress and velocity. We imported the experimental data in Comsol Multiphysics as .txt files. Then, two separate interpolation functions (of time) are created using the nearest neighbor interpolation method so that the software can handle the input data. The two functions are then multiplied to obtain the heat source, positioned only in the model nodes coinciding with the rock-rock sliding interface.

In the case of C&J analytical solution, we calculated the temperature increase with the following code:

```
const=rho*cp*sqrt(pi*ath);  
  
for jj=1:size(time,1)  
  
if jj==1 dT(jj,1)=(shear1(jj,1).*1e6.*vel(jj,1).*sqrt(time(jj,1)))./const; dT=real(dT);  
  
T(jj,1)=dT(jj,1);  
  
else dT(jj,1)=(shear1(jj,1).*1e6.*vel(jj,1).*(sqrt(time(jj,1))-sqrt(time(jj-1,1))))./const;  
dT=real(dT);  
  
T(jj,1)=T(jj-1,1)+dT(jj,1);  
  
end  
  
end  
  
T=T+T0;
```

(shear stress “shear1”, velocity “vel”, time are all experimentally measured)

In each timestep, we assumed a constant heat source and then calculated the temperature increment. In each step (except the first one) the temperature increment was added to the one in the previous step. In the end, to obtain the temperature evolution we added the initial temperature.

Perhaps an even more interesting question is how the temperatures varied as a function especially of time but also position, say along a line perpendicular to the sliding surface taken at the radius of the effective velocity, at about 2/3 of the sample radius. They could show a series of profiles taken at different slip displacements for experiment 96. The point here is that since the input heat source (from the decaying friction) drops rapidly in the first 0.3 m of slip, it seems possible that the temperatures were actually interestingly higher during the first 0.3 m of slip than they are at the end of the experiment, which lasts 30 seconds, and is the time for which Figure 1B shows the final FEM results. This should be an easy output to get from the FEM calculations.

The plots on the right show the FEM modelled temperature versus time and displacement in three radial positions (points): the inner radius (coinciding with the rotation axis), the equivalent radius (at about 2/3 of the sample radius), and the outer radius.

The reviewer suggestion is correct as the decrease of friction coefficient can result in a slightly delayed slope decrease in the calculated temperature profile, after the abrupt increase occurring within the first second (ca. 0.1 m of slip) of this experiment. Also, as temperature increase is sensitive to the frictional power, the simultaneous slip rate increase and friction coefficient decrease result in a temperature rise lower than expected at the very beginning of the experiment.

Line 51-52 These localized shear surfaces are pretty interesting, but are very poorly imaged in Figure 2B. It would certainly be nice to get a better image of them. I note that the face-on images of them from taking sample 97 apart has relatively low magnification SEM images of these surfaces. In the image files I have attached to this review, note that the flow features in what we believe to be silica gel are seen in much higher magnification SEM images. Furthermore we don't see them on the smooth shiny surface of the sample (that we attribute to being a smeared-out layer of silica gel), but only in the edges of the original pits (like those they show in their Figure 1A) where the gel has been dragged over the edge of the plateau and into the pits. The amount of slip that is required to create this thin layer of continuous (except for the pits) silica gel is tens of meters. Only two of their

experiments (98 and 102 slid that far – Table 1). Although they studied the wear materials from those samples in detail (* in Table 1) they show no images of those samples in Figure 2. Were those samples taken apart for SEM imaging as was 97 or were they only studied in cross section as was experiment 101? They may have no silica gel in their samples, but it still seems that they could have! Amorphous silica in the presence of water becomes hydrated (it is used as a desiccant) and becomes a silica gel – its properties depend on how much water is in it.

It is not possible to examine the same intact sample in cross section and in plan view, as these preparations require different timing of epoxy stabilization and different geometries of preparation. Thus, we prepared two identical experiments (97 and 101 at 1 cm/s) in order to have equivalent observations in both dimensions. The other slip rates were examined only in plan view. Our cross-sectional view of sample 97 shows multiple localized slip surfaces embedded within a particulate layer. This is consistent with the interlayered powder and shiny plates we observed in plan view on sample 101. TEM analysis of these plates in cross section (Figure 3C) shows that the shiny surfaces only ~1 μm thick are composed of packed nanopowder. We cannot comment on how this compares to Tullis and collaborators' result as they did not describe any cross-sectional structure or any particulate wear material, although they reported its existence in the 2006 USGS Grant Report.

Silica gels are a wide variety of materials of diverse chemical composition and internal structure. Hydrophylic properties depend on the precise pore size, only silica gels with very specific internal structure have this property. The gel itself is the desiccant, amorphous silica in other forms does not become a gel due to hydration.

Lines 58-62 This interpretation of their localized slip surfaces is the same as our interpretation of our entire smoothed surface. As described above, and shown in the attached images, one of which is included in reference 3, the gel flow features cannot be seen on the smooth surface itself, but only near the edges of the pits where it get smeared partially into a hole. Because their shiny localized slip surfaces are more ephemeral and within the gouge layer rather than covering the entire sample surface, they don't have the opportunity to see the flow features that we observe at high magnification in SEM.

A ubiquitous observation (the shiny surface) should be explained by the ubiquitous material. Our TEM observations show that the nanopowder, which is abundant, is smeared into smooth surfaces. Neither the sliding surfaces nor the fragments of shiny surfaces within the powder layer showed evidence for silica gel.

Caption to figure 4 – Reference is made to HH, HV, VH, and VV in the caption, but in the figure HH and HV are not shown, but only the latter two. Presumably HH and HV should be removed from the caption. And to make a more clear connection to the supplement, I assume that these are the items within the parenthesis in the notation in the supplement, such as Z(HV)Z and Z(VV)Z. Some rewriting would make this more clear.

Modified as suggested

Lines 75-97 There is a rather long discussion of Raman spectroscopy results, but the bottom line of that seems to not end up contributing a lot to the paper, even though they did a bunch of work on it. I wonder if putting nearly all of this material in the Supplement and including only a summary like that in lines 95-97 in the body of the paper, but include

lines 98-110. It's fine to leave it all in the paper, but it does make it longer and distracts perhaps a bit from the overall paper's impact. I found the anisotropy the most interesting part of this. If it is possible to say more about the nature of the preferred bond orientations from this and thus the nature of the strain in the amorphous silica, or something else besides the fact of anisotropy and the obvious "perhaps related to the shear- parallel smearing observed in the striated plates and on the slip surface" (lines 91-92), it would be interesting.

Several essential insights from the Raman spectroscopy are relevant to the interpretation of the wear material, even if further work is required to completely develop all possible implications. First, the lack of a dominant mode (Fig 4A) indicates that there is no dominant coordination number of oligomeric ring structures. This is in contrast to amorphous silica nanoparticles that are produced commercially by precipitation from a liquid or by fuming methods. The anisotropy is also very important and supports our argument for smearing of solid amorphous material during sliding. We were unable to find any previously published Raman spectra which matches our wear material, which alone is an interesting attribute and worth reporting.

Line 111-121 They should say something about what the FT-IR spectra say about the quantity of water present, not just that "adsorbed water" was there. Often this is how FT-IR spectra are used – for quantitative analysis, not just existence of species. Given the debate about the role of water as combined with amorphous silica (i.e. is it "silica gel" or not) knowing how much water is there is very important. Presumably the amorphous nanosilica has a known quantity of water in it, so some quantitative comparisons could be made. I don't know enough about FT-IR spectroscopy to know what wavelength range is typically used to infer water content or whether the appropriate ones are shown in Figure 4B. Something should be said about this issue. I note in their response to my earlier comment on lines 88-94, that they say they are unable to determine the quantity of water in the sample from the FT-IR measurements, due to "uncertainties in path length due to thickness and packing variations." This is totally reasonable response to my question – some explicit statement to this effect should be made in the paper to quell the sort of question that I was asking in this paragraph (I'd forgotten their response to my earlier comment).

Using FT-IR to measure the water content quantitatively requires ideal sample preparation and preservation, and a thorough understanding of the FT-IR spectra of the other component (in our case, the silica). Without the latter, we are unable to quantitatively identify the water content. Regardless, no predictions have been made regarding how much water a solidified silica gel should have versus a hydrated amorphous nanopowder, so there would be nothing to compare to. Further characterization is clearly required.

Lines 113-115 This sentence says the wear material differs from the commercial amorphous nanosilica. These differences are not described in the text, but only in the Supplement. That location should be referred, and if they feel some part of that is particularly interesting it should be specifically mentioned in the text here.

Modified as suggested

Supplement final paragraph – the peak at ~694 is twice referred to as being at ~695 – change that to 694 for clarity as is labeled in Figure 4B.

The wavenumbers of the peaks are reported as measured – the peak location between analyses in different labs is not actually repeatable to a the last decimal /cm between samples so these peak locations are identical within error and are equated in the literature as being the same peak. A comment on this point has been added to the supplementary material.

Lines 119-121. This sentence may indeed be true, but the combination of FT-IR and Raman spectra by no means proves this postulated configuration to be the case. As far as I can see it only shows that the water content is below the Raman detection limits, not where it resides. It is important to state the Raman detection limits.

We do not have precise detection limits for the Raman spectroscopy because due to the configuration of our powder sample we do not know exactly the path length of the laser through the analysed material. However, the general statement that the FT-IR is ~4 orders of magnitude more sensitive is robust and sufficient for this purpose. The interpretation that the water resides on the surfaces of the particles is supported by the abundance of adsorbed water (see FT-IR spectra, Fig 4B inset)

Table 2. It is not clear what the * for some table entries refer to.

The caption now indicates that one or more of SEM, TEM, Raman and FT-IR were used to analyze the wear material of the * experiments.

Line 123-125 The “hydrated surfaces” part of this is important in their argument, but all that I see that addresses this is the label of “3360 adsorbed H₂O” within the inset of the top half of Figure 4B and perhaps the “3660 silanol” in the same inset and the “976 silanol bending” of the larger lower inset in Fig. 4B. Typically silanol groups (Si-O-H) are found on the surfaces of quartz, e.g. However, silanol can exist in the interior of the gel globules, not just on the surface and the FT-IR seem not to be distinguishable. I’m no expert, but a quick web search turned up this reference, for example, that seems to me to call into question the uniqueness of your interpretation of your FR-IR spectra. This needs more discussion.

Davydov, V.Y., Kiselev, A.V., and Zhuravl, L.T., 1964, Study of the Surface and Bulk Hydroxyl Groups of Silica by Infra-red Spectra and D₂O-exchange, Transactions of the Faraday Society, 60, 2254-2264.

Our data clearly show that the all the analyzed amorphous silica types have much more silanol than the powdered novaculite. This may include some OH- groups in the interior of the particles, but not enough to be detectable by Raman spectroscopy. We thank Prof. Tullis for the reference. We have not yet reported drying experiments on the wear material that would be comparable to the experiments of Davydov et al. (1964) but are considering these for future work. Our data is reported as “consistent with” our structural model (Line 121-124) – which will be certainly refined as we and hopefully other workers continue to expand the available observations of silica wear material.

Line 124-138 In the case of study 28, as far as we were able to ascertain, a layer of quartz gouge as close to the localized slip surface as we could get was comprised of up to 40-50% amorphous silica. While this is clearly not a continuous layer of 100% amorphous material, it should be significantly weaker than pure comminuted crystalline quartz if amorphous silica is weaker at room temperature and low slip rates. However, it was not.

This contributed to our conclusion that the thixotropic properties of silica gel, not the mere presence of amorphous silica, contributes to the high speed weakening.

This is consistent with, and explained by, our interpretations. We also note that the Yund et al. experiments (reference 28) produced a continuous matrix of amorphous material containing fragments of starting gouge. This is (in our view) not equivalent to our experiments so the comparison may not be valid (but should be tested).

Lines 139-146 I totally agree with this conclusion, The explanation for it is what is in question.

Agreed.

Lines 146-148 Here is where we disagree. Based on our experiments, we believe that the presence of water within the amorphous silica created a silica gel that has flowed at high strain rates as a thixotropic material. So we both call upon water to contribute to the material being weak, but we believe the material has undergone more continuous flow when it was deforming rapidly, where as they call upon particulate flow. In some sense this may be a matter of scale, since the flow of a silica gel must involve the relative motion of units within the gel. We are agnostic as to the nature of these units since we have no data that bears on it.

This study was motivated by a desire to provide that data, and we believe we have done so. The Tullis group showed no evidence of there being “units” (presumably colloids) within the flowy material; no attempt was made to demonstrate a structure consistent with gel in the original papers. We have also reviewed the definition of ‘gel’ and found that it does not describe the substance we have analyzed. We have been careful in the main manuscript to make direct comparisons to the interpretations of Tullis and collaborators only where a significant inconsistency with our data makes such comparisons necessary, but we have concluded that their use of the term ‘gel’ was perhaps premature given the available information.

They could be as small as Si-O₄ tetrahedra or as large as hydrated SiO₂ globules that would be small enough, perhaps 50 nm (?), that they would be indistinguishable in our SEM images in which the surfaces of the flowing gel look perfectly smooth.

We have shown that imaging our experimental samples on the same scale (and additional scales) as the Tullis group did not produce equivalent observations (Fig. X, Y, Z).

Lines 163-168 The qualitative role of water and temperature alluded to here is undoubtedly correct. What is unclear is whether the unknown water contents and modes temperature rises involved in these experiments are sufficient to cause crossings of the glass transition (their references (35, 36) refer to organic materials so offer no quantitative help here.

The role of water is still not understood quantitatively, but attempting to answer this question led me to a study of pressure-induced amorphisation in silicon (metal) leading to the survival of meta-stable amorphous material after depressurization (Deb et al. 2001 Nature). This offers another potential mechanism for the solid-state production of amorphous silica and has been added to the discussion.

Lines 178-190 I strenuously object to the use of the term hydrodynamic lubrication here and everywhere else in the paper. They are using a well-established term in the lexicon of machinery to refer to something totally different. If this term becomes accepted with this new meaning in the geologic literature we will have done a great disservice to clear communication. The accepted meaning of hydrodynamic lubrication applies to the situation where: 1) A lubricant layer, which must be a viscous fluid, separates the surfaces; 2) The fluid undergoes hydrodynamic flow in the space between the moving surfaces; and 3) the surfaces between which the fluid films move must be convergent. It is this convergence, combined with the hydrodynamic flow of the viscous fluid film, that builds up the hydrodynamic pressure within the fluid that holds the surfaces apart. In my previous review I objected to their use of this term, but muddied the waters by bringing up elastohydrodynamic lubrication which is also irrelevant here, but is not the term they are using. I note that reviewer number 2 also objected to the authors' use of the term hydrodynamic lubrication. That objection was more based on a question as to whether the water inferred to be at the particle boundaries acts as a lubricant. I share that concern somewhat, although they cite many references supporting the idea and it is probably correct. However, I still strongly object to using the term because it has another accepted meaning. There is not a thick enough layer of water there to cause the kind of hydrodynamic flow that this terminology implies, it is not thick enough to act as a viscous fluid, and the converging geometry required does not clearly exist. Some other terminology should be used.

We (again) respectfully disagree with Prof. Tullis. We hypothesize a layer of water (a viscous fluid) trapped between particles during inter-particle motion, which by geometric necessity, involves at least half of the surfaces converging at any given moment. The condition of hydrodynamic lubrication may be transient but the pressure exerted by this trapped and flowing layer on the surrounding particles meets the definition of hydrodynamic lubrication. As the interlayer water is no longer trapped after cessation of the experiment, we did not observe or imply any thickness of the water layer but perhaps our cartoon (Figure 6) has given the reviewer the impression that we mean a single molecule layer? In order to clarify this potential misconception, we have added the clause "of unknown thickness" to the text when we propose the trapped water layer.

Figure 4. Reviewer 2 likes this. It indeed makes clear what they have in mind as a model, and therefore has some value. What seems most speculative to me is what is described in the "A" part of the drawing. It is not clear to me that there is enough warming to lead to plastic flow, since the temperature rises are not large, but it is certainly possible. I also think it is likely that water in the bulk of the particles is likely to contribute to their ability to flow, i.e. it is silica gel. I have proof only of the bulk flow of the material from our SEM photos, not of the details of why it happens. However, it is clearly thixotropic since the textures show that it flows at high rates and is weak whereas it fractures and is strong when the rapid slip is over.

The behaviors Prof. Tullis describes could also be attributed to locally melted wear material. We reiterate that the glass transition temperature or pressure, and/or the melting conditions of the amorphous silica nanopowder are completely unstudied. It is well known that the average temperature of the slip surface, and the average stress, both are exceeded at asperity contacts by orders of magnitude (see discussion in manuscript).

A solid material (e.g. amorphous silica) with water dissolved or included in it is not, by definition, a gel. Some, but not all, gels are thixotropic. Some, but not all, thixotropic

materials are gels.

Lines 198-199 I do not believe that they have shown this. They have no evidence that silica gel is present, but they have none that it is not (this is of course hard to do). Our evidence that it is present, is primarily the flow features that clearly are seen in our samples as is shown in some of the figures that I have attached to this review (only one of which is published), and that the surface layer of our samples has approximately 2 percent water based on FRES analysis for hydrogen in this thin layer. Although they have done extensive, careful, and interesting imaging of their samples, as I discuss earlier, due to the low slip amounts of the samples they used to look at the sliding surfaces (at least as shown in the provided figures), the relatively low magnification of their submitted SEM images compared to ours, and the lack of favorable geometry to see flow features at the edge of their slip surface, they could well have missed the small but significant flow features contained in our sample that show flow occurred in our surface layer.

We stand by the sentence: “Our experiments were modeled after those which first reported silica gel (5; 6), but we have documented that the wear material produced in our experiments is not gel.” We believe we have met every reasonable standard of evidence in careful examination of multiple samples by several analytical methods, which significantly exceeds the standard of documentation (presented or otherwise) by the Tullis lab.

Line 199-201 I more or less agree with this sentence, but would be in perfect agreement if it says “material” rather than “particles.” Again the difference may be more in terminology than in substance. We chose to call this “silica gel” because it clearly flowed, contained amorphous silica and water, and behave thixotropically. This last property, thixotropy, we attribute to a competition between time-dependent healing and strain dependent disruption of bonds in the material. We chose not to discuss the possible temperature dependence of this process, because we didn’t think that the shear-heating temperature increase was enough to have a significant effect, but no one seems to have quantitative knowledge concerning that. Clearly it is above absolute zero, so time-dependent healing should occur.

Our observations did not show the flowy material. We only saw particles, smeared particles and flattened particles. Our interpretations of our own observations remain unchanged.

Line 207-208 Maybe “above the glass transition,” but as this T is not known, this remains speculative. If the behavior being weak can be taken as a proof of this, then I guess it may be true, but I’d rather see an independent determination that such a known temperature was crossed.

The amorphization could also have been caused by increasing pressure above the glass transition – we have inserted this into the sentence and added a citation to support it.

Lines 227-233 I think we all agree that “silica lubrication” may be important in earthquakes, although using this term is a bit vague. I still like “silica gel” better because it is more specific, but it may or may not apply to their results. “Silica lubrication” is a bit problematical as a term because people will immediately say, “how can quartz cause lubrication,” since silica means SiO₂. Also, one has to be careful not to end up getting

confused with lubrication via the use of silicone oil.

We decline to use the word “gel” because we have shown no evidence of “gel” in our experiments. “Silica lubrication” is more general and allows for the mechanism to be further described without replacing terminology in the future, in acknowledgement that all workers agree that the mechanism depends on silica. Readers should take care not to confuse silica with quartz (we have consistently referred to ‘amorphous silica’ in our text) or with silicone oil.

REVIEWERS' COMMENTS:

Reviewer #1 (Remarks to the Author):

This is the third iteration of this manuscript and it ripens like good wine. The authors have addressed the concerns raised by the reviewers in a detailed manner and yet again improved the manuscript (which was very good to start with!). Some parts in the discussion about the presence and absence of silica gel might be still controversial in the community, nevertheless the authors of this manuscript have characterized the wear products present in their experiments to the greatest detail available in published literature so far. This manuscript should be published so that future work can further refine our understanding of amorphous fault rocks. The reason for the presence vs. absence of silica gel will best be addressed by future studies.

One minor comment I have to the last version regards:

On lines 164 - 166 the authors state: Any defect which concentrates stress within the amorphous material plays a role during deformation which is analogous to (and can be modeled as) a dislocation in a crystalline structure (35).

I get what the authors want to communicate but I would phrase this sentence a little differently to prevent confusion (i.e. defect in amorphous material = dislocation). According to Falk & Langers (ref. 35) deformation in amorphous solid-like materials is mediated by shear transformation zones (STZ). STZ hence play the same role as dislocations in crystalline materials (agents of deformation) as correctly stated by the authors. However STZ and dislocations are fundamentally different, dislocations are line defects whereas STZ are point-like defects among other distinctions. It might be clearer to explicitly name STZ as the carriers of deformation in the amorphous material and state that both STZ and dislocations are carriers of deformation despite their different nature.

Anyway, given that this nit-picking detail is my only suggestion for improvement I strongly suggest that this manuscript gets published and doesn't go through a 4th round of reviews.

Reviewer #2 (Remarks to the Author):

The 3rd round review:

The authors have addressed all my concerns on the manuscript. This paper is now in good shape for publication in Nature Communications.

I only have one further concern about the flash temperature calculation. What is the M-value (number of asperities on the slip surface) or the asperity size you used in your calculation? They are not clear in the supplement. Your results (Figure S1 and Table 1) surprised me that the flash temperatures can be as high as 1000C for nanomaterial. Just from my experience, in order to get 1000C, the asperity size you used would be much larger than the particle size of nanosilica (~200 nm). This may be not consistent with your interpretation of particulate flow as one of the dominant mechanisms. If granular flow occurs, the asperity size would be smaller than or at most of the order of the particle size. This is simply because an asperity (contacting point of a grain) is unlikely to serve as asperity as the same grain moves/rolls to the next position. I suggest you use smaller asperity size, then you would have even lower flash temperature. Of course, this would not affect your interpretation and conclusion.

Looking at Tullis' review and the SEM images provided, it is very clear that some gel material was produced at their experiments. Shared with the same feeling as the authors, before seeing these pictures, I was not very convinced of the existence of silica gel either. As I said in my first round of review, I appreciate this work very much. Discrepancy can make us move forward!!

REVIEWERS' COMMENTS:

Reviewer #1 (Remarks to the Author):

This is the third iteration of this manuscript and it ripens like good wine. The authors have addressed the concerns raised by the reviewers in a detailed manner and yet again improved the manuscript (which was very good to start with!). Some parts in the discussion about the presence and absence of silica gel might be still controversial in the community, nevertheless the authors of this manuscript have characterized the wear products present in their experiments to the greatest detail available in published literature so far. This manuscript should be published so that future work can further refine our understanding of amorphous fault rocks. The reason for the presence vs. absence of silica gel will best be addressed by future studies.

One minor comment I have to the last version regards:

On lines 164 - 166 the authors state: Any defect which concentrates stress within the amorphous material plays a role during deformation which is analogous to (and can be modeled as) a dislocation in a crystalline structure (35).

I get what the authors want to communicate but I would phrase this sentence a little differently to prevent confusion (i.e. defect in amorphous material = dislocation). According to Falk & Langens (ref. 35) deformation in amorphous solid-like materials is mediated by shear transformation zones (STZ). STZ hence play the same role as dislocations in crystalline materials (agents of deformation) as correctly stated by the authors. However STZ and dislocations are fundamentally different, dislocations are line defects whereas STZ are point-like defects among other distinctions. It might be clearer to explicitly name STZ as the carriers of deformation in the amorphous material and state that both STZ and dislocations are carriers of deformation despite their different nature.

Revised the sentence to say: “Any defect which concentrates stress within the amorphous material plays a role during deformation which is analogous to (and can be modeled as) a dislocation in a crystalline structure, although these ‘shear transformation zones’ are not geometrically similar to dislocations.”

Anyway, given that this nit-picking detail is my only suggestion for improvement I strongly suggest that this manuscript gets published and doesn't go through a 4th round of reviews.

Reviewer #2 (Remarks to the Author):

The 3rd round review:

The authors have addressed all my concerns on the manuscript. This paper is now in good shape for publication in Nature Communications.

I only have one further concern about the flash temperature calculation. What is the M-value (number of asperities on the slip surface) or the asperity size you used in your calculation? They are not clear in the supplement. Your results (Figure S1 and Table 1) surprised me that the flash temperatures can be as high as 1000C for nanomaterial. Just from my experience, in order to get

1000C, the asperity size you used would be much larger than the particle size of nanosilica (~200 nm). This may be not consistent with your interpretation of particulate flow as one of the dominant mechanisms. If granular flow occurs, the asperity size would be smaller than or at most of the order of the particle size. This is simply because an asperity (contacting point of a grain) is unlikely to serve as asperity as the same grain moves/rolls to the next position. I suggest you use smaller asperity size, then you would have even lower flash temperature. Of course, this would not affect your interpretation and conclusion.

Looking at Tullis' review and the SEM images provided, it is very clear that some gel material was produced at their experiments. Shared with the same feeling as the authors, before seeing these pictures, I was not very convinced of the existence of silica gel either. As I said in my first round of review, I appreciate this work very much. Discrepancy can make us move forward!!

We have modified Figure S1 and moved it to the main text (Now Figure 2) to show the flash heating temperature calculated for a range of asperity sizes corresponding to the smallest and most abundant at the scale of asperity size = single ~10 nm particle (Fig 3, $M = 5E7$) up to the scale of the ~30 μm particle clumps (Fig 2, $M = 600$). Figure 2 now shows that flash heating temperature is expected to reach ~1000°C for asperities larger than a few microns, and if the ~30 μm particle clumps act as asperities, the temperature might exceed cristobalite melting temperature for slip rates of a few cm/s. This is consistent with the slip rates of the Tullis et al. experiments, therefore we maintain our position that flash melting has not yet been ruled out as an explanation for the “flowy” textures produced by Tullis' group and displayed in the review comments.